# Harmonic amide bond density as a game-changer for deciphering the crosslinking puzzle of polyamide

Yu-Ren Xue[1,2,4], Chang Liu [1,2,4], Zhao-Yu Ma[1,2], Cheng-Ye Zhu[1,2], Jian Wu[3], Hong-Qing Liang [1,2] ✉, Hao-Cheng Yang[1,2], Chao Zhang [1,2] & Zhi-Kang Xu [1,2] ✉

It is particularly essential to analyze the complex crosslinked networks within polyamide membranes and their correlation with separation efficiency for the insightful tailoring of desalination membranes. However, using the degree of network crosslinking as a descriptor yields abnormal analytical outcomes and limited correlation with desalination performance due to imperfections in segmentation and calculation methods. Herein, we introduce a more rational parameter, denoted as harmonic amide bond density (HABD), to unravel the relationship between the crosslinked networks of polyamide membranes and their desalination performance. HABD quantifies the number of distinct amide bonds per unit mass of polyamide, based on a comprehensive segmentation of polyamide structure and consistent computational protocols derived from X-ray photoelectron spectroscopy data. Compared to its counterpart, HABD overcomes the limitations and offers a more accurate depiction of the cross-linked networks. Empirical data validate that HABD exhibits the expected correlation with the salt rejection and water permeance of reverse osmosis and nanofiltration polyamide membranes. Notably, HABD is applicable for analyzing complex crosslinked polyamide networks formed by highly functional monomers. By offering a powerful toolbox for systematic analysis of cross-linked polyamide networks, HABD facilitates the development of permselective membranes with enhanced performance in desalination applications.

Polyamide membranes have garnered significant attention for their selective separation efficiencies across a wide range of applications, including gas purification, desalination, and drug concentration. This popularity stems from their highly adjustable network structure, allowing for property customization to meet specific separation requirements[1-4]. Typically, polyamide membranes are formed through interfacial polymerization, which involves the reaction between acyl chloride and amine monomers at the organic-water interface[5].

Desalination-oriented polyamide membranes for nanofiltration and reverse osmosis are primarily developed by reacting trimesoyl chloride (TMC) with piperazine (PIP) and m-phenylenediamine (MPD) on microfiltration or ultrafiltration substrates, respectively[5,6]. While the properties of the substrate and the testing conditions can influence the performance of the polyamide thin-film composite membranes, it is the permselective polyamide layer that plays a central role in achieving water/salt separation. The permeation of water or salt through a

[1]Key Lab of Adsorption and Separation Materials & Technologies of Zhejiang Province, MOE Engineering Research Center of Membrane and Water Treatment, Department of Polymer Science and Engineering, Zhejiang University, Hangzhou 310058, China. [2]The "Belt and Road" Sino-Portugal Joint Lab on Advanced Materials, International Research Center for X Polymers, Zhejiang University, Hangzhou 310058, China. [3]Department of Chemistry, Zhejiang University, Hangzhou 310058, China. [4]These authors contributed equally: Yu-Ren Xue, Chang Liu. ✉e-mail: liang.hongqing@zju.edu.cn; xuzk@zju.edu.cn

polyamide thin-film composite membrane is a complex problem that necessitates simplification. Researchers consistently direct their attention towards polyamide nanofilm and have been actively exploring various strategies, including the modification of monomers, solvents, and polymerization conditions, to regulate and customize the crosslinked networks of polyamides to enhance their separation efficiencies[7–9]. Consequently, it is indispensable to analyze the crosslinked networks of permselective polyamide nanofilms and establish their correlation with desalination performance[10].

The degree of network crosslinking (DNC) is a commonly used structural parameter to depict the polyamide networks formed by trifunctional and bifunctional monomers[11,12]. The polyamide networks are typically categorized into crosslinking structures and linear structures, wherein the proportion of crosslinking structure is defined as DNC[12]. Representing the branching degree of polyamide, DNC provides a qualitative depiction of the compactness of polyamide networks. However, it has become increasingly challenging to establish a reliable and consistent correlation between DNC and water permeance or salt rejection in recent desalination researches[8]. This primarily stems from the oversight of the significant number of terminal groups introduced by the emerging regulated methods of interface polymerization. These methods, such as free interfacial polymerization[13,14], interlayer construction[11,15], control agent addition[16,17], and reactive monomer spraying[18,19], inhibit the diffusion of amine monomers and accelerate the hydrolysis of TMC, resulting in large residue of carboxyl groups in the polyamide network. On the other hand, conducting interfacial polymerization at low temperatures introduces amino groups into the polyamide network, as evidenced by the positive charge on one side of the polyamide membrane[20]. These terminal groups significantly influence the

ionization behavior[21] and electrical properties[22,23] of polyamide, ultimately affecting the rejection of high-valence ions[24,25]. Regrettably, these terminal groups were ignored in the calculation process of DNC, resulting in abnormal DNC values below 0% or exceeding 100%. Moreover, the calculation of DNC usually relies on X-ray photoelectron spectroscopy (XPS) data[11,12], which employs the charge shift of the full spectrum, theoretical binding energy (BE), and peak fitting primarily based on the 1$s$ peak of adventitious carbon[7,26]. This approach, however, is not ideally compatible with carbon-based organic polymer materials with complex chemical environments[27,28], which further undermines the credibility of DNC as a reliable parameter for characterizing polyamide networks. On the one hand, the chemical environment of C in polyamide is complex, and the peaks in XPS are very wide. The highest position of C peak is usually the superposition of the peak positions of skeleton C and oxygen-containing C, which is difficult to confirm as the C 1$s$ BE commonly used for calibration (284.8-285.0 eV). On the other hand, XPS is difficult to avoid the interference of exogenous C, which has a wide range due to its poor electrical contact with the sample and will interfere with the peak shape of C. It is challenging to determine the extent to which the C 1$s$ spectrum is affected by the presence of adventitious carbon. Therefore, conducting charge shift with C is likely to result in the peak of N and O not appearing in the theoretical BE. In this case, it is easy to obtain inaccurate peak fitting results by directly using the theoretical peak position for peak fitting. Nonetheless, XPS remains an irreplaceable and high-precision method to analyze the element composition and chemical environment of the 10 nm region on the surface of polyamide selective layer. In light of these challenges, it becomes imperative to put forth a more accurate parameter and data analysis method

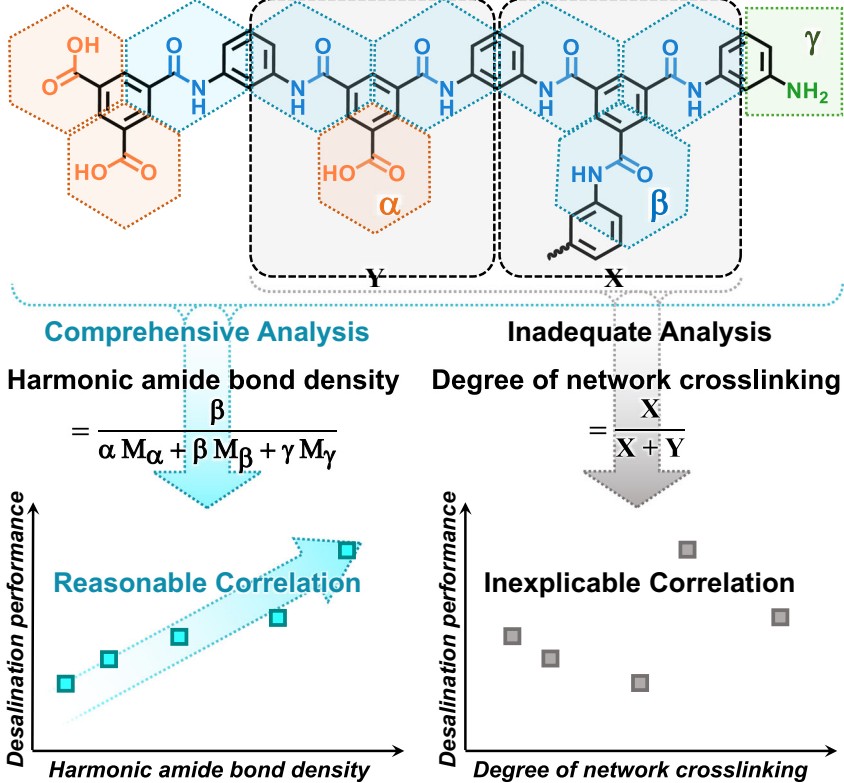

**Fig. 1 | Comparison of harmonic amide bond density (HABD) and degree of network crosslinking (DNC) in analyzing the structures and describing performances of polyamide membranes.** Here, α, β and γ are carboxyl structure cell, amide bond structure cell and amino structure cell in HABD method. X and Y are crosslinking structure and linear structure in DNC method. HABD covers terminal groups that cannot be considered by DNC, and allows comprehensive analysis of polyamides, resulting in a credible and reasonable correlation with the core performance of desalination polyamide membranes (with MPD-TMC polyamide as an example).

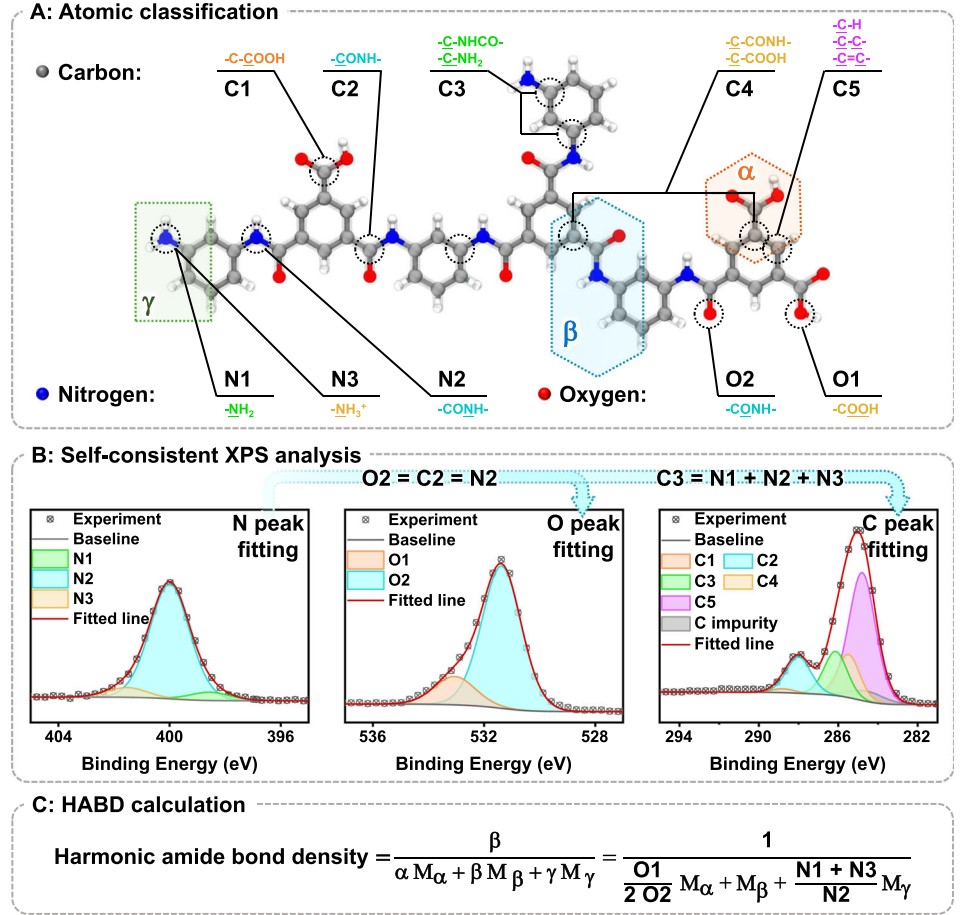

**Fig. 2 | Graphical representation of analyzing the polyamide structure and calculating HABD using MPD-TMC polyamide as an example.** Atoms of polyamide are classified into 10 types based on their BEs and the structure cell segmentation (**A**). Some types of atoms should satisfy the cross-peak self-consistency in XPS peak fitting process (**B**). Harmonic amide bond density can be calculated by the proportion of atoms (**C**). The polyamide examples were synthesized according to the following conditions: [MPD] = 20.0 g L⁻¹, [TMC] = 1.5 g L⁻¹, reaction time = 120 s. XPS test measured through Thermo Scientific K-Alpha+ at Al Kα line (1486.6 eV, 15 mA×15 kV), the vacuum level is 5×10⁻⁹ mbar, X-ray beam spot is 300 μm, total X-ray irradiation time is 154.1 s (68 s for full spectrum, 28.7 s for C, 30.2 s for O, and 27.2 s for N). Peak fitting was based on the recommended peak position in Supplementary Table 1[11,34,35].

that can effectively depict the crosslinked polyamide networks and guide the structural optimization and performance enhancement of polyamide membranes.

Herein, we propose a comprehensive structural parameter, the harmonic amide bond density (HABD, mmol g⁻¹) along with a matching XPS data processing protocol. Unlike the conventional approach that divides polyamide networks based on repeating units, we introduce a new division based on the characteristic groups present in polyamides. This allows us to account for all possible terminal groups and provide a more comprehensive depiction of the crosslinked polyamide networks. Additionally, our XPS data processing method precisely determines the proportion of atoms in diverse chemical environments and cross-peak self-consistency enabling accurate calculation of HABD. Through our investigations, we find that HABD accurately describes the network structure of two widely used desalination polyamide membranes: MPD-TMC polyamide and PIP-TMC polyamide. Importantly, we established a credible correlation between HABD and the core performance of desalination membranes. Our findings show that increasing HABD leads to a consistent decrease in water permeance and a significant increase in salt rejection, in line with the expected structure-performance relationship in desalination membranes. Additionally, HABD proves applicable to polyamides formed by highly functional monomers[29,30], expanding its utility as a powerful research tool for analyzing the water-salt separation performance of

desalination polyamides and designing customized crosslinked polyamide networks.

## Results

Crosslinked polyamides, such as MPD-TMC and PIP-TMC, can be divided into crosslinking structure (X), linear structure (Y) and two terminate structures with either amino or carboxyl groups[31], according to the repeating unit of the polymer. Conventionally, the calculation of DNC considers only the crosslinking structure (X) and linear structure (Y), while neglecting the terminal structures. Thus, DNC can be calculated using the following Formula[11,12]:

$$DNC = \frac{X}{X+Y} = \frac{4 - 2\frac{O}{N}}{1 + \frac{O}{N}} \quad (1)$$

The value of DNC is only meaningful in the highly crosslinked polyamide networks with rare terminal groups. It significantly limits the application range and decreases the correlation of DNC with desalination performance.

To address this issue, we propose a refined segmentation method for polyamide, dividing it into three basic structure cells: carboxyl structure cell α, amide bond structure cell β, and amino structure cell γ. Our framework is based on the presence of carboxyl, amide bonds, and amino groups in polyamide (Fig. 1 and Supplementary Fig. 1).

According to this framework, each structure cell is distinguished by the presence of exclusively one hydrolyzed carboxyl group, one amide bond, or one unreacted amino group. The amino structure cell represents the amine-terminated structure, while the rest of the repeating units can be further divided into combinations of carboxyl structure cell and amide bond structure cell. This approach comprehensively covers all atoms within the polyamide network by structure cells, including terminal structures often overlooked in traditional segmentation. The implementation of this segmentation method allows for a more precise depiction of polyamide networks, paving the way for a profound understanding of their properties and behavior. In polyamide crosslinked networks, the vacancy (free volume), volume and mass contributed by a structure cell are uniform across all positions. Since the HABD calculation encompasses all structural aspects of crosslinked polyamide, theoretically, deducing the polyamide fractional free volume (FFV) is feasible by calculating the van der Waals volume occupied by each structure cell. As an illustration, we set up five simulation boxes of MPD-TMC polyamides based on the different positions of α and β within the polymer chains, either as chain terminators or situated within the chain. The proportions of three structure cells α, β, and γ were fixed as 10%, 90% and 10%. (Supplementary Fig. 2). Regardless of whether α is distributed in the middle or at the end of the polyamide chain, crosslinked polyamide network boxes with identical structure cells consistently have a fractional free volume of about 34%[32,33]. Simultaneously, by subtracting the van der Waals volumes occupied by all structure cells from the total volume of the simulation boxes, we derived the FFV as predicted by HABD, which matched the FFV obtained from Monte Carlo sampling, consistently at 34%.

Based on the one-to-one correspondence between the proportion of structure cells and the fractional free volume, the structure of polyamide can be quantified and correlated with the performance of polyamide membranes. We define and compute the harmonic amide bond density (HABD, mmol g⁻¹) to further quantify the structural characteristics of crosslinked polyamides:

$$\text{HABD} = \frac{\beta}{N_A m} = \frac{\beta}{\alpha M_\alpha + \beta M_\beta + \gamma M_\gamma} \tag{2}$$

Here, $N_A$ is the Avogadro constant, m is the mass of polyamide network, $M_\alpha$, $M_\beta$, $M_\gamma$ are the molecular weight of relative structure cells, and α, β, γ are the count of relative structure cells.

The concept of HABD shares similarities with the number of crosslinking nodes in rubber and is inherently adaptable for describing the structural characteristics of randomly crosslinked networks. In the case of polyamide, the conversion of acyl chloride and amino groups into amide bonds occurs progressively during interfacial polymerization, leading to an increase in HABD. Consequently, the crosslinked network becomes denser, resulting in higher rejection and lower water permeance. When the polyamide is fully crosslinked, the value derived from Formula 2 is equal to the reciprocal of the molar mass of the amide bond structure cell, yielding values of 9.3 mmol g⁻¹ and 10.5 mmol g⁻¹ for MPD-TMC and PIP-TMC, respectively. Conversely, an inadequate reaction and hydrolysis cause an increase in the number of amino and carboxyl groups, leading to a decrease in HABD, accompanied by changes in membrane performance. Therefore, HABD exhibits a strong theoretical correlation with the performance of polyamide membranes.

To accurately determine the values of HABD, we propose a matching XPS data processing method with MPD-TMC polyamide as an example (Fig. 2 and Supplementary Fig. 3). Firstly, we categorize the atoms of polyamide based on the structure cell segmentation (Fig. 2A and Supplementary Table 1). Nitrogen (N) atoms are divided into amino N atoms (N1, in amino group), amide N atoms (N2, in amide bond), and protonated amino N atoms (N3, in amino group), while Oxygen (O) atoms are categorized into non-amide O atoms (O1) and amide O atoms (O2, in amide bond). Carbon (C) atoms are also divided into carboxyl C atoms (C1, in carboxyl group), amide C atoms (C2, in amide bond), N-related C atoms (C3, connected to the N atom, but not in amide bond), and O-related C atoms (C4, connected to O atom, but not in carboxyl group) and common C atoms (C5).

Subsequently, we utilize the N element (specifically, the BE of N2) for the charge shift of the full-spectrum. The N peak in polyamide is mainly contributed by the N atom of the amide bond, which is minimally affected and interfered by external impurities, making its N2 peak the most easily identifiable. This step is beneficial for subsequent data analysis and peak fitting process, although it does not significantly improve the accuracy of the analysis.

Then, the N peak is divided into three peaks based on the given relative peak positions of N1, N2 and N3 as outlined in the relevant research or handbooks (Fig. 2B and Supplementary Table 1)[11,34,35]. Peak fitting should be carried out under Gaussian–Lorentzian product pseudo-Voigt peak shape, with a Gaussian component comprising 60% ~ 80% of the peak, as Gaussian components are commonly observed in XPS peaks of polymers[36,37]. Throughout this process, attention should be paid to constraining the peak positions of the three N segments within the recommended range while ensuring that their full width at half maxima (FWHM) are similar.

The peak fitting of C and O is carried out with strict adherence to the cross-peak self-consistent relationship[38], in addition to considering curve shape, peak position, and FWHM. The proportion of O2 and C2 can be determined based on the cross-peak self-consistent relationship between the elements within the amide bond or amino in polyamide. In a single amide bond, there should be one C atom, one O atom, and one N atom, resulting in equal quantities of C2, O2, and N2. In MPD-TMC, each N atom is connected to a C atom in a benzene ring, implying that the number of C3 should be equal to the total number of N atoms (Fig. 2B):

$$C2 = O2 = N2 \tag{3}$$

$$C3 = N1 + N2 + N3 \tag{4}$$

The content of C1, C4, C5, and O1, cannot be directly calculated through the peak fitting results of N. However, self-consistent relationships also impose constraints on their cross-peak self-consistency.

In the absence of impurities, both C and O originate from polyamide. A carboxyl group consists of two O1 atoms, one C1 atom and is connected to a C4 atom. Similarly, an amide bond is also connected to a C4 atom, thus:

$$C1 = \frac{O1}{2} \tag{5}$$

$$C4 = \frac{O1}{2} + N2 \tag{6}$$

According to the division and statistics of our structure cells, we can also obtain statistical relationships:

$$C5 = \frac{O1}{2} + 3N2 + 2(N1 + N3) \tag{7}$$

The remaining atoms that cannot be separated should be considered as impurities, rather than forcibly attributed to specific atomic types such as C1, C4, C5 or O1. For example, if the total number of C after peak fitting exceeds the sum of C1-C5 under these constraint conditions, it indicates the presence of impurity C in the XPS test. Similarly, if the total number of C is insufficient to satisfy the Formula

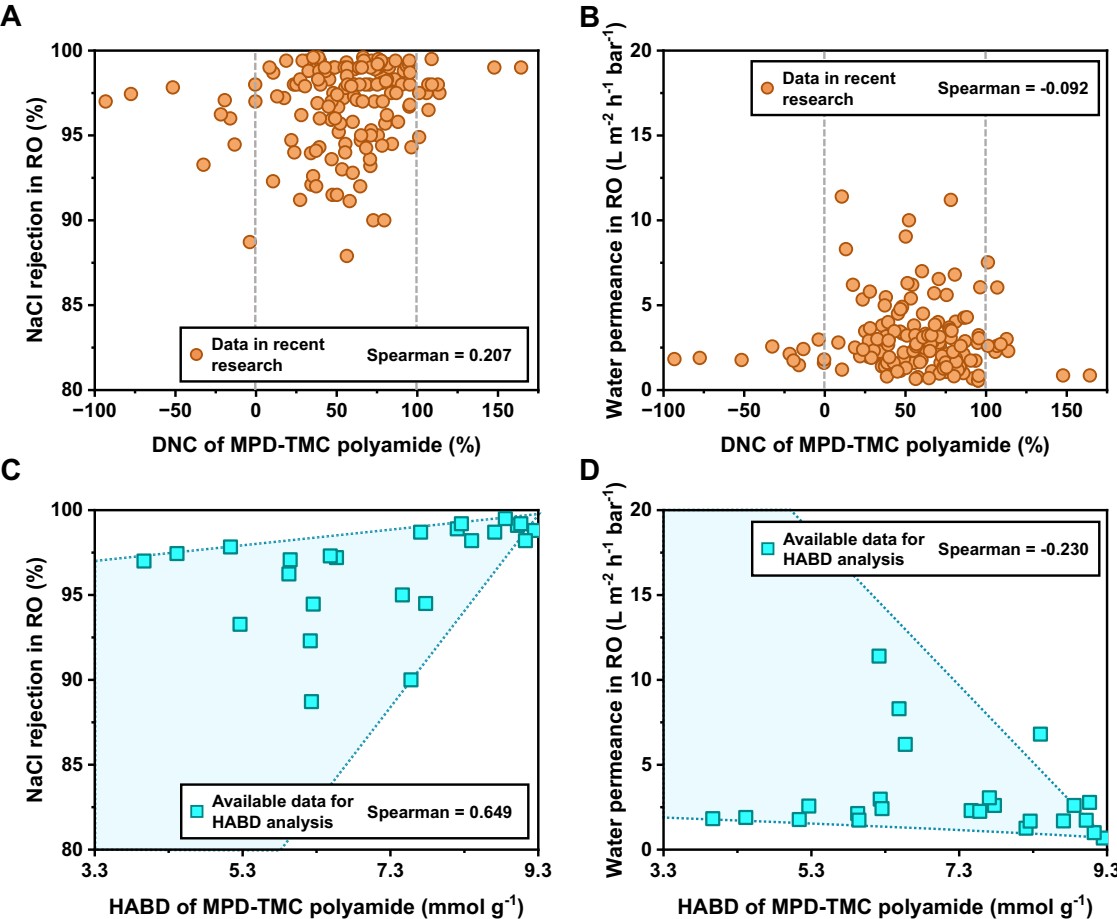

**Fig. 3 | The correlation analysis between the performance of reverse osmosis membranes with the DNC or HABD of MPD-TMC polyamides.** The DNC parameter does not show a reliable correlation with the NaCl rejection (**A**) and water permeance (**B**) of reverse osmosis polyamide membranes. In contrast, HABD demonstrates a reliable correlation with the NaCl rejection (**C**) and water permeance (**D**) of those membranes (The detailed statistical data in this figure is shown in Supplementary Tables 3 and 4).

5 ~ 7 after dividing the peak of O, it indicates that some impurities in O have been separated and attributed to O1.

Fig. 2B shows the peak fitting results of our self-prepared sample. We use N as the standard for correction. N peak is fully attributed to different N atoms within the polyamide. When the peak conditions and cross-peak self-consistency constraints are met, the C peak does not exclusively correspond to O1 ~ O2 and C1 ~ C5 atoms of polyamide. The remaining portions are treated as impurities with unspecific peak positions. The quantities of various atoms (Supplementary Fig. 4) must adhere to the self-consistent requirements in Formula 3-7, which is a crucial aspect of our method. Once the values of N1, N2, N3, O1, and O2 are obtained, HABD can be calculated by the Formula in Fig. 2C (the proof process is shown in Formula 2-1 ~ 2-4 in Supplementary Information). A similar analysis process for PIP-TMC polyamide is also provided (Section 2.4 in Supplementary Information).

Spearman's rank correlation coefficient (Spearman) is then employed to quantify the relationship between the HABD of polyamide membranes and their desalination performance metrics, such as salt rejection, water permeance, salt permeability coefficient, and water permeability (Formula 3-1 ~ 3-4 in Supplementary Information). A Spearman value close to -1 indicates a negative correlation, while a value approaching 1 suggests a positive correlation. On the other hand, a value close to 0 indicates little to no correlation between the variables[39,40].

We have selected representative studies on MPD-TMC type reverse osmosis membranes to examine the correlation between performance and structure parameters (DNC or HABD). Theoretical, considerations suggest that as interfacial polymerization progresses, the amino and acyl chloride groups convert into amide bonds resulting an increase in crosslinking structure X. Simultaneously, the crosslinked network of polyamide becomes more compact due to the reduction of the void and free volume[32,41]. These, in turn, result in higher NaCl rejection and lower water permeance of polyamide membranes[33,42,43]. Nonetheless, statistical analysis reveals that there is no significant correlation between the DNC of reverse osmosis polyamide membranes and their water permeance or NaCl rejection, since all correlation coefficients in Fig. 3A and B are all close to 0. This unexpected finding contradicts expectations as water permeances do not decline with the increase of DNC, and the NaCl rejections do not follow the expected upward trend. Similar issues arise in the correlation analysis between DNC and NaCl permeability coefficient (Supplementary Fig. 9). This situation demonstrates that disregarding terminal groups can lead to a significant deviation from the expected DNC value, especially for those polyamides with a proportion of terminal species more than 10%. The calculation method for DNC overestimates the value when considering the N atoms in the amino structure cell γ as part of crosslinking structure X. Conversely, when the carboxyl O on the terminal group generated by TMC hydrolysis is assigned to the linear structure Y, the calculated DNC underestimates the actual value. Therefore, the calculated DNC value consistently diverges from the actual value. These discrepancies are magnified by the presence of excessive carboxyl and

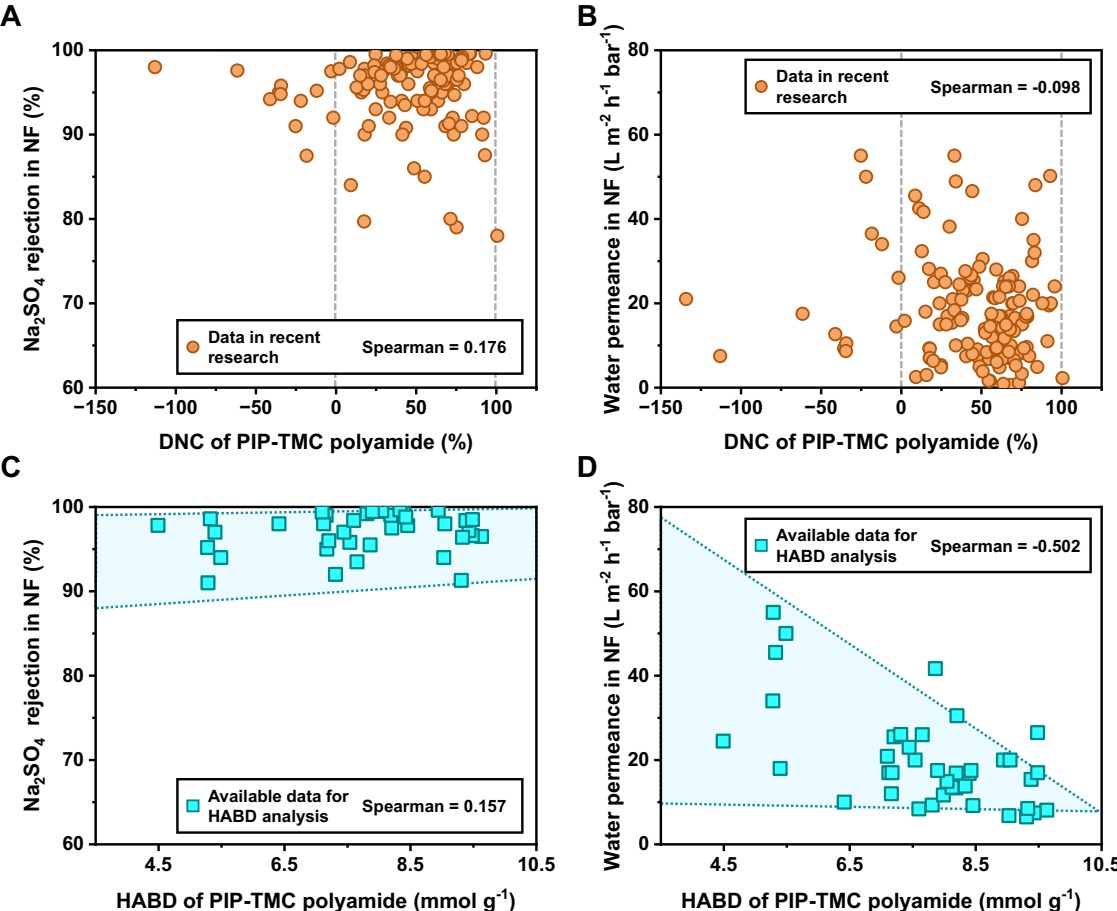

**Fig. 4 | The correlation analysis between the performance of nanofiltration membranes with the DNC or HABD of PIP-TMC polyamides.** DNC cannot form a reliable correlation with the Na$_2$SO$_4$ rejection (**A**) and water permeance (**B**) of nanofiltration polyamide membranes. In contrast, HABD demonstrates a reliable correlation with the Na$_2$SO$_4$ rejection (**C**) and water permeance (**D**) of those membranes. (The detailed statistical data in this figure is shown in Supplementary Tables 5 and 6).

amino groups, and can lead to abnormal DNC values even below 0% or exceeding 100%[15,18,44]. Such inconsistencies can significantly impact researchers' assessment of the structure-performance relationship in individual studies, and further cause confusion during a comprehensive review of the structural requirements of polyamide desalination membranes.

In contrast, when HABD is used to describe the crosslinked networks of polyamide, we observe a clear correlation between HABD and the performance of reverse osmosis polyamide membranes. Fig. 3C illustrates an overall improvement of NaCl rejection with the increase of HABD. The Spearman correlation coefficient for this relationship is 0.649, much larger than that of DNC. This indicates a strong positive correlation between HABD and NaCl rejection. Likewise, a discernible trend of decreasing water permeance and NaCl permeability coefficient is visible with the increase of HABD (Fig. 3D and Supplementary Fig. 10). Thanks to the comprehensive analysis of the polyamide structure, the calculated HABD consistently holds physical significance regardless of the proportion of terminal groups. The increase of terminal groups only decreases the value of HABD but will not cause it to become negative. This ensures that HABD can work for loose polyamides with rich terminal groups. Furthermore, HABD more accurately describes the correlations between crosslinked networks and performance. The closer the value of HABD is to 9.3 mmol g$^{-1}$, the fewer carboxyl and amino groups are present in polyamide, indicating a higher conversion to amide bonds during interfacial polymerization. Crosslinked polyamide membranes with higher HABD thus exhibit

reduced free volume and pore size resulting in higher NaCl rejection and lower water permeance.

Our analysis then extends to PIP-TMC type nanofiltration membranes. In this case, the correlation between crosslinked networks and nanofiltration performance becomes more challenging due to the contribution of both pore size and charge properties of polyamide membranes. A denser network provides a higher blocking ability to salt ions owing to the size exclusion effect, while the presence of abundant charge on the membrane surface can also block high-valent salt ions through the Donnan effect. Most PIP-TMC polyamide membranes possess a substantial number of hydrolyzed TMC terminal structures and exhibit negative charges, exacerbating the difficulty to establish the correlation between crosslinked networks and nanofiltration performance. Therefore, it is crucial to conduct a comprehensive analysis of the structure of PIP-TMC polyamide to gain a better understanding of its nanofiltration behavior.

As mentioned earlier, DNC is not suitable for dealing with polyamide with numerous terminal groups. The calculated DNC values for polyamides are mostly close to 0 or even negative. Fig. 4A and B (and Supplementary Fig. 11) show that DNC again fails to establish a reliable correlation with nanofiltration performance and produces a significant amount of calculated data beyond the definition of DNC. Conversely, the Spearman correlation coefficients of HABD for Na$_2$SO$_4$ rejection (Fig. 4C) and water permeance (Fig. 4D) are larger than those of DNC, indicating a significant correlation between performance and HABD. It should be noted that the correlation coefficients of HABD for Na$_2$SO$_4$

**Fig. 5 | HABD is applicable for analyzing complex crosslinked polyamide networks formed by highly functional monomers. A** TAM-TMC polyamide, **B** TAME-BTEC polyamide, **C** PEI-TMC polyamide.

rejection (Fig. 4C) and $Na_2SO_4$ permeability coefficient (Supplementary Fig. 12) are not as strong as those observed for water permeance. This discrepancy is primarily attributable to the influence of charge effects. As the amide bond density decreases and the number of terminal groups increases in the crosslinked network, PIP-TMC polyamide membrane exhibits higher water permeance. This is expected because a more loosely crosslinked polyamide network with increased terminal groups provides larger voids and facilitates the transport of water molecules. However, the change in $Na_2SO_4$ rejection is not significantly affected by the variation in HABD. This observation can be explained by the presence of rich charges on the membrane surface, which blocks high-valent salt ions through the Donnan effect as mentioned. To quantify terminal groups and analyze their ionization behavior, we have introduced two analogous parameters, namely the harmonic carboxyl density (HCD) and the harmonic amino density (HAD). These parameters, reflecting the contents of carboxyl and amino groups per mass of polyamide, are essential in accurately representing the surface charge of polyamides (Section 4 in Supplementary Information). It was demonstrated that PIP-TMC polyamide with higher HCD exhibits a stronger negative charge and higher $Na_2SO_4$ rejection (Supplementary Table 9, Supplementary Figs. 17 and 18). This finding strongly supports the notion that incorporating strong negative charges is effective in rejecting divalent anions when designing nanofiltration membranes. Nonetheless, the rejection of nanofiltration membranes is contributed by two factors: the screening of pores brought by polyamide networks and the Donnan effect of surface charge. Therefore, it is challenging to establish reliable structure-activity relationships by isolating these two factors. We recommend the comprehensive use of our proposed structural parameters – HABD and HCD. Our method offers a possibility to comprehensively analyze the rejection of nanofiltration, by providing information on HABD for the compactness of crosslinked polyamide networks, as well as HCD and HAD for the charge density of the polyamide nanofilms. These structure parameters can be cross-referenced with other characterization methods, providing a valuable tool for analyzing polyamides.

Furthermore, it is widely accepted that water permeability, which is the product of water permeance and thickness of polyamide nanofilms, serves as an intrinsic performance parameter reflecting the water diffusion ability in polyamide nanofilms. We have counted the thicknesses of the polyamide nanofilms and calculated their water permeability in recent studies on RO and NF membranes (Supplementary Tables 7 and 8). Due to the insufficient data of reverse osmosis membranes to support reliable correlation analysis (Supplementary Figs. 13 and 14), we primarily focused on examining the correlation between DNC or HABD and the water permeability of PIP-TMC polyamide in the nanofiltration process. Compared to DNC (Spearman correlation coefficient = 0.011, Supplementary Fig. 15), HABD forms a more significant correlation with the water permeability of NF membranes. The Spearman correlation coefficient between HABD and water permeability is -0.298 (Supplementary Fig. 16), which indicates a strong negative correlation between HABD and water permeability of NF membranes. Crosslinked polyamide network with higher HABD exhibits reduced fractional free volume and pore size resulting in lower water permeability, which is in line with theoretical expectations. It is worth noting that HABD can relatively reasonably describe the water permeance in polyamide membranes. Regardless of whether the thickness of the polyamide selective layer is considered, the water transport capacity of polyamide selective layer will increase with the decrease of HABD. This result strongly supports current research, which indicates that the thickness of polyamide is not a decisive factor for water permeance[32,45].

It is important to acknowledge that the element proportions and N peak fitting results in the calculation of HABD for MPD-TMC and PIP-TMC polyamide were directly obtained from corresponding references (Supplementary Tables 4 and 6). A considerable portion of the studies presented in Figs. 3A, B and 4A, B do not provide complete XPS analysis results. As a result, we only considered studies that provided the necessary data for HABD calculation. Comparatively, the peak fitting results for C or O in the cited works were abandoned because most of them did not account for the cross-peak self-consistency relationship between atoms. Therefore, in our HABD calculation process, O2 in these works was directly considered equal to N2. In addition, the determination of atomic composition is greatly influenced by the XPS equipment and testing conditions[37,46] (Supplementary Tables 10 and 11). However, the current research on RO and NF lacks sufficient emphasis on XPS. The differences in separation testing conditions, as well as inconsistencies in XPS testing conditions are the reasons why it

is difficult to form a direct linear relationship between HABD and performance. In order to enhance the standardization of XPS testing and improve the accuracy of the peak fitting results and HABD calculation, we referred to the standards of International Organization for Standardization (ISO 20579), American Society for Testing Materials (E1078), and National Standard of the People's Republic of China (GB/T SJT10458-1993) and related researches[37,47,48] to provide a series of recommendations regarding sample preparation and testing conditions for analyzing the surface of polyamides (Section 5.2 in Supplementary Information). We strongly encourage researchers to refer to these standards and research guidelines before conducting XPS test.

Another significant advantage of HABD is its applicability to polyamide membranes formed by higher functional monomers. The core principle of using HABD lies in the segmentation of polyamide networks. To demonstrate this, we provide segmentation diagrams for three varieties of polyamides: tetra (4-aminophenyl) methane (TAM) with TMC (Fig. 5A), tetra (4-aminophenyl) ethene (TAPE) with 2,2′,4,4′-biphenyl tetradecyl chloride (BTEC) (Fig. 5B) and even polyethylene imide (PEI) with TMC (Fig. 5C). In fact, our method is adaptable for crosslinked polyamides formed by any two symmetric monomers, as long as those monomers can be evenly divided into several structure cells, and the number of structure cells is equal to the number of functionalities. We have identified several highly symmetrical monomers compatible with the HABD approach (Supplementary Table 12), which are also the preferred choice for interfacial polymerization[5,49].

While the examples of separation application using these monomers are insufficient for statistical analysis, we do provide the HABD calculation process for polyamide membranes with TAM-TMC polyamide (Supplementary Fig. 19, Supplementary Table 13, and Formula 6-1 ~ 6-3 in Supplementary Information), TAPE-TMC polyamide (Supplementary Fig. 20, Supplementary Table 14, and Formula 6-4 ~ 6-6 in Supplementary Information), TAPE-BTEC polyamide (Supplementary Fig. 21, Supplementary Table 15, and Formula 6-7 ~ 6-9 in Supplementary Information), PEI-TMC polyamide (Supplementary Fig. 22, Supplementary Table 16, and Formula 6-10 ~ 6-13 in Supplementary Information) as examples. Readers can utilize the recommended calculation process provided in this work to calculate and analyze their polyamide membranes.

## Discussion

In this study, we propose HABD as a comprehensive descriptor of the crosslinked networks of selective polyamide membranes. HABD incorporates a refined segmentation method for the crosslinked polyamide network, taking into account the carboxyl group, amino group, and amide bonds of polyamide as the basis for dividing structure cells. Additionally, we have introduced a corresponding XPS data processing method to acquire accurate HABD values, based on the inherent identity of the atoms between the structure cells. Compared to the conventional structural parameter DNC, HABD has demonstrated its ability to establish reliable and theoretically expected correlations with the performance of widely used desalination membranes formed by MPD-TMC and PIP-TMC Notably, HABD can be applied to analyze complex crosslinked polyamide networks formed by highly functional monomers. Researchers are strongly encouraged to embrace and evaluate the utility of HABD along with the provided data processing method. Adopting HABD as a standardized parameter can advance our understanding of polyamide networks and optimize their performance for various separation applications.

## Methods

### Materials

Trimesoyl chloride (TMC), piperazine (PIP), and m-phenylenediamine (MPD) were bought from Aladdin Industrial, China. Isopar H was purchased from Sharun Chemical Co., ltd., China. Polyethersulfone microfiltration membrane with an average pore size of 0.22 μm was provided by Haining Xindongfang Technology Co. ltd., China. All chemicals and materials were used without further treatment. The used ultrapure water was produced by an ELGA LabWater system (VWS ltd., High Wycombe, France) up to a conductivity of 18.2 MΩ.

### Preparation of polyamide sample for peak fitting

2.0 mL of aqueous solution of MPD or PIP ([MPD] = 20.0 g L$^{-1}$, [PIP] = 1.0 g L$^{-1}$) was filtered into the Polyethersulfone microfiltration membrane under 100 kPa at 298.15 K. Then, 2.0 mL of TMC solution in Isopar H ([TMC] = 1.5 g L$^{-1}$) was poured onto the surface of microfiltration membrane to conduct the interfacial polymerization. After the reaction time of 120 s, pure Isopar H was used to wash the unreacted monomer on the surface of as-formed polyamides. The as-prepared composite membranes were airdried under 333.5 K for 30 min to further crosslink and firmly cling onto the porous substrate.

### X-ray photoelectron spectroscopy test

Polyamide thin-film composite membrane sample was soaked in ultrapure water for 24 h, and then subjected to X-ray photoelectron spectroscopy (XPS) testing after vacuum drying. XPS test measured through Thermo Scientific K-Alpha+ at Al Kα line (1486.6 eV, 15 mA×15 kV), the vacuum level is 5×10$^{-9}$ mbar, X-ray beam spot is 300 μm. XPS data was analyzed through Avantage.

## Data availability

All data are available in the main text and Supplementary Information. All other data are available from the corresponding author upon request.

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

## Acknowledgements

We thank the financial support by the Natural Science Foundation of Zhejiang Province (Grant no. LD22E030001), the National Natural Science Foundation of China (Grant no. 22135006), and the National Key Research & Development Program of China (Grant no. 2021YFB3801503).

## Author contributions

Y.-R Xue and C. Liu conceived the idea and designed the research. Y.-R Xue, C. Liu, Z.-Y. Ma, and C.-Y. Zhu collected and processed data. H.-Q. Liang, H.-C. Yang, C. Zhang, and Z.-K. Xu provided constructive suggestions for results and discussion. Y.-R Xue, C. Liu and H.-Q. Liang

contributed to writing the manuscript. J. Wu and Z.-K. Xu revised the paper. All coauthors discussed the results.

## Competing interests

The authors declare no competing interests.

## Additional information

**Peer review information** : *Nature Communications* thanks the anonymous reviewers for their contribution to the peer review of this work. A peer review file is available.

