## [Peer Review File · Nature Communications]

Harmonic Amide Bond Density as a Game-Changer for Deciphering the Crosslinking Puzzle of PolyamideReviewers' Comments:

Reviewer #1:

Remarks to the Author:

B u i l d a clear relation between polymer structure and their the performance is very important for designing RO and NF membranes. The author's work is very important. But I dont think using the new parameter 'HABD'could give a good indication to estimate salt rejection and membrane flux. Although interfacial polymerization is a self-inhibition reaction, membrane thickness varies for different TFC membranes. using HABD without membrane thickness can not give a reasonable prediction for membrane flux. For salt rejection, HABD may give some information on charge density. However, donnon effect is very important for NF membrane not for RO membrane. But in this report, HABD is more suitable for depicting salt rejection to RO membrane than NF membranes. In addition, the author claimed that HABD is a better parameter than DNC. In fact, DNC was barely used to explain salt rejection and permeance properties. Nowadays, using PALs can directly explain the packing density of the IP layer. Surface charge can be also measured. The accuracy of HABD is highly depend on analysis of XPS data. In my view, XPS seems more important than HABD. More importantly, in Fig. 3c, even at the same value of HABD, rejection of RO membrane varied from 86 to 96%. This is hard to be accepted. I can see a good relation between permeability with FFV. If the HABD paremeter can have a similar accuracy as FFV, I w i l l s u g g e s t t o a c c e p t t h i s p a p e r .

Reviewer #2:

Remarks to the Author:

Ref: ESPR-D-22-1628

Title: Harmonic Amide Bond Density: A Game-Changer for Deciphering the Crosslinking Puzzle of Polyamide

Authors: Yu-Ren Xue, Chang Liu, Zhao-Yu Ma, Cheng-Ye Zhu, Jian Wu, Hong-Qing Liang, Hao-Cheng Yang, Chao Zhang, Zhi-Kang Xu

Corresponding author: Prof. Dr. Zhi-Kang Xu

Reviewed September 2023

In this work, Yu-Ren Xue, Chang Liu and co-worked deal with correlating the polymeric structure and selectivity properties of nanofiltration and reverse osmosis (desalination) membranes by statistical methods. This issue is very interesting to develop polymeric membranes but, at the same time, it is not such a simple question. From my experience, membrane performance (selectivity properties) depends not only of the polymer structure but also on the properties of the solid/liquid interface (surface tension or wettability, polymer surface charge and electric potentials or zeta potential, which depends also of the pH and salt type and concentration, fouling, ...), the membrane thickness and the whole membrane structure (most desalination membranes are composite). Despite this, it is necessary to simplify complex problems to understand them, and the correlation between some structural parameter of the polymer with the membrane performance is a common simplification in the literature.

The manuscript is well written and the correlation results are correctly analysed according to scientific methods. The supporting information is also sufficiently completes to correlate the new proposed index for describing the polyamide structure with the selectivity properties. However, in my opinion correlate the membrane performance solely to the polymer structure, and characterize the structure of the polymer only with XPS analysis, a technique that analyses only the externa surface of the film, is oversimplifying the problem. I think authors should discuss briefly these issues in the introduction, although this does not detract from the validity of the work.

Comparing the proposed Harmonic Amide Bond Density (HABD) method with the Degree of Network Crosslinking (DNC), the new method use three fit parameters while the DNC use only two. Obviously, the more parameters a model has, the better the fit of the experimental data will be. However, the

three parameters of the new model are well justified and more realistic than the DNC model to describe the structure of polyamides. The authors focus too much on highlighting the improvement of their description of the polymer structure compared to DNC, when in my opinion the main goal of this work is the way in which a parameter (let us call it HABD or DNC) related to the structure of the polymer is determined. And this point is not explained clearly enough either in the manuscript or in the supporting information.

Particularly, in lines 66-73 the authors argue that the DNC is miscalculated 1) because it does not take into account the polymer terminal groups and 2) because it is calculated by performing an incorrect analysis of the XPS spectra:

1) Regarding the first point, I agree the new model with three parameters better describes the polyamide structure, and on the surface (XPS is a technique with high surface sensitivity) there must be a high concentration of these terminal groups. Indeed the number of terminal groups near the surface can be estimated from XPS analysis. The authors must estimate them and comment on the results obtained.

2) The second point is the trickiest part of the work. Firstly, the methods used to calculate the HABD index from the XPS spectra are not explained in either the manuscript or the supplementary information (SI). In my opinion this is mandatory to publish the work, because it is very important for the reliability of all the work data.

Similarly to Table S2 (SI), where brief details of the membrane experiments are given, another Table should be included with the conditions under which the XPS spectra have been recorded and analysed (sample conditioning, vacuum level, area of analysis, X-ray type and energy, X-ray exposure time, ...) in the bibliography. Moreover, the method that the authors have followed to compute the HABD parameter with the data from the bibliography must be clearly explained. In addition to mentioning the number of peaks that are included in the analysis of the spectra (briefly described in Fig 2 and S2 and Tab. S1), more details must be included in the SI about how they have got the original spectra, the software that has been used, type of background, type of fitting curve (Gaussian, Lorentzian, ...) and any possible fitting constrictions (maximum or minimum peak width or height, peak position, ...), as well as giving some parameter that quantifies the fit quality. If the authors have obtained the atomic concentrations of the N1, N2, ... bands directly from the bibliography, some of this information should be summarized in this additional Table.

Secondly, the authors said that the DNC index is also calculated from XPS data processing, and references 28 and 29 are used to justify this. As far as I know, XPS is not the technique usually used to calculate the DNC, and particularly DNC is not calculated in any of these two references. What is said in ref. 28 is that the shape of the C1s and O1s peak is greatly influenced by absorbed water and other contaminants, and other references in the bibliography also have showed that X-ray can modify the surface of some polymers (for example ref. DOI10.1002/sia.1542). All these effects will be significant or not depending on the sample and XPS measurements conditions. That is why the experimental conditions under which XPS measurements were performed in the bibliography should be briefly summarized in an additional Table, as well as the subsequent analysis of the spectra.

In this technique, as in many others, it is necessary to fit the signal to quantify the abundance of an atom in a certain chemical environment. This fitting process has many more variables than the exact position in binding energy (BE) of the peaks. The signal used to correct the charge effect does not have a great influence on the concentration of certain type of atom if one works within each signal (N1s, O1s or C1s) with relative displacements between the different chemical environments of the atom. The binding energy (or BE displacements) of the different bands in a peak is what allows chemical states to be identified by XPS. This is the fundamentals of this analysis technique. What is important is not the exact BE values, but the displacements of some bands with respect to others within the complete peak. There is no other way to quantify the N1 and N2 number of atoms by XPS. Therefore, the discussion between lines 68 and 73 of the Introduction is absolutely incorrect.

The authors do not explain how they have worked. The only XPS spectrum they show is the N1s signal in Fig 2, where they observe two types of nitrogen atoms (N1 which I guess is the area highlighted in green, and N2 which apparently is the blue peak), and both bands are at the same binding energy. In this case, the band N1 may be due entirely to uncertainty in the background type that has been subtracted from the original spectrum. The authors have to give binding energies of the different

species of atoms that they want to quantify to obtain the HABD parameter. Tab. S1 only shows the binding energies of N2 and O2 bands. What displacement did they use to get respectively N1 and O1? What width have they assume for the N2 and O2 bands? Have they used Gaussian or Lorentzian bands? Without answering these questions, the N1/N2 and O1/O2 ratios necessary to calculate the HABD index are not credible.

All the above does not detract from the validity of the method they propose to estimate a parameter related to the structure of the polyamide. I only point out that the manuscript must show or give indications that the XPS measurements have been worked on and interpreted correctly, and that is to be done in this work.

Although I consider the model of the polyamide structure is good, the derivation of the HABD from the XPS analysis is not sufficiently explained, and there are indications in the manuscript that it has not been done correctly. For these reasons, I recommend that the manuscript be rejected until all of these issues are properly addressed.

Reviewer #3:

Remarks to the Author:

1. "Moreover, the calculation of DNC usually relies on X-ray photoelectron spectroscopy (XPS) data processing 28,29, which employs the charge shift of the full spectrum, theoretical binding energy, and peak fitting primarily based on the 1s peak of adventitious carbon 11,30. This approach, however, is not ideally compatible with carbon-based organic polymer materials with complex chemical environments 31,32, which further undermines the credibility of DNC as a reliable parameter for characterizing polyamide networks." While I sympathize with all these comments I think that Authors should be more concrete in defining the main problem which is the lack of reliable charge reference in XPS studies of insulating materials. The commonly used C 1s method based on the adventitious carbon has been shown to suffer from many issues which make it unreliable.). Adventitious carbon is in general an unknown compound, not an inherent part of the sample, does not make proper electrical contact to the analyzed sample, BE of the C 1s peak varies in a wide range. When it comes to samples which contain C (as is the case for materials considered in this paper) the situation is potentially better as one can directly refer to the 1s level of the specific chemical group. However, the main concern is – how much is the C 1s spectrum affected by the presence of adventitious carbon? The latter is present on all surfaces exposed to air which obviously adds to the confusion. Greater care during sample handling and minimized exposure time could be considered for more reliable results. I would suggest extending the discussion along these lines to better motivate the need for an alternative approach.

2. For the specific class of materials addressed by the proposed HABD method a very critical experimental variable, often completely neglected, is the X-ray exposure time. It is well known that such materials are prone to x-ray damage, which directly affects the bonding and, hence, the degree of crosslinking. This point is not mentioned at all in the paper.

3. The critical ingredient of the HABD method is the peak fitting of XPS spectra. Peak areas extracted from such models are used in the calculations of bond density parameter. Thus, the accuracy of the proposed approach relies on how accurate these peak models are. As Authors must be aware there is no unique way to decompose XPS spectra into component peaks (J. Vac. Sci. Technol. A 38, 061203 (2020); J. Vac. Sci. Technol. A 40, 063201 (2022)). Here, I'm missing the rigorous description of the peak fitting procedure. N 1s spectrum shown in Fig. 2B brings more questions than answers. Why are both peaks (N1 and N2) at the same binding energy if the chemical environment is not identical? How was the area ratio optimized?

4. "In this method, peak division of the N element is the only requirement, and the proportions of O and C atoms are calculated by assigning values based on the peak splitting results of the N element. For example, ..." If I understand it correctly Authors make the point here that the XPS peak

models need to be self-consistent. For example, COHN unit should give peaks in C 1s, O 1s, and N 1s spectra such that the quantified ratios would be close to 1:1:1. IS that correct? If yes, then the following paper can be useful in further development of this approach: Applied Surface Science 387 (2016) 294.

5. I find the description of the new method rather unclear. I think that showing an example of all XPS core level peak models (not only N 1s) would be appreciated by NC readers. This should be further used to illustrate how the fitting procedure should be conducted for reliable results (see point 3 above).

Response to the comments by Reviewer 1

Comment 1: Build a clear relation between polymer structure and their performance is very important for designing RO and NF membranes. The author's work is very important. But I don't think using the new parameter 'HABD' could give a good indication to estimate salt rejection and membrane flux.

Response: We are pleased to see that the reviewer found our strategy intriguing. We have carefully conducted further data analysis and result discussion to make our new parameter HABD rigorous and reasonable.

Comment 2: Although interfacial polymerization is a self-inhibition reaction, membrane thickness varies for different TFC membranes. using HABD without membrane thickness cannot give a reasonable prediction for membrane flux.

Response: We thank the reviewer for the insightful comments. The thickness of the polyamide selective layer is indeed a factor affecting water permeance. To inspect the possible influence of thickness on the prediction, we further correlated HABD with the water permeability of MPD-TMC polyamide RO membranes and PIP-TMC polyamide NF membranes. The water permeability (the product of water permeance and thickness of polyamide nanofilms) is the intrinsic performance parameter reflecting the water diffusion ability in polyamide nanofilms. When it comes PIP-TMC polyamide membranes, HABD can still form a significant correlation with water permeability (Fig. R1-1A, Spearman correlation coefficient: -0.289) compared to DNC (Fig. R1-1B) after taking the thickness into consideration. Regardless of whether the thickness of the polyamide selective layer is considered or not, the water transport capacity of polyamide selective layer will increase with the decrease of HABD. This result effectively supports current research that the thickness of polyamide is not a decisive factor for water permeance, while water permeance is mainly determined by the fractional free volume and pore distribution of the polyamide nanofilm (*Science*, 2023, 380: 242-244). Thick but loose polyamide nanofilms can achieve higher water permeance than thin and dense polyamide nanofilms (*Science*, 2021, 371: 72-75).

The data of RO membranes are insufficient to form a reliable correlation between HABD and water permeability. But the statistical results of water permeance and thickness (Fig. R1-1C) also indicate that the reduction in thickness does not result in an unrestricted increase in water permeance.

All these demonstrate that HABD is a comprehensive structure parameter, which can precisely depict the compactness of polyamide structures and form reasonable correlation with water permeance and water permeability. We believe that analyzing the relationship between HABD and water permeance can better reflect the uniqueness of HABD. Nonetheless, analyzing the relationship between HABD and water permeability coefficient is important and necessary.

Fig. R1-1 Correlation analysis between water permeability of NF membranes with DNC (A) and HABD (B) of PIP-TMC polyamides. Correlation analysis between thickness of MPD-TMC polyamide with water permeance of RO membranes (C).

Following the reviewer's suggestions, the above discussions were added in revised manuscript on page 16 to 17. Fig. R1-1A and B were added in the Supporting Information marked as Fig. S15 and Fig. S16. Tab. S7 and S8 were added in the Supporting Information, which summarized the thickness, DNC, HABD, water permeability of MPD-TMC RO membranes and PIP-TMC NF membranes. Fig. S13 and Fig. S14 were added in the Supporting Information, which analyze the correlation between DNC or HABD with water permeability in RO.

On page 16 to 17:

“Furthermore, it is widely accepted that water permeability, which is the product of water permeance and thickness of polyamide nanofilms, serves as an intrinsic performance parameter reflecting the water diffusion ability in polyamide nanofilms. We have counted the thicknesses of the polyamide nanofilms and calculated their water permeability in recent studies on RO and NF membranes (Tab. S7 and Tab. S8 in SI). Due to the insufficient data of reverse osmosis membranes to support reliable correlation analysis (Fig. S14 in SI), we primarily focused on examining the correlation between DNC or HABD and the water permeability of PIP-TMC polyamide in the nanofiltration process. Compared to DNC (Spearman correlation coefficient = 0.011, Fig. S15 in SI), HABD forms a more significant correlation with the water permeability of NF membranes. The Spearman correlation coefficient between HABD and water permeability is -0.298 (Fig. S16 in SI), which indicates a strong negative correlation between HABD and water permeability of NF membranes. Crosslinked polyamide network with higher HABD exhibits reduced fractional free volume and pore size resulting in lower water permeability, which is in line with theoretical expectations. It is worth noting that HABD can relatively reasonably describe the water permeance in polyamide membranes. Regardless of whether the thickness of the polyamide selective layer is considered, the water transport capacity of polyamide selective layer will increase with the decrease of HABD. This result strongly supports current research, which indicates that the thickness of polyamide is not a decisive factor for water permeance^{32,46}.”

Tab. S7. Summary of thickness, DNC and HABD of MPD-TMC polyamide nanofilms and RO water permeability of high-performance membranes.

DNC (%)	HABD (mmol g ⁻¹)	Water permeance (L m ⁻² h ⁻¹ bar ⁻¹)	Thickness (nm)	Thickness testing method and equipment	Water permeability (L nm m ⁻² h ⁻¹ bar ⁻¹)	Ref.
75.5	N. A.	5.60	148.7	SEM, Inspect F, FEI	832.72	18
53.4	N. A.	5.40	163.7		883.98	
47.5	N. A.	4.80	172.3		827.04	
70.3	N. A.	4.00	68	FESEM, JEOL	272.00	19
54.2	N. A.	6.20	55		341.00	
67.9	N. A.	5.70	147		837.90	
55.3	N. A.	3.80	136		516.80	
47.9	N. A.	4.90	197		965.30	
60.9	N. A.	4.50	177		796.50	
34.4	N. A.	1.92	299	FESEM, JSM-7600F	574.08	20
55.4	N. A.	1.86	346		643.56	
27.3	N. A.	1.95	292		569.40	
66.4	N. A.	1.10	31	FESEM, JEOL JSM7200F	34.10	21
71.8	N. A.	1.60	24		38.40	
75.8	N. A.	1.70	20		34.00	
10.7	N. A.	1.20	24		28.80	
31.4	N. A.	2.10	217.2	SEM, FEI Inspect F50	456.12	23
39.6	N. A.	3.00	214.1		642.30	
45.4	N. A.	4.50	216.7		975.15	
51.3	N. A.	6.30	212.1		1336.23	
91.7	9.25	0.68	141		95.88	24
77.2	8.71	1.69	83		140.27	

85.9	9.13	1.00	71	FESEM, Carl Zeiss, SUPRA 55VP	71.00	
83.8	N. A.	1.36	139	FESEM, JSF-7500F, JEOL	189.04	25
86.0	N. A.	2.12	122		258.64	
71.4	N. A.	2.58	154		397.32	
65.0	N. A.	2.76	179		494.04	
60.0	N. A.	2.41	194		467.54	
27.4	N. A.	2.90	6.7	AFM, Shimadzu	19.43	27
-20.5	N. A.	18.00	3.3	SPM-9700	59.40	
90.6	N. A.	1.73	20	SEM, HitachiS-4300	34.67	28
81.3	N. A.	1.87	20		37.33	
77.8	N. A.	2.07	70		144.67	
76.0	N. A.	2.13	150		320.00	
70.9	N. A.	2.80	250		700.00	
78.1	N. A.	3.07	186	TEM, Tecnai G2 F20, FEI	571.83	30
N. A.	N. A.	3.64	171		621.69	
N. A.	N. A.	2.94	147		431.81	
108.9	8.86	2.60	7.2	AFM, NX10, Park systems	18.72	34
46.8	N. A.	1.13	40	SEM, Nova NanoSEM	45.20	38
30.4	N. A.	2.32	13.6		31.55	
60.8	N. A.	3.15	12.8		40.32	
62.4	N. A.	3.31	11.5		38.07	
73.8	8.20	1.26	425	TEM, Hitachi	536.20	39
81.6	8.26	1.68	150	HT-7700	251.70	
38.7	N. A.	1.60	331	SEM, FEI	529.60	40
18.8	N. A.	2.50	319	Inspect F50	797.50	

28.9	N. A.	1.90	328		623.20	
52.2	N. A.	10.00	320		3200.00	
35.6	N. A.	3.80	330		1254.00	
147.8	N. A.	0.86	-		-	
164.4	N. A.	0.86	17	AFM, Veeco	14.59	41
95.1	N. A.	0.56	80	Nanoscope V	44.90	
80.2	N. A.	0.81	32		26.01	
49.0	N. A.	2.21	458		1012.92	
55.0	N. A.	2.32	375		870.97	
56.6	N. A.	2.58	250	SEM, Hitachi S-	645.16	49
63.9	N. A.	2.90	237	4800	688.06	
71.1	N. A.	3.13	158		495.10	
65.6	N. A.	2.71	152		411.87	
40.0	N. A.	1.94	175		338.71	
56.8	N. A.	2.71	225		609.68	
61.0	N. A.	3.10	250	SEM, Nova	774.19	52
64.2	N. A.	3.23	280	NanoSEM 430	903.23	
66.2	N. A.	3.23	300		967.74	
80.6	8.40	6.80	15	AFM, MultiMode, Veeco	102.00	
-3.8	6.23	2.97	8.6		25.56	
-32.5	5.27	2.57	8.6		22.06	
-13.3	6.26	2.42	10.1		24.39	
-21.8	5.93	2.12	12.8	SPM, Veeco	27.18	54
-77.5	4.41	1.89	10.4	MultiMode	19.69	
-51.6	5.14	1.77	12.6		22.29	
-93.2	3.97	1.82	11.8		21.50	

-19.4	5.94	1.73	12		20.81	
65.2	7.46	2.30	50	AFM, Veeco, MultiMode	115.00	55
79.5	7.58	2.25	120		270.00	
84.3	7.78	2.60	75		195.00	
95.5	7.71	3.05	45		137.25	

Tab. S8. Summary of thickness, DNC and HABD of PIP-TMC polyamide nanofilms and NF water permeability of high-performance membranes.

DNC (%)	HABD (mmol g ⁻¹)	Water permeance (L m ⁻² h ⁻¹ bar ⁻¹)	Thickness (nm)	Thickness testing method and equipment	Water permeability (L nm m ⁻² h ⁻¹ bar ⁻¹)	Ref.
47.3	N. A.	7.4	33.4	FESEM, NOVA NANOSEM 450	247.16	62
52.2	N. A.	13.9	28.8		400.32	
56.8	N. A.	16.8	23.5		394.80	
64.4	N. A.	25.2	18.2		458.64	
57.5	N. A.	21.4	20.1		430.14	
56.7	7.16	12.0	166.6	SEM, S-4800	1999.20	64
68.0	7.12	17.0	44.5		756.50	
69.3	7.54	20.0	27		540.00	
68.2	N. A.	0.6	220.8	FESEM, Hitachi S-4800	138.00	65
63.7	N. A.	0.9	140.2		122.68	
73.4	N. A.	1.1	104.9		118.01	
55.3	N. A.	1.6	86.6		140.73	
100.6	N. A.	2.3	67		150.75	
84.9	N. A.	4.9	568	SEM, S4800, Hitachi	2783.20	66
81.0	N. A.	N. A.	468		N. A.	
78.2	N. A.	9.6	235		2256.00	

47.5	N. A.	9.1	75	FE-SEM,	679.50	67
32.1	N. A.	21.0	64	SU8010, Hitachi	1341.44	
24.9	N. A.	5.3	N. A.	TEM, Tecnai F20	N. A.	69
17.6	N. A.	9.3	25		232.50	
17.9	N. A.	7.0	48		336.00	
24.7	N. A.	4.8	75		360.00	
9.3	N. A.	2.5	98		245.00	
44.3	N. A.	46.6	8.5	AFM,	396.10	71
33.2	N. A.	55.0	8.8	Agilent5500	484.00	
61.3	N. A.	4.8	79.3	FETEM, JEM 2100 F	380.64	73
67.8	N. A.	7.0	171.5		1200.50	
68.2	N. A.	10.0	180.2		1802.00	
68.2	N. A.	14.5	243.5		3530.75	
73.8	N. A.	20.6	269.9		5559.94	
15.8	N. A.	3.0	164	SEM, Hitachi SU8020,	492.00	74
-	N. A.	21.0	87		1827.00	
134.2						
-35.6	N. A.	9.3	94		874.20	
17.9	N. A.	7.0	129		903.00	
48.9	N. A.	5.0	143		715.00	
28.9	7.17	17.0	87	SEM,	1479.00	75
38.6	7.44	23.0	100	CSPM5500,	2300.00	
45.2	7.20	25.5	125	Being Nano-	3187.50	
43.0	7.65	26.0	63	Instruments,	1638.00	
50.8	8.20	30.5	42		1281.00	
63.1	N. A.	6.8	28.6	FESEM, NOVA	194.48	77
63.1	N. A.	14.2	26.6	NANOSEM 450	376.39	
62.1	N. A.	17.0	22.6		383.64	

58.6	N. A.	21.2	21.5		456.34	
48.7	N. A.	28.7	16.8		481.74	
54.9	9.03	6.8	106		720.80	
52.4	N. A.	13.0	82		1066.00	
60.9	9.48	17.0	45	AFM, SPM-9700	765.00	79
71.1	N. A.	16.5	42		693.00	
75.5	N. A.	15.0	38		570.00	
70.5	9.31	6.5	45	TEM, HT-7700,	292.50	80
67.1	9.33	8.5	20	Hitachi	170.00	
92.9	N. A.	50.2	80	TEM	4016.00	82
24.7	N. A.	27.0	13.4	AFM, Shimadzu	361.80	27
-18.3	N. A.	36.5	4.2	SPM-9700	153.30	
-8.3	5.73	N. A.	N. A.		N. A.	
-17.7	5.81	N. A.	N. A.	TEM, JEOL-	N. A.	84
-22.0	5.49	50.0	N. A.	1230	N. A.	
-25.2	5.29	55.0	15.5		852.50	
47.2	N. A.	N. A.	N. A.		N. A.	
-3.0	N. A.	14.5	71		1029.50	
2.2	N. A.	15.8	46		728.33	
23.9	N. A.	20.0	60	TEM, Hitachi	1200.00	85
24.0	N. A.	15.0	95	7650	1425.00	
33.2	N. A.	18.3	84		1540.00	
46.9	N. A.	23.3	76		1773.33	
38.1	N. A.	16.5	21		346.50	
37.2	N. A.	16.0	35.6	AFM,	569.60	86
28.1	N. A.	15.0	50.6	Agilent5500	759.00	
91.2	N. A.	11.0	75.6		831.60	

95.8	N. A.	24.0	93.6		2246.40	
44.4	N. A.	8.0	71.4	FESEM, SIGMA	571.20	87
63.6	N. A.	24.0	28.6	300	686.40	
53.3	7.60	8.4	78.5	SEM, Hitachi S4800	659.40	88
59.1	7.81	9.3	65.3		607.29	
64.3	7.99	11.7	52.6		615.42	
69.8	8.13	13.5	43.2		583.20	
78.6	8.40	16.8	35.7		599.76	
77.6	8.19	16.9	38.3		647.27	
75.2	N. A.	3.3	566	SEM, JSM - 7500F, JEOL	1861.84	89
71.4	N. A.	5.3	195		1026.32	
59.6	N. A.	6.1	135		817.11	
79.8	N. A.	7.5	102		765.00	
70.3	N. A.	13.4	41.2	SEM, JSM- 6700F, JEOL	552.08	90
67.1	N. A.	24.0	134.4		3225.60	
40.0	N. A.	27.6	132.2		3648.72	
34.1	N. A.	48.9	373.3		18254.37	
75.4	N. A.	40.0	15	TEM, Tecnai G2 F20 S-Twin	600.00	91
67.9	8.33	13.8	32	SPM, Veeco, MultiMode	441.60	95
65.4	8.04	14.9	30		447.30	
56.3	7.90	17.5	21		367.71	
37.6	7.10	20.9	15		313.20	
36.3	4.49	24.5	8.6		210.44	
93.4	8.94	20.0	19.2	SPM, Veeco, MultiMode	384.00	96
78.3	8.43	17.5	23.6		413.00	
88.1	9.05	20.0	18.7		374.00	

8.7	5.32	45.5	10.2	SEM, ZEISS	464.10	99
-----	------	------	------	------------	--------	----

Fig. S13. The correlation analysis between DNC and water permeability in RO.

Fig. S14. The correlation analysis between HABD and water permeability in RO.

Comment 3: For salt rejection, HABD may give some information on charge density. However, donnon effect is very important for NF membrane not for RO membrane. But in this report, HABD is more suitable for depicting salt rejection to RO membrane than NF membranes.

Response: HABD primarily reflects the content of amide bonds, representing the extent of reaction and the compactness of polyamide, while it does not include the Donnan effect of ionized terminal group. Therefore, HABD alone cannot establish a significant linear correlation with the rejection of NF membranes, but it is suitable for describing the water permeance of NF membranes.

To compensate for this deficiency, we introduced two analogous parameters, namely the harmonic carboxyl density (HCD) and the harmonic amino density (HAD), to accurately represent the surface charge of polyamides, by reflecting the contents of carboxyl and amino groups per mass of polyamide.

Considering the ionizing of carboxyl and amino groups in usual environment for nanofiltration testing (pH = 7), the total charge of the PIP-TMC polyamides can be considered to be related to the difference of dissociated carboxyl and amino groups ($[R-COO^-] - [R_2-NH_2^+]$), which is nearly proportional to our HCD:

$$[R-COO^-] - [R_2-NH_2^+] \propto 10^{4.88}[R-COOH] - 10^{2.83}[R_2-NH] = 10^{4.88}HCD - 10^{2.83}HAD \approx 10^{4.88}HCD \quad (R1-1)$$

The detailed calculation procedures were added in the revised Supporting Information. To illustrate the significance of Formula R1-1, we examined the correlation between HCD and zeta potential. Zeta potential refers to the electric potential at the shear plane within the surface double layer of polyamide nanofilms. Formula R1-1 indicates that the negative charge carried on polyamide nanofilms should theoretically form a positive correlation with the content of surface carboxyl groups (HCD). Consequently, with an increase in HCD, polyamide membranes are anticipated to demonstrate lower Zeta potentials.

We analyzed the correlation between the HCD values calculated by our method and the zeta potential as well as Na₂SO₄ rejection of recent NF membranes (Fig. R1-2). As expected, a strong negative correlation was observed between HCD and Zeta potential. PIP-TMC polyamide with higher HCD will exhibit stronger negative charge and increased Na₂SO₄ rejection. We recommend the comprehensive utilization of our proposed structural parameters,

specially HABD and HCD, to more accurately characterize the compactness and charge properties of polyamide networks.

Fig. R1-2 Correlation analysis between Zeta potential (A) and Na₂SO₄ rejection (B) of NF membranes with the HCD of PIP-TMC polyamides (pH = 7).

In response to the reviewer’s suggestion, the above discussions were added in revised manuscript on page 15. Fig. R1-2A and B were added in the Supporting Information marked as Fig. S17 and Fig. S18. The details regarding the calculation of HCD and HAD have been incorporated into the Supporting Information in Section 4. Tab. S9 was added in the Supporting Information, which summarized the Zeta potential, Na₂SO₄ rejection in NF and HCD in PIP-TMC polyamide.

On page 15:

“To quantify terminal groups and analyze their ionization behavior, we have introduced two analogous parameters, namely the harmonic carboxyl density (HCD) and the harmonic amino density (HAD). These parameters, reflecting the contents of carboxyl and amino groups per mass of polyamide, are essential in accurately representing the surface charge of polyamides (Section 4 in SI). It was demonstrated that PIP-TMC polyamide with higher HCD exhibits a stronger negative charge and higher Na₂SO₄ rejection (Tab. S9, Fig. S17 and Fig. S18 in SI). This finding strongly supports the notion that incorporating strong negative charges is effective in rejecting divalent anions when designing nanofiltration membranes. Nonetheless, the

rejection of nanofiltration membranes is contributed by two factors: the screening of pores brought by polyamide networks and the Donnan effect of surface charge. Therefore, it is challenging to establish reliable structure-activity relationships by isolating these two factors. We recommend the comprehensive use of our proposed structural parameters – HABD and HCD. Our method offers a possibility to comprehensively analyze the rejection of nanofiltration, by providing information on HABD for the compactness of crosslinked polyamide networks, as well as HCD and HAD for the charge density of the polyamide nanofilms. These structure parameters can be cross-referenced with other characterization methods, providing a valuable tool for analyzing polyamides.”

In Section 4, SI:

“We provided two parameters suitable for describing the surface charge of polyamides, named the harmonic carboxyl density (HCD) and the harmonic amino density (HAD) to reflect the contents of carboxyl and amino groups per mass of polyamide, respectively:

$$\text{HCD} = \frac{\alpha}{N_A m} = \frac{\alpha}{\alpha M_\alpha + \beta M_\beta + \gamma M_\gamma} \quad (\text{S4-1})$$

$$\text{HAD} = \frac{\gamma}{N_A m} = \frac{\gamma}{\alpha M_\alpha + \beta M_\beta + \gamma M_\gamma} \quad (\text{S4-2})$$

Through HCD and HAD, we can determine the density of terminal groups on the surface of polyamide and correlate it with the dissociation of polyamide¹⁰¹. It can even be mutually confirmed with the terminal group content of polyamide obtained by precision analysis devices such as titration, Quartz Crystal Microbalance or Rutherford Backscattering Spectrometry¹⁰²⁻¹⁰⁴.”

In Section 4.1, SI:

“For PIP-TMC polyamide, the ratio of α/β is numerically equivalent to the ratio of carboxyl group to amide bonds since one carboxyl group contains two O1 atom. That is, the ratio of $\frac{O1}{2}$ to O2. Similarly, the ratio of γ/β is numerically equivalent to the ratio of amino group to amide bonds, which is the ratio of (N1+N3) to N2. N1, N2, and N3 are the number of amino N, amide N, and protonated N of polyamide. Here, amino N are further classified as N1 and N3 for the convenience of peak fitting during XPS data process, which are detailed discussed afterwards.

Thus, HCD and HAD are equal to:

$$\text{HCD} = \frac{\frac{\alpha}{\beta}}{\frac{\alpha}{\beta}M_{\alpha}+M_{\beta}+\frac{\gamma}{\beta}M_{\gamma}} = \frac{\frac{\alpha}{\beta}}{\frac{01}{202}M_{\alpha}+M_{\beta}+\frac{N1+N3}{N2}M_{\gamma}} \quad (\text{S4-3})$$

$$\text{HAD} = \frac{\frac{\gamma}{\beta}}{\frac{\alpha}{\beta}M_{\alpha}+M_{\beta}+\frac{\gamma}{\beta}M_{\gamma}} = \frac{\frac{\gamma}{\beta}}{\frac{01}{202}M_{\alpha}+M_{\beta}+\frac{N1+N3}{N2}M_{\gamma}} \quad (\text{S4-4})$$

In Section 4.2, SI:

“In the actual separation process, it is necessary to consider the hydrolysis of carboxyl and amino groups. Take PIP-TMC polyamide as an example, here are two hydrolysis process in polyamide network:

$$\frac{[\text{H}_3^+\text{O}][\text{R-COO}^-]}{[\text{R-COOH}]} = 10^{-2.12} \quad (\text{S4-5})$$

$$\frac{[\text{H}_3^+\text{O}][\text{R}_2\text{-NH}]}{[\text{R}_2\text{-NH}_2^+]} = 10^{-9.83} \quad (\text{S4-6})$$

The dissociated carboxyl and amino groups will contribute opposite charge for polyamide, respectively. The total charge of the polyamide nanofilms can be considered to be related to the difference of dissociated carboxyl and amino groups:

$$[\text{R-COO}^-] - [\text{R}_2\text{-NH}_2^+] = \frac{10^{-2.12}[\text{R-COOH}]}{[\text{H}_3^+\text{O}]} - \frac{[\text{H}_3^+\text{O}][\text{R-NH}]}{10^{-9.83}} \quad (\text{S4-7})$$

7)

Considering that the usual nanofiltration process involves separating electrically neutral Na_2SO_4 solutions, pH equals to 7.

$$[\text{R-COO}^-] - [\text{R-NH}_2^+] \propto 10^{4.88}[\text{R-COOH}] - 10^{2.83}[\text{R}_2\text{-NH}] \quad (\text{S4-8})$$

where $[\text{R-COOH}]$ and $[\text{R}_2\text{-NH}]$ are the number of carboxyl and amino groups per unit volume of polyamide, proportional to our HCD and HAD:

$$[\text{R-COO}^-] - [\text{R}_2\text{-NH}_2^+] \propto 10^{4.88}\text{HCD} - 10^{2.83}\text{HAD} \approx 10^{4.88}\text{HCD} \quad (\text{S4-9})$$

Formula S4-9 indicates that the content of surface carboxyl groups (HCD) should theoretically form a positive correlation with the negative charge carried on polyamide nanofilms in $\text{pH} = 7$.”

Tab. S9. Summary of Zeta potential, Na₂SO₄ rejection in NF and HCD in PIP-TMC polyamide

N1+N3 (%)	N2 (%)	O1 (%)	HCD (mmol g ⁻¹)	Na ₂ SO ₄ rejection (%)	Zeta potential (mV)	Testing conditions of zeta potential testing conditions	Ref.
0.33	12.83	2.80	1.05	96.5	N. A.	N. A.	63
0.36	12.80	3.35	1.24	96.6	N. A.		
2.97	8.43	7.06	3.00	99.0	-24	CeO ₂ + Polysulfone, on Malvern Zetasizer Nano ZS,	64
3.54	8.96	7.25	2.88	98.0	-34		
4.09	9.11	4.73	1.96	95.8	-42		
0.16	12.90	3.83	1.40	97.2	N. A.	N. A.	72
0.07	11.16	8.55	3.13	99.2	N. A.		
0.30	11.65	7.34	2.66	97.8	N. A.		
0.16	12.90	4.03	1.47	98.4	N. A.		
2.25	8.22	7.64	3.33	95.0	-8	Alginate Hydrogel + Polyethersulfone, on SurPASS, Anton Paar GmbH, Graz	75
2.06	8.20	6.65	3.02	97.0	-35		
2.36	8.18	7.31	3.22	96.0	-35		
2.21	9.31	6.73	2.77	93.5	-30		
0.93	16.76	11.67	2.86	97.5	-35		
N. A.	N. A.	N. A.	3.79^a	92.0	-40	EVOH + PET, on Austrian Anton Paar surpass	78
N. A.	N. A.	N. A.	3.31^a	95.5	-52		
0.31	11.58	4.79	1.87	94.0	-24	Gelatin + PSf, on SurPASSTM3, Anton-Paar GmbH	79
N. A.	N. A.	N. A.	N. A.	98.5	-28		
0.40	11.88	3.02	1.20	98.5	-28		
N. A.	N. A.	N. A.	N. A.	98.5	-48		
N. A.	N. A.	N. A.	N. A.	98.5	-50		
0.30	5.39	1.54	1.33	91.3	N. A.	N. A.	80
0.38	9.48	2.81	1.38	96.4	N. A.		

2.40	7.10	13.14	5.31	N. A.	N. A.	N. A.	84
1.95	7.55	14.21	5.47	N. A.	N. A.		
2.11	6.78	14.29	5.78	94.0	N. A.		
2.30	6.71	15.21	5.99	91.0	N. A.		
N. A.	N. A.	N. A.	3.96^a	98.4	-38.83	Amino functionalized PSf, on SurPASSTM3	88
N. A.	N. A.	N. A.	3.65^a	99.2	-34.71		
N. A.	N. A.	N. A.	3.31^a	99.5	-27.84		
N. A.	N. A.	N. A.	3.06^a	99.5	-24.12		
N. A.	N. A.	N. A.	2.54^a	99.2	-15.41		
N. A.	N. A.	N. A.	2.78^a	99.0	-12.54		
2.05	10.74	5.12	1.98	99.7	-22	PES, on SurPASS Anton Paar, GmbH	95
2.33	10.12	5.58	2.22	99.6	-24		
2.20	10.09	6.39	2.50	99.5	-25		
2.58	9.08	8.71	3.40	99.4	-25		
5.86	5.22	11.83	5.08	97.8	-28		
2.22	11.69	2.85	1.09	99.6	N. A.	N. A.	96
2.28	10.54	4.28	1.71	98.8	N. A.		
1.82	11.77	2.94	1.13	98.0	N. A.		
3.21	7.02	9.63	4.21	N. A.	N. A.	N. A.	98
2.82	6.41	13.82	5.68	95.2	N. A.		
3.77	6.37	11.77	4.99	97.0	N. A.		
2.91	6.93	8.47	3.91	98.0	N. A.		
3.58	6.26	12.19	5.18	98.6	N. A.	N. A.	99

a: This work directly provided the proportion of carboxyl group, amide bond, and amino group.

Comment 4: In addition, the author claimed that HABD is a better parameter than DNC. In fact, DNC was barely used to explain salt rejection and permeance properties. Nowadays, using

PALs can directly explain the packing density of the IP layer. Surface charge can be also measured.

Response: We thank the reviewer for the insightful comments. We agree with the reviewer that positron annihilation lifetime spectrum (PALs) can conveniently determine the pore size and fractional free volume (FFV) of polyamide nanofilms. The content of surface groups can be obtained through various characterization methods, such as titration, quartz crystal microbalance (QCM) or Rutherford backscattering spectrometry (*Environ. Sci. Technol.*, 2021, 55: 6984-6994; *J. Membr. Sci.*, 2013, 429: 23-33; *Environ. Sci. Technol.*, 2008, 42: 5260-5266). However, there is a lack of direct correlation between the molecular structure with the porous structure of polyamide networks.

Our aim is to obtain a superior substitution structural parameter for DNC, which can be associated with advanced characterization methods like PALs. Since the HABD calculation encompasses all structural aspects of crosslinked polyamide, theoretically, deducing the polyamide FFV is feasible by calculating the van der Waals volume occupied by each structure cell. To validate the compatibility of HABD with PALS in determining FFV, we performed all-atom molecular modeling and structural optimization of crosslinked polyamide networks, accompanied by Monte Carlo simulations of FFV. This methodology has been demonstrated to accurately reflect the pore size distribution and FFV of crosslinked polymers (*Nat. Mater.*, 2016, 15, 760-767; *Angew. Chem. Int. Ed.*, 2021, 60, 14636-14643). Subsequently, we compared this approach with FFV calculations based on HABD through structure cell volume occupancy.

In our molecular models, the proportions of three structure cells α , β , and γ in MPD-TMC polyamide were fixed as 10%, 90% and 10%. Five crosslinked polyamide simulation boxes were created, varying the positions of α and β in the polymer chain (Fig. R1-3). Regardless of whether α is distributed in the middle or at the end of the polyamide chain, polyamide crosslinked network boxes with the identical structure cells consistently exhibit a FFV of approximately 34%. Simultaneously, we optimized the molecular structures of each structure cell through density functional theory (DFT) calculations and determined their van der Waals volumes in the condensed phase using electron density isosurfaces (Fig. R1-3I). By subtracting

the van der Waals volumes occupied by all structure cells from the total volume of the simulation boxes, we derived the FFV as predicted by HABD, which matched the FFV obtained from Monte Carlo sampling, consistently at 34%.

The simulation results confirm that polyamide networks containing the same HABD exhibit the same FFV. This finding provides a theoretical basis for the good correlation between HABD and performance of polyamide membranes. This favorable result is attributed to our segmentation method, where all atoms of polyamide are counted in the calculation of our structural parameters.

Fig. R1-3 Fractional free volume of five types of polyamides with different structure cell distributions. Almost identical distribution positions of vacancy (The green to yellow dots indicate the size of the vacancy from small to large) in different polyamide crosslinked networks with 100%(A), 75%(B), 50%(C), 25%(D), 0%(E) of structure cell α located in the middle of polyamide chain respectively. Structural formula of parts of MPD-TMC polyamide when

structure cell α located in the middle (F) or in the middle (H) of polyamide chain respectively. Polyamides with different α location distribution own similar fractional free volume (G). Van der Waals volumes of structure cells α , β , and γ in condensed phase are calculated from corresponding molecules using electron density isosurfaces (I).

In addition, HCD and HAD can be correlated to the results of terminal group analysis as indicated in our response to Comment 3. Thus, our method has the potential to serve as a bridge to connect various characterization methods such as XPS, PALs.

In response to the reviewer's suggestion, Fig. R1-3 was added in the Supporting Information marked as Fig. S2, and the above discussions were added in revised manuscript on page 7. The simulation details were added in the Supporting Information in Section 1.4.

On page 7:

“In polyamide crosslinked networks, the vacancy (free volume), volume and mass contributed by a structure cell are uniform across all positions. Since the HABD calculation encompasses all structural aspects of crosslinked polyamide, theoretically, deducing the polyamide fractional free volume (FFV) is feasible by calculating the van der Waals volume occupied by each structure cell. As an illustration, we set up five simulation boxes of MPD-TMC polyamides based on the different positions of α and β within the polymer chains, either as chain terminators or situated within the chain. The proportions of three structure cells α , β , and γ were fixed as 10%, 90% and 10%. (Fig. S2 in SI). Regardless of whether α is distributed in the middle or at the end of the polyamide chain, crosslinked polyamide network boxes with identical structure cells consistently have a FFV of about 34%^{32,33}. Simultaneously, by subtracting the van der Waals volumes occupied by all structure cells from the total volume of the simulation boxes, we derived the FFV as predicted by HABD, which matched the FFV obtained from Monte Carlo sampling, consistently at 34%.”

In Section 1.4, SI:

“In the structural simulation of polyamides, simulated crosslinking was employed to construct an all-atom molecular model of polyamide. Deprotonated MPD and TMC with removed

chlorine atoms served as the main building blocks. To introduce hydrolysis effects during modeling, some acyl chloride groups in TMC were substituted with carboxyl groups. The complete monomers were constructed with Gaussview and optimized via Gaussian 16 software package ¹ at the B3LYP-D3(BJ) ^{2,3} level of theory with a def2-TZVP basis set ⁴. These optimized monomers were then modified into building blocks for further use.

The simulation box, comprising an orthogonal periodic cell with dimensions of $70 \times 70 \times 70$ Å³, was filled with different building blocks using Packmol ⁵ at a density of 0.4 g cm⁻³. The crosslinking process was simulated using the Polymatic program package ⁶ under the polymer consistent force field (PCFF) ⁷, using RESP2 atomic charges calculated by Multiwfn ⁸. This package methodically identified crosslinking chemical bonds by assessing the distance and molecular angles between atoms ready for bonding. Key bonding atoms, specifically amino nitrogen and carbonyl carbon, were provisionally assigned virtual charges of -0.5 and +0.5, respectively. Upon satisfying the bonding criteria, these atoms were connected to form an amide bond, followed by the removal of the virtual charges, and subsequent energy minimization and MD relaxation. After the simulated crosslinking process concluded, amino nitrogen atoms that did not participate in crosslinking were saturated with hydrogen atoms, and uninvolved carbonyl carbon atoms were saturated with hydroxyl groups. The finalized structure then underwent a 21-step simulated annealing procedure using the LAMMPS program, facilitating structural relaxation under diverse temperature and pressure conditions to yield a fully atomistic model of the crosslinked polyamide. The supplementary settings for the simulated crosslinking were adapted from the methodologies of Abbott et al. and our preceding research ^{6,9}.

The fractional free volume within the polyamide networks, considering different hydrolysis sites, were computed using the ZEO++ program with a probe radius of 0.1 Å, based on Voronoi decomposition. The molecular volumes of the structure cells were calculated employing the Multiwfn program, which utilized the Marching Tetrahedron algorithm with an isosurface of 0.002 a.u. ^{8,10}. All visualizations of the structures were generated using VMD 1.9.4 ¹¹.”

Comment 5: The accuracy of HABD is highly depend on analysis of XPS data. In my view, XPS seems more important than HABD.

Response: We appreciate the reviewer's insightful comment. The accuracy of XPS process significantly affects the calculated HABD. In response to the reviewer's suggestion, we optimized the process of obtaining HABD through XPS testing results.

We emphasize stricter constraints, including cross-peak self-consistency requirement (*Appl. Surf. Sci.*, 2016, 387: 294), more accurate recommendations for peak positions (*J. Colloid Interface Sci.*, 2002, 247: 149-158; *Science*, 2015, 348, 1347-1351; *Environ. Sci. Technol.*, 2012, 46: 852-859), peak shape, and full width at half maxima (FWHM), to limit the peak fitting process of XPS data (*J. Vac. Sci. Technol. A*, 2020, 38: 061203; *J. Vac. Sci. Technol. A*, 2022, 40: 063201).

With the hope of improving credibility and accuracy obtained HABD through XPS data, we have added these contents in the revised manuscript on page 9 to 11. Tab. S1 was added in the Supporting Information, showing the recommended peak positions for XPS peak fitting of MPD-TMC polyamide. Fig. S3 was added in the Supporting Information, which shown the flowchart of XPS data analysis process with MPD-TMC polyamide as an example. Fig. S4 was added in the Supporting Information, which shown recommended atom proportion result of N, O, and C of example MPD-TMC polyamide.

After optimizing the XPS processing method, the obtained HABD data in Fig. 3 and Fig. 4 has changed accordingly. But the superiority of HABD in correlation performance remains unchanged.

On page 9 to 11:

“To accurately determine the values of HABD, we propose a matching XPS data processing method with MPD-TMC polyamide as an example (Fig. 2 and Fig. S3 in SI). Firstly, we categorize the atoms of polyamide based on the structure cell segmentation (Fig. 2A and Tab. S1 in SI). Nitrogen (N) atoms are divided into amino N atoms (N1, in amino group), amide N atoms (N2, in amide bond), and protonated amino N atoms (N3, in amino group), while Oxygen (O) atoms are categorized into non-amide O atoms (O1) and amide O atoms (O2, in amide

bond). Carbon (C) atoms are also divided into carboxyl C atoms (C1, in carboxyl group), amide C atoms (C2, in amide bond), N-related C atoms (C3, connected to the N atom, but not in amide bond), and O-related C atoms (C4, connected to O atom, but not in carboxyl group) and common C atoms (C5).

Subsequently, we utilize the N element (specifically, the binding energy of N2) for the charge shift of the full-spectrum. The N peak in polyamide is mainly contributed by the N atom of the amide bond, which minimally affected and interfered by external impurities, making its N2 peak the most easily identifiable. This step is beneficial for subsequent data analysis and peak fitting process, although it does not significantly improve the accuracy of the analysis.

Then, the N peak is divided into three peaks based on the given relative peak positions of N1, N2 and N3 as outlined in the relevant research or handbooks (Fig. 2B and Tab. S1 in SI)^{11,34,35}. Peak fitting should be carried out under Gaussian–Lorentzian product pseudo-Voigt peak shape, with a Gaussian component comprising 60% ~ 80% of the peak, as Gaussian components are commonly observed in XPS peaks of polymers^{36,37}. Throughout this process, attention should be paid to constraining the peak positions of the three N segments within the recommended range while ensuring that their full width at half maxima (FWHM) are similar.

The peak fitting of C and O is carried out with strict adherence to the cross-peak self-consistent relationship,³⁸ in addition to considering curve shape, peak position, and FWHM. The proportion of O2 and C2 can be determined based on the cross-peak self-consistent relationship between the elements within the amide bond or amino in polyamide. In a single amide bond, there should be one C atom, one O atom, and one N atom, resulting in equal quantities of C2, O2, and N2. In MPD-TMC, each N atom is connected to a C atom in a benzene ring, implying that the number of C3 should be equal to the total number of N atoms (Fig. 2B):

$$C2 = O2 = N2 \quad (3)$$

$$C3 = N1 + N2 + N3 \quad (4)$$

The content of C1, C4, C5, and O1, cannot be directly calculated through the peak fitting results of N. However, self-consistent relationships also impose constraints on their cross-peak self-consistency.

In the absence of impurities, both C and O originate from polyamide. A carboxyl group consist of two O1 atoms, one C1 atom and is connected to a C4 atom. Similarly, an amide bond is also connected to a C4 atom, thus:

$$C1 = \frac{O1}{2} \quad (5)$$

$$C4 = \frac{O1}{2} + N2 \quad (6)$$

According to the division and statistics of our structure cells, we can also obtain statistical relationships:

$$C5 = \frac{O1}{2} + 3N2 + 2(N1 + N3) \quad (7)$$

The remaining atoms that cannot be separated should be considered as impurities, rather than forcibly attributed to specific atomic type such as C1, C4, C5 or O1. For example, if the total number of C after peak fitting exceeds the sum of C1-C5 under these constraint conditions, it indicates the presence of impurity C in the XPS test. Similarly, if the total number of C is insufficient to satisfy the Formula 5~7 after dividing the peak of O, it indicates that some impurities in O have been separated and attributed to O1.

Fig. 2B shows the peak fitting results of our self-prepared sample. We use N as the standard for correction. N peak is fully attributed to different N atoms within the polyamide. When the peak conditions and cross-peak self-consistency constraints are met, the C peak does not exclusively correspond to O1~O2 and C1~C5 atoms of polyamide. The remaining portions are treated as impurities with unspecific peak positions. The quantities of various atoms (Fig. S4) must adhere to the self-consistent requirements in Formula 3~7, which is a crucial aspect of our method. Once the values of N1, N2, N3, O1, and O2 are obtained, HABD can be calculated by the formula in Fig. 2C (the proof process is shown in Formula S2-1~S2-4 in SI). A similar analysis process for PIP-TMC polyamide is also provide (Section 2.4 in SI).”

Fig. 2. Graphical representation of analyzing the polyamide structure and calculating HABD using MPD-TMC polyamide as an example. The polyamide example were synthesized according to the following conditions: [MPD] = 20.0 g L⁻¹, [TMC] = 1.5 g L⁻¹, reaction time = 120 s. XPS test measured through Thermo Scientific K-Alpha+ at Al K α line (1486.6 eV, 15 mA×15 kV), the vacuum level is 5×10⁻⁹ mbar, X-ray beam spot is 300 μ m, total X-ray irradiation time is 154.1 s (68 s for full spectrum, 28.7 s for C, 30.2 s for O, and 27.2 s for N). Peak fitting was based on the recommended peak position in Tab. S1^{11,34,35}.

Fig. S3. Flowchart of XPS data analysis process of MPD-TMC polyamide.

Tab. S1. Classification and the number of various atoms of α , β , and γ structure cells in MPD-TMC polyamide and recommended peak positions for XPS peak fitting of MPD-TMC polyamide¹²⁻¹⁴.

Type	Group	MPD-TMC polyamide			
		XPS peak position	α	β	γ
C1	-C-COOH	288.8 ± 0.1 eV	1	0	0
C2	-C-CONH-	288.3 ± 0.2 eV	0	1	0
C3	-C-NH ₂ , -CONH-C-,	286.3 ± 0.2 eV	0	1	1
C4	-C-COOH, -C-CONH-	285.7 ± 0.2 eV	1	1	0
C5	-C-H, -C-C, -C=C	284.8 ± 0.2 eV	1	3	2
O1	-COOH	533.1 ± 0.3 eV	2	0	0

O2	-CONH-	531.6 ± 0.4 eV	0	1	0
N1	-NH ₂	398.5 ± 0.2 eV	0	0	1
N3	-NH ₃ ⁺	401.7 ± 0.2 eV	0	0	1
N2	-CONH-	400.0 eV	0	1	0

Fig. S4. Recommended atom proportion result of N, O, and C of example MPD-TMC polyamide. The polyamide example was synthesized according to the following conditions: [MPD] = 20.0 g L⁻¹, [TMC] = 1.5 g L⁻¹, reaction time = 120 s. XPS test measured through Thermo Scientific K-Alpha+ at Al K α line (1486.6 eV, 15 mA \times 15 kV), the vacuum level is 5 \times 10⁻⁹ mbar, X-ray beam spot is 300 μ m, total X-ray irradiation time is 154.1 s (68 s for full spectrum, 28.7 s for C, 30.2 s for O, and 27.2 s for N).

Fig. 3. The correlation analysis between the performance of reverse osmosis membranes with the DNC or HABD of MPD-TMC polyamides. The DNC parameter does not show a reliable correlation with the NaCl rejection (A) and water permeance (B) of reverse osmosis polyamide membranes. In contrast, HABD demonstrates a reliable correlation with the NaCl rejection (C) and water permeance (D) of those membranes (The detailed statistical data in this figure is shown in Tab. S3 and S4 in SI).

Fig. 4. The correlation analysis between the performance of nanofiltration membranes with the DNC or HABD of PIP-TMC polyamides. DNC cannot form a reliable correlation with the Na₂SO₄ rejection (A) and water permeance (B) of nanofiltration polyamide membranes. In contrast, HABD demonstrates a reliable correlation with the Na₂SO₄ rejection (C) and water permeance (D) of those membranes. (The detailed statistical data in this figure is shown in Tab. S5 and S6 in SI).

Comment 6: More importantly, in Fig. 3c, even at the same value of HABD, rejection of RO membrane varied from 86 to 96%. This is hard to be accepted. I can see a good relation between permeability with FFV. If the HABD parameter can have a similar accuracy as FFV, I will suggest to accept this paper.

Response: We thank the reviewer for the insightful comments. The fractional free volume (FFV), also known as fractional vacancy volume is a key parameter that characterizes the pore structure of crosslinked polyamide nanofilms. It establishes a reliable phenomenological

connection with the performance of polyamide membranes, such as water permeance (*Science*, 2021, 371: 72-75). According to our simulation results (Fig. R1-3), polyamides structure with identical HABD exhibit the same FFV, establishing an internal basis for the good correlation between the separation performance with HABD.

However, in particle analysis, the performance of RO or NF does not exhibit a one-to-one correspondence with HABD.

On the one hand, the HABD data calculated in our work are all from existing researches, and their measurement process did not follow the optimized XPS analysis method recommended in our work. We firmly believe that by employing our optimized XPS analysis method, we can enhance the accuracy of the XPS data consequently improving the precision of the calculated HABD.

On the other hand, the water permeance of polyamide membranes is influenced by various factors beyond the structural properties of polyamide selective layer. In addition to considering parameters such as the polyamide surface charge, pore size, and fractional free volume concerned by our HABD or PALs, the hydrophilicity of polyamide layer and substrate, along with the pore structure properties of substrate play significant roles in determining the performance of polyamide membranes. We exclude these factors when analyzing polyamide selective layers using HABD or FFV. Furthermore, variations in salt concentration, testing temperature, cross flow velocity, and equipment used for measuring water separation performance contribute to differences in water permeance and salt rejection among polyamide membranes with identical HABD values. Consequently, a direct one-to-one correspondence between HABD and the performance of RO membranes cannot be established.

Notwithstanding, we maintain our belief that our method is still progressive. HABD exhibit a strong correlation with performance, surpassing what DNC can achieve. Our optimized XPS method allows for the accurate determination of the HABD, HCD and HAD values, all of which can be correlated with other characterization methods.

In response to the reviewer's suggestion, we have added these contents briefly in the revised manuscript on page 17.

On page 17:

“The differences in separation testing conditions, as well as inconsistencies in XPS testing condition are the reasons why it is difficult to form a direct linear relationship between HABD and performance.”

Response to the comments by Reviewer 2

Comment 1: In this work, Yu-Ren Xue, Chang Liu and co-workers deal with correlating the polymeric structure and selectivity properties of nanofiltration and reverse osmosis (desalination) membranes by statistical methods. This issue is very interesting to develop polymeric membranes but, at the same time, it is not such a simple question. From my experience, membrane performance (selectivity properties) depends not only of the polymer structure but also on the properties of the solid/liquid interface (surface tension or wettability, polymer surface charge and electric potentials or zeta potential, which depends also of the pH and salt type and concentration, fouling, ...), the membrane thickness and the whole membrane structure (most desalination membranes are composite). Despite this, it is necessary to simplify complex problems to understand them, and the correlation between some structural parameter of the polymer with the membrane performance is a common simplification in the literature.

The manuscript is well written and the correlation results are correctly analysed according to scientific methods. The supporting information is also sufficiently complete to correlate the new proposed index for describing the polyamide structure with the selectivity properties. However, in my opinion correlate the membrane performance solely to the polymer structure, and characterize the structure of the polymer only with XPS analysis, a technique that analyses only the external surface of the film, is oversimplifying the problem. I think authors should discuss briefly these issues in the introduction, although this does not detract from the validity of the work.

Response: We are pleased to see that the reviewer found our strategy intriguing and thank the reviewer for the recognition and valuable suggestions. It is a very complex and cumbersome issue about the relationship between the performance of polyamide thin film composite membranes with the structure of polyamide selective layer. Although we only focus on polyamide nanofilms as the core selective layer, we agree with the reviewer that many factors including substrate properties and testing conditions indeed affect the performance of polyamide thin film composite membranes. We described these contents in Introduction in the revised manuscript.

On page 3:

“Desalination-oriented polyamide membranes for nanofiltration and reverse osmosis are primarily developed by reacting trimesoyl chloride (TMC) with piperazine (PIP) and m-phenylenediamine (MPD) on microfiltration or ultrafiltration substrates, respectively ^{5,6}. While the properties of the substrate and the testing conditions can influence the performance of the polyamide thin-film composite membranes, it is the permselective polyamide layer that plays a central role in achieving water/salt separation. The permeation of water or salt through a polyamide thin-film composite membrane is a complex problem that necessitates simplification. Researchers consistently direct their attention towards polyamide nanofilm and have been actively exploring various strategies, including the modification of monomers, solvents, and polymerization conditions, to regulate and customize the crosslinked networks of polyamides to enhance their separation efficiencies ⁷⁻⁹. Consequently, it is indispensable to analyze the crosslinked networks of permselective polyamide nanofilms and establish their correlation with desalination performance”

On page 4:

“Nonetheless, XPS remains an irreplaceable and high-precision method to analyze the element composition and chemical environment of the 10 nm region on the surface of polyamide selective layer.”

Comment 2: Comparing the proposed Harmonic Amide Bond Density (HABD) method with the Degree of Network Crosslinking (DNC), the new method use three fit parameters while the DNC use only two. Obviously, the more parameters a model has, the better the fit of the experimental data will be. However, the three parameters of the new model are well justified and more realistic than the DNC model to describe the structure of polyamides. The authors focus too much on highlighting the improvement of their description of the polymer structure compared to DNC, when in my opinion the main goal of this work is the way in which a parameter (let us call it HABD or DNC) related to the structure of the polymer is determined.

And this point is not explained clearly enough either in the manuscript or in the supporting information.

Response: We thank the reviewer for the insightful comments. For thin film composite membranes with polyamide as the selective layer, water and salt will permeate through the pores and cavities between polyamide chains (*Sci. Adv.*, 2023, 9: eadf8488). The performance of polyamide membranes is significantly related to their pore size and fractional free volume (FFV). Recent researches further revealed the FFV of the polyamide nanofilm significantly affect the water permeance (*Science*, 2021, 371: 72-75; *Science*, 2023, 380: 242-244). Therefore, structural parameters should be correlated with these pore structure of polyamide network, and in this regard, our HABD is far superior to DNC.

DNC is obtained by simplifying the structure of polyamide. There is no quantitative relationship between DNC with the pore size, fractional free volume and surface charge of polyamide nanofilms. Researchers usually can only qualitatively assume that the higher the DNC of polyamide, the smaller its pore size and porosity, resulting in higher rejection and lower water permeance. As the accuracy of DNC calculations decreases, this function is gradually being abandoned, as we mentioned the Introduction Section.

Our aim is to obtain a superior substitution structural parameter for DNC, which can be associated with advanced characterization methods like PALs. Since the HABD calculation encompasses all structural aspects of crosslinked polyamide, theoretically, deducing the polyamide FFV is feasible by calculating the van der Waals volume occupied by each structure cell. To validate the compatibility of HABD with PALs in determining FFV, we performed all-atom molecular modeling and structural optimization of crosslinked polyamide networks, accompanied by Monte Carlo simulations of FFV. This methodology has been demonstrated to accurately reflect the pore size distribution and FFV of crosslinked polymers (*Nat. Mater.*, 2016, 15, 760-767; *Angew. Chem. Int. Ed.*, 2021, 60, 14636-14643). Subsequently, we compared this approach with FFV calculations based on HABD through structure cell volume occupancy.

In our molecular models, the proportions of three structure cells α , β , and γ in MPD-TMC polyamide were fixed as 10%, 90% and 10%. Five crosslinked polyamide simulation boxes

were created, varying the positions of α and β in the polymer chain (Fig. R2-1). Regardless of whether α is distributed in the middle or at the end of the polyamide chain, polyamide crosslinked network boxes with the identical structure cells consistently exhibit a FFV of approximately 34%. Simultaneously, we optimized the molecular structures of each structure cell through density functional theory (DFT) calculations and determined their van der Waals volumes in the condensed phase using electron density isosurfaces (Fig. R2-1I). By subtracting the van der Waals volumes occupied by all structure cells from the total volume of the simulation boxes, we derived the FFV as predicted by HABD, which matched the FFV obtained from Monte Carlo sampling, consistently at 34%.

The simulation results confirm that polyamide networks containing the same HABD exhibit the same FFV. This finding provides a theoretical basis for the good correlation between HABD and performance of polyamide membranes. This favorable result is attributed to our segmentation method, where all atoms of polyamide are counted in the calculation of our structural parameters.

Fig. R2-1 Fractional free volume of five types of polyamides with different structure cell distributions. Almost identical distribution positions of vacancy (The green to yellow dots indicate the size of the vacancy from small to large) in different polyamide crosslinked networks with 100%(A), 75%(B), 50%(C), 25%(D), 0%(E) of structure cell α located in the middle of polyamide chain respectively. Structural formula of parts of MPD-TMC polyamide when structure cell α located in the middle (F) or in the middle (H) of polyamide chain respectively. Polyamides with different α location distribution own similar fractional free volume (G). Van der Waals volumes of structure cells α , β , and γ in condensed phase are calculated from corresponding molecules using electron density isosurfaces (I).

In response to the reviewer's suggestion, Fig. R2-1 was added in the Supporting Information marked as Fig. S2, and the above discussions were added in revised manuscript on page 7. The simulation details were added in the Supporting Information in Section 1.4.

On page 7:

“In polyamide crosslinked networks, the vacancy (free volume), volume and mass contributed by a structure cell are uniform across all positions. Since the HABD calculation encompasses all structural aspects of crosslinked polyamide, theoretically, deducing the polyamide fractional free volume (FFV) is feasible by calculating the van der Waals volume occupied by each structure cell. As an illustration, we set up five simulation boxes of MPD-TMC polyamides based on the different positions of α and β within the polymer chains, either as chain terminators or situated within the chain. The proportions of three structure cells α , β , and γ were fixed as 10%, 90% and 10%. (Fig. S2 in SI). Regardless of whether α is distributed in the middle or at the end of the polyamide chain, crosslinked polyamide network boxes with identical structure cells consistently have a FFV of about 34%^{32,33}. Simultaneously, by subtracting the van der Waals volumes occupied by all structure cells from the total volume of the simulation boxes, we derived the FFV as predicted by HABD, which matched the FFV obtained from Monte Carlo sampling, consistently at 34%.”

In Section 1.4, SI:

“In the structural simulation of polyamides, simulated crosslinking was employed to construct an all-atom molecular model of polyamide. Deprotonated MPD and TMC with removed chlorine atoms served as the main building blocks. To introduce hydrolysis effects during modeling, some acyl chloride groups in TMC were substituted with carboxyl groups. The complete monomers were constructed with Gaussview and optimized via Gaussian 16 software package¹ at the B3LYP-D3(BJ)^{2,3} level of theory with a def2-TZVP basis set⁴. These optimized monomers were then modified into building blocks for further use.

The simulation box, comprising an orthogonal periodic cell with dimensions of $70 \times 70 \times 70 \text{ \AA}^3$, was filled with different building blocks using Packmol⁵ at a density of 0.4 g cm^{-3} . The crosslinking process was simulated using the Polymatic program package⁶ under the polymer consistent force field (PCFF)⁷, using RESP2 atomic charges calculated by Multiwfn⁸. This package methodically identified crosslinking chemical bonds by assessing the distance and molecular angles between atoms ready for bonding. Key bonding atoms, specifically amino

nitrogen and carbonyl carbon, were provisionally assigned virtual charges of -0.5 and +0.5, respectively. Upon satisfying the bonding criteria, these atoms were connected to form an amide bond, followed by the removal of the virtual charges, and subsequent energy minimization and MD relaxation. After the simulated crosslinking process concluded, amino nitrogen atoms that did not participate in crosslinking were saturated with hydrogen atoms, and uninvolved carbonyl carbon atoms were saturated with hydroxyl groups. The finalized structure then underwent a 21-step simulated annealing procedure using the LAMMPS program, facilitating structural relaxation under diverse temperature and pressure conditions to yield a fully atomistic model of the crosslinked polyamide. The supplementary settings for the simulated crosslinking were adapted from the methodologies of Abbott et al. and our preceding research ^{6,9}.

The fractional free volume within the polyamide networks, considering different hydrolysis sites, were computed using the ZEO++ program with a probe radius of 0.1 Å, based on Voronoi decomposition. The molecular volumes of the structure cells were calculated employing the Multiwfn program, which utilized the Marching Tetrahedron algorithm with an isosurface of 0.002 a.u. ^{8,10}. All visualizations of the structures were generated using VMD 1.9.4 ¹¹.”

Comment 3: Particularly, in lines 66-73 the authors argue that the DNC is miscalculated 1) because it does not take into account the polymer terminal groups and 2) because it is calculated by performing an incorrect analysis of the XPS spectra:

1) Regarding the first point, I agree the new model with three parameters better describes the polyamide structure, and on the surface (XPS is a technique with high surface sensitivity) there must be a high concentration of these terminal groups. Indeed the number of terminal groups near the surface can be estimated from XPS analysis. The authors must estimate them and comment on the results obtained.

Response: We appreciate the reviewer’s insightful comment. HABD mainly reflects the content of amide bonds, reflecting the extent of reaction and the compactness of polyamide. HABD do not included the Donnan effect of the terminal group. Therefore, HABD itself cannot form a significant linear correlation with the rejection of NF membranes, and is only suitable for

describing the water permeance of NF membranes. Following the reviewer's suggestion, we provide two analogous parameters suitable for describing the surface charge of polyamides, named the harmonic carboxyl density (HCD) and the harmonic amino density (HAD) to reflect the contents of carboxyl and amino groups per mass of polyamide.

$$\text{HCD} = \frac{\frac{\alpha}{N_A}}{m} = \frac{\alpha}{\alpha M_{\alpha} + \beta M_{\beta} + \gamma M_{\gamma}} = \frac{\frac{\alpha}{\beta}}{\frac{\alpha}{\beta} M_{\alpha} + M_{\beta} + \frac{\gamma}{\beta} M_{\gamma}} \quad (\text{R2-1})$$

$$\text{HAD} = \frac{\frac{\gamma}{N_A}}{m} = \frac{\gamma}{\alpha M_{\alpha} + \beta M_{\beta} + \gamma M_{\gamma}} = \frac{\frac{\gamma}{\beta}}{\frac{\alpha}{\beta} M_{\alpha} + M_{\beta} + \frac{\gamma}{\beta} M_{\gamma}} \quad (\text{R2-2})$$

2)

For PIP-TMC polyamide, the ratio of α/β is numerically equivalent to the ratio of carboxyl group to amide bonds since one carboxyl group contains two O1 atom. That is, the ratio of $\frac{O1}{2}$ to O2. Similarly, the ratio of γ/β is numerically equivalent to the ratio of amino group to amide bonds, which is the ratio of (N1+N3) to N2. N1, N2, and N3 are the number of amino N, amide N, and protonated N of polyamide. Here, amino N are further classified as N1 and N3 for the convenience of peak fitting during XPS data process, which are detailed discussed afterwards. HCD and HAD are equal to:

$$\text{HCD} = \frac{\frac{\alpha}{\beta}}{\frac{\alpha}{\beta} M_{\alpha} + M_{\beta} + \frac{\gamma}{\beta} M_{\gamma}} = \frac{\frac{\alpha}{\beta}}{\frac{O1}{2O2} M_{\alpha} + M_{\beta} + \frac{N1+N3}{N2} M_{\gamma}} \quad (\text{R2-3})$$

3)

$$\text{HAD} = \frac{\frac{\gamma}{\beta}}{\frac{\alpha}{\beta} M_{\alpha} + M_{\beta} + \frac{\gamma}{\beta} M_{\gamma}} = \frac{\frac{\gamma}{\beta}}{\frac{O1}{2O2} M_{\alpha} + M_{\beta} + \frac{N1+N3}{N2} M_{\gamma}} \quad (\text{R2-4})$$

4)

Thus, we obtained the calculation formula for two parameters quantifying terminal groups of polyamides. Through HCD and HAD, we can determine the density of terminal groups on the surface of polyamide and correlate it with the dissociation of polyamide (*Proc. Natl. Acad. Sci. U. S. A.*, 2020, 117: 30191-30200). It can even be mutually confirmed with the terminal group content of polyamide obtained by precision analysis devices such as titration, quartz crystal

microbalance (QCM) or Rutherford backscattering spectrometry (*Environ. Sci. Technol.*, 2021, 55: 6984-6994; *J. Membr. Sci.*, 2013, 429: 23-33; *Environ. Sci. Technol.*, 2008, 42: 5260-5266). In the actual separation process, it is necessary to consider the hydrolysis of carboxyl and amino groups:

$$\frac{[\text{H}_3^+\text{O}][\text{R-COO}^-]}{[\text{R-COOH}]} = 10^{-2.12} \quad (\text{R2-5})$$

$$\frac{[\text{H}_3^+\text{O}][\text{R-NH}]}{[\text{R-NH}_2^+]} = 10^{-9.83} \quad (\text{R2-6})$$

The dissociated carboxyl and amino groups will contribute opposite charge for polyamide, respectively. The total charge of the polyamide nanofilms can be considered to be related to the difference of dissociated carboxyl and amino groups:

$$[\text{R-COO}^-] - [\text{R-NH}_2^+] = \frac{10^{-2.12}[\text{R-COOH}]}{[\text{H}_3^+\text{O}]} - \frac{[\text{H}_3^+\text{O}][\text{R-NH}]}{10^{-9.83}} \quad (\text{R2-7})$$

Considering that the usual nanofiltration process involves separating electrically neutral Na_2SO_4 solutions, pH equals to 7.

$$[\text{R-COO}^-] - [\text{R-NH}_2^+] \propto 10^{4.88}[\text{R-COOH}] - 10^{2.83}[\text{R-NH}] \quad (\text{R2-8})$$

Where, $[\text{R-COOH}]$ and $[\text{R-NH}]$ are the number of carboxyl and amino groups per unit volume of polyamide, proportional to our HCD and HAD:

$$[\text{R-COO}^-] - [\text{R-NH}_2^+] \propto 10^{4.88}\text{HCD} - 10^{2.83}\text{HAD} \approx 10^{4.88}\text{HCD} \quad (\text{R2-9})$$

To illustrate the significance of Formula R2-9, we examined the correlation between HCD and zeta potential. Zeta potential is the potential of the shear plane in the surface double layer of polyamide nanofilms. The value of Zeta potential should be related to the amount of negative charge on the surface of the polyamide. Therefore, researchers often use Zeta potential to measure the negative charge carried by polyamides membranes, especially in NF membranes. Formula R2-9 indicates that the content of surface carboxyl groups (HCD) should theoretically

form a positive correlation with the negative charge carried on polyamide nanofilms. As HCD increases, polyamide membranes should exhibit lower Zeta potentials.

Fig. R2-2 Correlation analysis between Zeta potential (A) and Na₂SO₄ rejection (B) of NF membranes with the HCD of PIP-TMC polyamides (pH = 7).

We analyzed the correlation between the HCD values calculated by our method and zeta potential and Na₂SO₄ rejection of recent NF membranes (Fig. R2-2). As expected, there is a good negative correlation between HCD and Zeta potential. PIP-TMC polyamide with higher HCD will exhibit stronger negative charge and higher Na₂SO₄ rejection. Nonetheless, the rejection of nanofiltration membranes is contributed by two factors: the screening of pores brought by polyamide networks and the Donnan effect of surface charge. It is difficult to obtain reliable structure-activity relationships cutting the two factors apart. We recommend the comprehensive use of our proposed structural parameters HABD and HCD. Our method provides a possibility to comprehensively analyze the rejection of nanofiltration, as it can provide HABD for the compactness of polyamide crosslinked networks, as well as HCD and HAD for the charge density of the polyamide nanofilms. These structural parameters can be cross-referenced with other characterization methods, providing a valuable tool for analyzing polyamides.

In response to the reviewer's suggestion, we have added these reliable results in the revised manuscript on page 15. Fig. R2-2A and B were added in the Supporting Information marked as

Fig. S17 and Fig. S18. Tab. S9 was added in the Supporting Information, which summarized the Zeta potential, Na₂SO₄ rejection in NF and HCD in PIP-TMC polyamide. The details regarding the calculation of HCD and HAD have been incorporated into the Supporting Information in Section 4.

On page 15:

“To quantify terminal groups and analyze their ionization behavior, we have introduced two analogous parameters, namely the harmonic carboxyl density (HCD) and the harmonic amino density (HAD). These parameters, reflecting the contents of carboxyl and amino groups per mass of polyamide, are essential in accurately representing the surface charge of polyamides (Section 4 in SI). It was demonstrated that PIP-TMC polyamide with higher HCD exhibits a stronger negative charge and higher Na₂SO₄ rejection (Tab. S9, Fig. S17 and Fig. S18 in SI). This finding strongly supports the notion that incorporating strong negative charges is effective in rejecting divalent anions when designing nanofiltration membranes. Nonetheless, the rejection of nanofiltration membranes is contributed by two factors: the screening of pores brought by polyamide networks and the Donnan effect of surface charge. Therefore, it is challenging to establish reliable structure-activity relationships by isolating these two factors. We recommend the comprehensive use of our proposed structural parameters – HAD and HCD. Our method offers a possibility to comprehensively analyze the rejection of nanofiltration, by providing information on HAD for the compactness of crosslinked polyamide networks, as well as HCD and HAD for the charge density of the polyamide nanofilms. These structure parameters can be cross-referenced with other characterization methods, providing a valuable tool for analyzing polyamides.”

Tab. S9. Summary of Zeta potential, Na₂SO₄ rejection in NF and HCD in PIP-TMC polyamide

N1+N3 (%)	N2 (%)	O1 (%)	HCD (mmol g⁻¹)	Na₂SO₄ rejection (%)	Zeta potential (mV)	Testing conditions of zeta potential testing conditions	Ref.
0.33	12.83	2.80	1.05	96.5	N. A.	N. A.	63

0.36	12.80	3.35	1.24	96.6	N. A.		
2.97	8.43	7.06	3.00	99.0	-24	CeO2 + Polysulfone, on Malvern Zetasizer Nano ZS,	64
3.54	8.96	7.25	2.88	98.0	-34		
4.09	9.11	4.73	1.96	95.8	-42		
0.16	12.90	3.83	1.40	97.2	N. A.		
0.07	11.16	8.55	3.13	99.2	N. A.	N. A.	72
0.30	11.65	7.34	2.66	97.8	N. A.		
0.16	12.90	4.03	1.47	98.4	N. A.		
2.25	8.22	7.64	3.33	95.0	-8		
2.06	8.20	6.65	3.02	97.0	-35	Alginate Hydrogel + Polyethersulfone, on SurPASS, Anton Paar GmbH, Graz	75
2.36	8.18	7.31	3.22	96.0	-35		
2.21	9.31	6.73	2.77	93.5	-30		
0.93	16.76	11.67	2.86	97.5	-35		
N. A.	N. A.	N. A.	3.79^a	92.0	-40		
N. A.	N. A.	N. A.	3.31^a	95.5	-52	EVOH + PET, on Austrian Anton Paar surpass	78
0.31	11.58	4.79	1.87	94.0	-24	Gelatin + PSf, on SurPASSTM3, Anton- Paar GmbH	79
N. A.	N. A.	N. A.	N. A.	98.5	-28		
0.40	11.88	3.02	1.20	98.5	-28		
N. A.	N. A.	N. A.	N. A.	98.5	-48		
N. A.	N. A.	N. A.	N. A.	98.5	-50		
0.30	5.39	1.54	1.33	91.3	N. A.	N. A.	80
0.38	9.48	2.81	1.38	96.4	N. A.		
2.40	7.10	13.14	5.31	N. A.	N. A.	N. A.	84
1.95	7.55	14.21	5.47	N. A.	N. A.		
2.11	6.78	14.29	5.78	94.0	N. A.		
2.30	6.71	15.21	5.99	91.0	N. A.		
N. A.	N. A.	N. A.	3.96^a	98.4	-38.83		

N. A.	N. A.	N. A.	3.65^a	99.2	-34.71	Amino functionalized PSf, on SurPASSTM3	
N. A.	N. A.	N. A.	3.31^a	99.5	-27.84		
N. A.	N. A.	N. A.	3.06^a	99.5	-24.12		
N. A.	N. A.	N. A.	2.54^a	99.2	-15.41		
N. A.	N. A.	N. A.	2.78^a	99.0	-12.54		
2.05	10.74	5.12	1.98	99.7	-22	PES, on SurPASS Anton Paar, GmbH	95
2.33	10.12	5.58	2.22	99.6	-24		
2.20	10.09	6.39	2.50	99.5	-25		
2.58	9.08	8.71	3.40	99.4	-25		
5.86	5.22	11.83	5.08	97.8	-28		
2.22	11.69	2.85	1.09	99.6	N. A.	N. A.	96
2.28	10.54	4.28	1.71	98.8	N. A.		
1.82	11.77	2.94	1.13	98.0	N. A.		
3.21	7.02	9.63	4.21	N. A.	N. A.	N. A.	98
2.82	6.41	13.82	5.68	95.2	N. A.		
3.77	6.37	11.77	4.99	97.0	N. A.		
2.91	6.93	8.47	3.91	98.0	N. A.		
3.58	6.26	12.19	5.18	98.6	N. A.	N. A.	99

a: This work directly provided the proportion of carboxyl group, amide bond, and amino group.

Comment 4-1: 2) The second point is the trickiest part of the work. Firstly, the methods used to calculate the HABD index from the XPS spectra are not explained in either the manuscript or the supplementary information (SI). In my opinion this is mandatory to publish the work, because it is very important for the reliability of all the work data.

Similarly to Table S2 (SI), where brief details of the membrane experiments are given, another Table should be included with the conditions under which the XPS spectra have been recorded and analysed (sample conditioning, vacuum level, area of analysis, X-ray type and energy, X-ray exposure time, ...) in the bibliography. Moreover, the method that the authors have followed

to compute the HABD parameter with the data from the bibliography must be clearly explained. In addition to mentioning the number of peaks that are included in the analysis of the spectra (briefly described in Fig 2 and S2 and Tab. S1), more details must be included in the SI about how they have got the original spectra, the software that has been used, type of background, type of fitting curve (Gaussian, Lorentzian, ...) and any possible fitting constrictions (maximum or minimum peak width or height, peak position, ...), as well as giving some parameter that quantifies the fit quality. If the authors have obtained the atomic concentrations of the N1, N2, ... bands directly from the bibliography, some of this information should be summarized in this additional Table.

Response: We thank the reviewer for the comment and valuable suggestions. In our correlation analysis, we directly obtained the XPS analysis results from recent researches, including the proportion of C, N, O atoms and the peak splitting results of N1, N2 and N3.

We adopted their peak fitting results for N, rather than reprocessing them ourselves. However, we excluded the peak fitting results for C or O from the cited works due to their lack of consideration for the cross-peak self-consistency relationship between atoms, such as the equality of N2 and O2. In our calculation of HABD, we directly equated O2 in these works to N2. The remaining portion obtained by subtracting O2 from the total O is assumed to represent O1 of polyamide

$$O1 = O - O2 \quad (R2-10)$$

After obtaining the data of N1, N2, N3, O1, O2, the calculation of HABD can be performed using the following process. Taking MPD-TMC polyamide as an example, we divide polyamide into three basic structure cells: carboxyl structure cell α , amide bond structure cell β , and amino structure cell γ . Each structure cell is distinguished by the presence of exclusively one hydrolyzed carboxyl group, one amide bond, or one unreacted amino group. Thus, the number of amide bonds is equal to the number of amide bond structure cell β (βN_A). The total mass of polyamide (m) can be expressed as the product of each structure cell and its relative weight,

because our segmentation method counted all the atoms in polyamide:

$$m = \frac{\alpha M_{\alpha} + \beta M_{\beta} + \gamma M_{\gamma}}{N_A} \quad (\text{R2-11})$$

11)

thus, HABD is equal to:

$$\text{HABD} = \frac{\frac{\beta}{N_A}}{m} = \frac{\beta}{\alpha M_{\alpha} + \beta M_{\beta} + \gamma M_{\gamma}} = \frac{1}{\frac{\alpha}{\beta} M_{\alpha} + M_{\beta} + \frac{\gamma}{\beta} M_{\gamma}} \quad (\text{R2-12})$$

12)

where, the ratio of α/β is numerically equivalent to the ratio of carboxyl group to amide bonds since one carboxyl group contains two O1 atom. That is, the ratio of $\frac{O1}{2}$ to O2. Similarly, the ratio of γ/β is numerically equivalent to the ratio of amino group to amide bonds, which is the ratio of (N1+N3) to N2. N1, N2, and N3 are the number of amino N, amide N, and protonated N of polyamide. Here, amino N are further classified as N1 and N3 for the convenience of peak fitting during XPS data process, which are detailed discussed afterwards. HABD is equal to:

$$\text{HABD} = \frac{1}{\frac{\alpha}{\beta} M_{\alpha} + M_{\beta} + \frac{\gamma}{\beta} M_{\gamma}} = \frac{1}{\frac{O1}{2O2} M_{\alpha} + M_{\beta} + \frac{N1+N3}{N2} M_{\gamma}} \quad (\text{R2-13})$$

In this way, we have achieved the entire process of calculating HABD through XPS data.

Following the reviewer's suggestion, we have highlighted our process of obtaining XPS data in the revised manuscript on page 17, Section 3.3 and 3.6 in SI. Tab. S4 and Tab. S6 were added in the Supporting Information, which shown the obtained XPS peak fitting results and calculated HABD in MPD-TMC and PIP-TMC polyamides. When it comes to the XPS test conditions, Tab. S10 and Tab. S11 were added in the Supporting Information, which shown the sample conditioning, vacuum level, area of analysis, X-ray type and energy, X-ray exposure time of related researches.

On page 17:

“It is important to acknowledge that the element proportions and N peak fitting results in the calculation of HABD for MPD-TMC and PIP-TMC polyamide were directly obtained from corresponding references (Tab. S4 and Tab. S6 in SI). A considerable portion of the studies presented in Fig. 3A, B and Fig. 4A, B do not provide complete XPS analysis results. As a

result, we only considered studies that provided the necessary data for HABD calculation. Comparatively, the peak fitting results for C or O in the cited works were abandoned because most of them did not account for the cross-peak self-consistency relationship between atoms. Therefore, in our HABD calculation process, O₂ in these works was directly considered equal to N₂”

In Section 3.3, SI:

“We directly obtained the XPS analysis results from recent researches, including the proportion of C, N, O atoms and the peak splitting results of N₁, N₂ and N₃. We adopted their peak fitting results for N, and excluded the peak fitting results for C or O from the cited works due to their lack of consideration for the cross-peak self-consistency relationship between atoms, such as the equality of N₂ and O₂. In our calculation of HABD, we directly equated O₂ in these works to N₂. The remaining portion obtained by subtracting O₂ from the total O is assumed to represent O₁ of polyamide. The HABD of MPD-TMC polyamide is calculated by Formula 2 in main text based on the N₁, N₂, N₃ and O₁ from corresponding references.”

In Section 3.6, SI:

“We directly obtained the XPS analysis results from recent researches, including the proportion of C, N, O atoms and the peak splitting results of N₁, N₂ and N₃. We adopted their peak fitting results for N, and excluded the peak fitting results for C or O from the cited works due to their lack of consideration for the cross-peak self-consistency relationship between atoms, such as the equality of N₂ and O₂. In our calculation of HABD, we directly equated O₂ in these works to N₂. The remaining portion obtained by subtracting O₂ from the total O is assumed to represent O₁ of polyamide. The HABD of PIP-TMC polyamide is calculated by Formula 2 in main text based on the N₁, N₂, N₃ and O₁ from corresponding references.”

Tab. S4. Summary of rejection, permeability of NaCl and water permeance in RO and HABD of MPD-TMC polyamide membranes in Fig. 3C and D.

C (%)	O (%)	N (%)	N1+N3 (%)	N2 (%)	O1 (%)	HABD (mmol g ⁻¹)	Rejection (%)	Water permeance (L m ⁻² h ⁻¹ bar ⁻¹)	NaCl Permeability coefficient (L m ⁻² h ⁻¹)	Ref.
76.9	11.87	11.23	0	11.23	0.64	9.25	98.8	0.68	0.13	24
76.74	12.47	10.71	0.32	10.38	2.08	8.71	98.7	1.69	0.35	
76.56	12.27	11.17	0	11.17	1.10	9.13	98.2	1.00	0.28	
77	11.7	11.2	0.38	10.82	0.88	9.02	99.1	1.73	0.24	26
76.9	11.8	11.4	0.36	11.04	0.76	9.06	99.2	2.78	0.34	
76.28	13.3	10.41	N. A.	N. A.	N. A.	N. A.	98.4	3.64	0.95	
75.67	14.23	10.1	N. A.	N. A.	N. A.	N. A.	97.3	2.94	1.30	
73.1	13.1	13.9	1.28	12.62	0.48	8.86	99.5	2.60	0.20	34
76.35	12.75	10.7	0.92	9.78	2.97	8.20	98.9	1.26	0.21	39
76.28	12.52	11.07	1.10	9.97	2.55	8.26	99.2	1.68	0.20	
78.47	11.46	10.07	0.8	9.27	2.19	8.40	98.2	6.80	1.25	9
70.33	19.96	9.7	1.17	8.53	11.43	6.23	88.7	2.97	3.02	54
68.28	22.86	8.85	1.45	7.40	15.46	5.27	93.3	2.57	1.48	
71.06	19.94	9.01	0.71	8.30	11.64	6.26	94.5	2.42	1.13	
68.86	21.89	9.25	0.9	8.35	13.54	5.93	96.2	2.12	0.66	
65.91	27.14	6.96	0.63	6.33	20.81	4.41	97.4	1.89	0.40	
67.53	24.43	8.03	0.77	7.26	17.17	5.14	97.8	1.77	0.31	
65.61	28.27	6.12	0.5	5.62	22.65	3.97	97.0	1.82	0.45	
69.18	21.55	9.28	0.97	8.31	13.24	5.94	97.1	1.73	0.42	
73.64	14.71	11.65	1.86	9.79	4.92	7.46	95.0	2.30	1.21	55
74.33	13.71	11.96	2.16	9.80	3.91	7.58	90.0	2.25	2.50	
74.81	13.25	11.93	1.99	9.94	3.31	7.78	94.5	2.60	1.51	

74.54	12.92	12.54	2.51	10.03	2.89	7.71	98.7	3.05	0.40	
69.63	19.19	10.91	0.66	8.73	10.46	6.57	97.20	6.20	0.71	56
69.09	19.77	10.89	0.58	8.71	11.06	6.49	97.30	8.30	0.92	
68.31	20.20	10.92	1.42	8.74	11.46	6.22	92.30	11.40	3.80	

Tab. S6. Summary of rejection, permeability of Na₂SO₄ and water permeance in NF and HABD of PIP-TMC polyamide membranes in Fig. 4C and D.

C (%)	O (%)	N (%)	N1+N3 (%)	N2 (%)	O1 (%)	HABD (mmol g ⁻¹)	Rejection (%)	Water permeance (L m ⁻² h ⁻¹ bar ⁻¹)	Na ₂ SO ₄ permeability coefficient (L m ⁻² h ⁻¹)	Ref.
70.98	15.63	13.16	0.33	12.83	2.80	9.63	96.5	8.1	1.76	63
70.46	16.15	13.16	0.36	12.80	3.35	9.48	96.6	26.5	5.60	
73.35	15.25	11.4	2.97	8.43	7.06	7.16	99.0	12.0	0.73	64
71.5	15.49	12.5	3.54	8.96	7.25	7.12	98.0	17.0	2.08	
69.3	16.21	13.2	4.09	9.11	4.73	7.54	95.8	20.0	5.26	
69.65	18.19	12.17	0.16	12.90	3.83	9.43	97.2	7.5	2.15	72
70.21	16.73	13.06	0.07	11.16	8.55	8.19	99.2	13.4	1.08	
69.06	19.71	11.23	0.30	11.65	7.34	8.46	97.8	9.2	2.07	
69.05	18.99	11.95	0.16	12.90	4.03	9.38	98.4	15.4	2.50	
71.8	17.44	10.76	2.25	8.22	7.64	7.17	95.0	17.0	3.58	
73.75	15.86	10.47	2.06	8.20	6.65	7.44	97.0	23.0	2.85	75
74.48	14.85	10.26	2.36	8.18	7.31	7.20	96.0	25.5	4.25	
73.41	15.49	10.54	2.21	9.31	6.73	7.65	93.5	26.0	7.23	
71.48	16.04	11.52	0.93	16.76	11.67	8.20	97.5	30.5	3.13	78
70.3	19.89	9.82	N. A.	N. A.	N. A.	7.31^a	92.0	26.0	10.87	
70.29	19.12	10.59	N. A.	N. A.	N. A.	7.86^a	95.5	41.7	9.42	

73.61	15.18	11.21	0.31	11.58	4.79	9.03	94.0	6.8	4.34	
71.74	16.37	11.89	N. A.	N. A.	N. A.	N. A.	98.5	13.0	1.98	
70.5	16.67	12.83	0.40	11.88	3.02	9.48	98.5	17.0	2.59	79
72.82	14.9	12.28	N. A.	N. A.	N. A.	N. A.	98.5	16.5	2.51	
74.69	13.69	11.62	N. A.	N. A.	N. A.	N. A.	98.5	15.0	2.28	
87.18	6.93	5.69	0.30	5.39	1.54	9.31	91.3	6.5	3.10	80
77.49	12.29	9.86	0.38	9.48	2.81	9.33	96.4	8.5	1.59	
70.26	20.24	9.5	2.40	7.10	13.14	5.73	N. A.	N. A.	N. A.	
68.74	21.76	9.5	1.95	7.55	14.21	5.81	N. A.	N. A.	N. A.	84
70.04	21.07	8.89	2.11	6.78	14.29	5.49	94.0	50.0	19.15	
69.06	21.92	9.01	2.30	6.71	15.21	5.29	91.0	55.0	32.64	
71.65	16.38	11.97	N. A.	N. A.	N. A.	7.60^a	98.4	8.4	1.37	
71.93	15.95	12.12	N. A.	N. A.	N. A.	7.81^a	99.2	9.3	0.75	
71.9	15.72	12.38	N. A.	N. A.	N. A.	7.99^a	99.5	11.7	0.59	88
71.89	15.47	12.64	N. A.	N. A.	N. A.	8.13^a	99.5	13.5	0.68	
72.07	14.96	12.97	N. A.	N. A.	N. A.	8.40^a	99.2	16.8	1.35	
72.8	15.11	13.01	N. A.	N. A.	N. A.	8.19^a	99.0	16.9	1.71	
71.35	15.86	12.79	2.05	10.74	5.12	8.33	99.7	13.8	0.29	
71.85	15.7	12.45	2.33	10.12	5.58	8.04	99.6	14.9	0.40	
71.24	16.48	12.29	2.20	10.09	6.39	7.90	99.5	17.5	0.57	95
70.54	17.79	11.66	2.58	9.08	8.71	7.10	99.4	20.9	0.76	
71.87	17.05	11.08	5.86	5.22	11.83	4.49	97.8	24.5	3.27	
71.55	14.54	13.91	2.22	11.69	2.85	8.94	99.6	20.0	0.48	
72.63	14.82	12.82	2.28	10.54	4.28	8.43	98.8	17.5	1.28	96
71.71	14.71	13.59	1.82	11.77	2.94	9.05	98.0	20.0	2.45	
73.12	16.65	10.23	3.21	7.02	9.63	6.14	N. A.	N. A.	N. A.	
70.54	20.23	9.23	2.82	6.41	13.82	5.28	95.2	34.0	10.29	98
71.72	18.14	10.14	3.77	6.37	11.77	5.39	97.0	18.0	3.34	

74.74	15.4	9.85	2.92	6.93	8.47	6.41	98.0	10.0	1.22	
67.35	18.45	9.84	3.58	6.26	12.19	5.32	98.6	45.5	1.29	⁹⁹

a: This work directly provided the proportion of carboxyl group, amide bond, and amino group.

Tab. S10. XPS test conditions of RO membranes in Fig. 3.

Sample conditioning	Instruments	Vacuum level	Area of analysis	X-ray type and energy	X-ray exposure time	Data analysis	Ref.
TFC membrane	Axis Ultra DLD, Kratos Analytical	N. A.	N. A.	N. A.	N. A.	Casa XPS software	Sep. Purif. Technol. 2022, 281 , 119884. ¹⁸
TFC membrane	AXIS Supra	N. A.	300 × 700 μm	Al Kα (1486.6 eV)	N. A.	Casa XPS software	J. Membr. Sci. 2021, 640 , 119805. ¹⁹
TFC membrane	Thermo Scientific 49 ESCALAB 250Xi,	N. A.	N. A.	N. A.	N. A.	N. A.	J. Environ. Chem. Eng. 2022,

							10, 106958 . ²⁰
TFC membrane, rinsed 3 times with ultrapure water followed by freeze- drying	Kratos AXIS Supra (UK) system	N. A.	N. A.	Al K α	N. A.	N. A.	J. Membr. Sci. 2021, 633, 119395 . ²¹
TFC membrane	K-alpha	N. A.	N. A.	Al K α	N. A.	N. A.	Chem. Eng. Res. Des. 2021, 165, 1- 11. ²²
TFC membrane	X-tool spectrometer, ULVAC-PHI	N. A.	N. A.	N. A.	N. A.	N. A.	J. Membr. Sci. 2021, 620, 118870 . ²³
TFC membrane	Kratos Inc., AXIS-His	N. A.	N. A.	Al (1486.6eV)	N. A.	N. A.	J. Membr.

							Sci. 2021, 618 , 118677 . ²⁴
TFC membrane	JPS-9010 MC, JEOL	N. A.	N. A.	N. A.	N. A.	N. A.	Desalin ation 2021, 516 , 115222 . ²⁵
TFC membrane	ESCALAB 250 XI	N. A.	N. A.	N. A.	N. A.	N. A.	J. Membr. Sci. 2021, 638 , 119680 . ²⁶
TFC membrane	Quanta 200 spectrometer	N. A.	N. A.	N. A.	N. A.	N. A.	ACS Appl. Mater. Interfac es 2020, 12 , 25304- 25315. ²⁷

EDX	N. A.	N. A.	N. A.	N. A.	N. A.	N. A.	J. Membr. Sci. 2019, 574 , 1-9. ²⁸
TFC membrane	VG Multilab 2000, Thermo VG Scientific	N. A.	N. A.	Al K α (1486.6 eV)	N. A.	N. A.	J. Membr. Sci. 2019, 570 , 112-119. ²⁹
TFC membrane	Axis Ultra DLD, Kratos Analytical	N. A.	N. A.	N. A.	N. A.	N. A.	Rsc Adv. 2018, 8 , 25236-25247. ³⁰
TFC membrane	Escalab 250Xi, Thermo Fisher Scientific	N. A.	N. A.	N. A.	N. A.	N. A.	J. Appl. Polym. Sci. 2018, 135 , 46261. ³¹

EDX	N. A.	N. A.	N. A.	N. A.	N. A.	N. A.	Science 2018, 361 , 682- 685. ³²
TFC membrane	Thermal Scientific K- Alpha	N. A.	N. A.	N. A.	N. A.	N. A.	Desalin ation 2020, 480 , 114342 . ³³
TFC membrane	ULVAC PHI X- tool	N. A.	N. A.	N. A.	N. A.	N. A.	J. Membr. Sci. 2020, 614 , 118449 . ³⁴
TFC membrane	Shimadzu, AXIS Ultra DLD	N. A.	N. A.	N. A.	N. A.	N. A.	Desalin ation 2020, 491 , 114345 . ³⁵

TFC membrane	N. A.	6×10^{-10} Torr	N. A.	Al K α (1486.6 eV)	N. A.	N. A.	J. Membr. Sci. 2020, 611 , 118407 . 36
TFC membrane	Quanta 200 spectrometer, FEI.	N. A.	N. A.	N. A.	N. A.	N. A.	J. Membr. Sci. 2020, 612 , 118412 . 37
TFC membrane	ESCALAB 250Xi	N. A.	N. A.	N. A.	N. A.	N. A.	J. Membr. Sci. 2020, 614 , 118498 . 38
TFC membrane	ESCALAB 250 XI	N. A.	N. A.	N. A.	N. A.	N. A.	J. Membr. Sci. 2020, 604 , 118065 . 39

TFC membrane	X-tool spectrometer, ULVAC-PHI	N. A.	N. A.	N. A.	N. A.	N. A.	J. Membr. Sci. 2019, 578 , 220-229. ⁴⁰
Self-standing PA nanofilm	PHI-5000	N. A.	N. A.	Al K α (1.49 keV)	N. A.	N. A.	J. Membr. Sci. 2017, 526 , 52-59. ⁴¹
TFC membrane	Kratos AXIS ULTRA, UK	N. A.	N. A.	N. A.	N. A.	N. A.	J. Membr. Sci. 2017, 535 , 248-257. ⁴²
TFC membrane	Thermo escalab 250Xi	N. A.	N. A.	Al K α (1486.6 eV)	N. A.	N. A.	J. Taiwan Inst. Chem. Eng. 2017,

								80, 25-33. ⁴³
TFC membrane	PHI-1600	N. A.	N. A.	N. A.	N. A.	N. A.	N. A.	J. Membr. Sci. 2017, 541 , 174-188. ⁴⁴
TFC membrane	Thermo escalab 250Xi, USA	N. A.	N. A.	N. A.	N. A.	N. A.	N. A.	J. Membr. Sci. 2017, 541 , 39-52. ⁴⁵
TFC membrane	Thermo Electron, K-Alpha	N. A.	N. A.	N. A.	N. A.	N. A.	N. A.	J. Membr. Sci. 2017, 541 , 510-518. ⁴⁶

							2016, 515 , 79-85. 51
TFC membrane, storage in deionized water and completely dried under vacuum before XPS	PHI-1600	N. A.	N. A.	Mg K α	N. A.	PHI- MAT LAB softw are packa ge	Sep. Purif. Technol . 2010, 75 , 145- 155. ⁵²
Self-standing PA nanofilm loaded on gold-plated silicon wafers	Escalab250Xi, ThermoFisher	N. A.	N. A.	Al K α (1486.6 eV)	N. A.	N. A.	Angew. Chem., Int. Ed. 2021, 60 , 14636- 14643. 9

TFC membrane	PHI-1600	N. A.	N. A.	Al K α (1486.6 eV)	N. A.	N. A.	Ind. Eng. Chem. Res. 2020, 59 , 8230-8242. ⁵³
Self-standing PA nanofilm loaded on silicon wafers	Thermo Scientific K-Alpha	N. A.	N. A.	Al K α (1486.6 eV)	N. A.	N. A.	Sep. Purif. Technol. 2023, 310 , 123122. ⁵⁴
Self-standing PA nanofilm loaded on silicon wafers	Thermo Scientific K-Alpha	N. A.	N. A.	Al K α (1486.6 eV)	N. A.	N. A.	Desalination 2023, 545 , 116166. ⁵⁵
TFC membrane	K-Alpha, Thermo Fisher Scientific	N. A.	N. A.	N. A.	N. A.	N. A.	Environ. Sci. Technol. 2021, 55 ,

							6984-6994 ⁵⁶
TFC membrane	ThermoFisher K-alpha	N. A.	N. A.	Al K α (1486.6 eV)	N. A.	N. A.	Desalination 2011, 274 , 136-143. ⁵⁷
TFC membranes, extensively rinsed and soaked in MilliQ water for 24 h before dried in a vacuum.	SSI S-Probe Monochromatized XPS Spectrometer	N. A.	250 μm \times 1000 μm	Al K α (1486.6 eV)	N. A.	N. A.	Desalination 2009, 242 , 168-182. ⁶⁰

Tab. S11. XPS test conditions of NF membranes in Fig. 4.

Sample conditioning	Instruments	Vacuum level	Area of analysis	X-ray type and energy	X-ray exposure time	Data analysis	Ref.
-------------	--------------	------------------	-----------------------	---------------------	---------------	------

TFC membranes	ESCALAB 250	N. A.	N. A.	N. A.	N. A.	N. A.	J. Environ. Chem. Eng. 2022, 10 , 107015. 62
TFC membranes	Thermal Fisher Scientific ESCALAB 250 Xi)	N. A.	N. A.	N. A.	N. A.	XPSPE AK41 software	Proc. Natl. Acad. Sci. U. S. A. 2021, 118 , 118. 63
TFC membranes	XRF-1800	N. A.	N. A.	N. A.	N. A.	N. A.	J. Membr. Sci. 2022, 641 , 119887. 64
TFC membranes	K-alpha, Thermo Fisher, USA	N. A.	N. A.	N. A.	N. A.	N. A.	Sep. Purif. Technol. 2022, 280 ,

							119964. 65
TFC membranes	K-alpha, Thermo Fisher Scientific	N. A.	N. A.	Al K α (1486.6 eV)	N. A.	N. A.	J. Membr. Sci. 2021, 625 , 119154. 66
TFC membranes	Kratos Analytical-A	N. A.	N. A.	N. A.	N. A.	N. A.	Sep. Purif. Technol. 2021, 270 , 118802. 67
TFC membranes	Thermal Fisher Scientific ESCALAB	N. A.	N. A.	N. A.	N. A.	XPSPE AK software	Macromol. Chem. Phys. 2021, 222 , 2100222 68

TFC membranes	Axis Ultra DLD	N. A.	N. A.	N. A.	N. A.	N. A.	Sep. Purif. Technol. 2021, 275 , 119227. 69
TFC membranes	AXIS UltraDLD	N. A.	N. A.	N. A.	N. A.	N. A.	J. Membr. Sci. 2021, 638 , 119699. 70
N. A.	ESCALAB25 0Xi	N. A.	N. A.	N. A.	N. A.	N. A.	J. Membr. Sci. 2021, 634 , 119450. 71
N. A.	N. A.	N. A.	N. A.	N. A.	N. A.	N. A.	Desalination 2021, 512 , 115118. 72

							2021, 617 , 118645. 77
TFC membranes, samples were placed under the infrared lamp for 1 h.	Thermo Kalpha	N. A.	N. A.	N. A.	N. A.	N. A.	ACS Appl. Mater. Interfaces 2021, 13 , 23142-23152. ⁷⁸
TFC membranes	Quanta200 spectrometer	N. A.	N. A.	N. A.	N. A.	N. A.	Sep. Purif. Technol. 2021, 264 , 118391. 79
TFC membranes	N. A.	N. A.	N. A.	N. A.	N. A.	N. A.	J. Membr. Sci. 2021, 635 , 119523. 80

TFC membranes	Thermo Scientific MultiLab 2000	N. A.	specific spot size of 650 μm	Al K α (1486.6 eV)	N. A.	N. A.	J. Ind. Eng. Chem. Chem. 2021, 103 , 373-380. 81
TFC membranes	Thermo Fisher S4 Scientific Escalab 250Xi	N. A.	N. A.	Al K α (1486.6 eV)	N. A.	N. A.	Chem. Res. Chin. Univ. 2021, 37 , 1101-1109. ⁸²
TFC membranes	Quanta 200 spectrometer	N. A.	N. A.	N. A.	N. A.	N. A.	ACS Appl. Mater. Interface s 2020, 12 , 25304-25315. ²⁷

TFC membranes	PerkinElmer	N. A.	N. A.	Al K α (1486.6 eV)	N. A.	N. A.	J. Membr. Sci. 2020, 593 , 117444. 83
TFC membranes	AXIS Supra	N. A.	N. A.	Al K α (1486.6 eV)	N. A.	CasaXP S software	J. Mater. Chem. A 2020, 8 , 3238- 3245. ⁸⁴
TFC membranes	Thermo Scientific	N. A.	N. A.	N. A.	N. A.	N. A.	Membranes 2020, 10 , 12. ⁸⁵
TFC membranes	ESCALAB25 0Xi	N. A.	N. A.	N. A.	N. A.	N. A.	Desalination 2020, 491 , 114499. 86

TFC membranes	Quanta 200	N. A.	N. A.	N. A.	N. A.	N. A.	Desalination 2020, 488 , 114525. ⁸⁷
TFC membranes	Quanta 200	N. A.	N. A.	N. A.	N. A.	N. A.	Desalination 2020, 496 , 114340. ⁸⁸
TFC membranes	Kratos AXIS Ultra DLD	N. A.	N. A.	N. A.	N. A.	The Shirley-type background and the Gaussian-Lorentz peak deconvolution were used for fitting the high-resolution	Sep. Purif. Technol. 2020, 230 , 11585. ⁸⁹

						n C 1s spectra	
TFC membranes	K-alpha, Thermo Fisher	N. A.	N. A.	N. A.	N. A.	N. A.	J. Mater. Chem. A 2020, 8 , 25028. ⁹⁰
TFC membranes	Thermo Fisher Scientific ESCALAB 250Xi XPS	N. A.	N. A.	N. A.	N. A.	N. A.	ACS Nano 2019, 13 , 5278-5290. ⁹¹
TFC membranes, freeze dried before test	K-alpha, Thermo Fisher	N. A.	N. A.	N. A.	N. A.	N. A.	J. Membr. Sci. 2018, 550 , 36-44. ⁹²
TFC membranes	PerkinElmer 5300	N. A.	N. A.	Al K α (1486.6 eV)	N. A.	N. A.	Langmuir 2017, 33 , 2318-2324. ⁹³
TFC membranes	K-alpha, Thermo Fisher Scientific	N. A.	N. A.	N. A.	N. A.	N. A.	Chem. Eng. J. 2021, 416 ,

							129154. 94
TFC membranes	ThermoScientific, K Alpha+	N. A.	N. A.	Al K α (1486.6 eV)	N. A.	N. A.	J. Membr. Sci. 2021, 627 , 119142. 95
TFC membranes	ThermoScientific, K-Alpha+	N. A.	N. A.	Al K α (1486.6 eV)	N. A.	N. A.	J. Membr. Sci. 2020, 616 , 118557. 96
TFC membranes	PerkinElmer	N. A.	N. A.	Al K α (1486.6 eV)	N. A.	N. A.	J. Membr. Sci. 2016, 515 , 238-244. 97
TFC membranes	PerkinElmer	N. A.	N. A.	Al K α (1486.6 eV)	N. A.	N. A.	J. Mater. Chem. A 2017, 5 , 16289-16295 ⁹⁸

TFC membranes	Thermo-Fisher	N. A.	N. A.	N. A.	N. A.	N. A.	J. Membr. Sci. 2021, 618 , 118738. 99
TFC membranes, extensively rinsed and soaked in MilliQ water for 24 h before dried in a vacuum.	SSI S-Probe Monochromatized XPS Spectrometer	N. A.	250 μm \times 1000 μm	Al K α (1486.6 eV)	N. A.	N. A.	Desalination 2009, 242 , 168-182. 60

Comment 4-2: Secondly, the authors said that the DNC index is also calculated from XPS data processing, and references 28 and 29 are used to justify this. As far as I know, XPS is not the technique usually used to calculate the DNC, and particularly DNC is not calculated in any of these two references. What is said in ref. 28 is that the shape of the C1s and O1s peak is greatly influenced by absorbed water and other contaminants, and other references in the bibliography also have showed that X-ray can modify the surface of some polymers (for example ref. DOI10.1002/sia.1542). All these effects will be significant or not depending on the sample and XPS measurements conditions. That is why the experimental conditions under which XPS measurements were performed in the bibliography should be briefly summarized in an additional Table, as well as the subsequent analysis of the spectra.

Response: We thank the reviewer for the recognition and valuable suggestions. The comments of XPS methods are very helpful for us to improve the reliability of our manuscript.

As we mentioned in introduction, the degree of network crosslinking (DNC) is usually calculated by the number of N and O:

$$\text{DNC} = \frac{X}{X+Y} = \frac{4-2\frac{O}{N}}{1+\frac{O}{N}} \quad (\text{R2-14})$$

XPS, is frequently employed as an auxiliary analytical tool to accurately characterize the surface chemical environment of materials. It helps explain the C, N and O atom ratio on the surface as well as their respective chemical states in the obtained polyamide nanofilms. This utilization of XPS for this purpose was initially introduced by Feral Temelli *et al.* (*Desalination*, 2011, 278: 387-396) and further popularized by Andrew G. Livingston *et al.* (*Science*, 2015, 348, 1347-1351), who demonstrated its efficacy in calculating DNC. Now, calculating DNC has become a standard practice in the analysis of polyamide-based thin film composite membranes. Despite this, concerns have been raised regarding the limited correlation between DNC and the performance of RO and NF membranes. Majority of cited researches in our work conducted XPS test and calculated DNC based on the element proportion. We directly obtained DNC data from their main text or SI. Part of researches did not calculate DNC because their DNC exceeded the defined domain, but XPS testing was still conducted to analyze the element proportion. For these researches, we calculated the DNC of polyamide based on the element proportion. We have added an explanation about DNC data sources in Section 3.2, Section 3.5, Tab. S3 and Tab. S5 in Supporting Information. The incorrect references 28 and 29 have been corrected to “*Desalination*, 2011, 278: 387-396” and “*Science*, 2015, 348, 1347-1351”

On the other hand, we agree with the reviewer that the testing conditions of XPS do have a significant impact on the obtained results. Following the reviewer’s suggestion, Tab. S10 and Tab. S11 were added in the Supporting Information, which shown the sample conditioning, vacuum level, area of analysis, X-ray type and energy, X-ray exposure time of related researches. However, the current research on RO and NF does not pay enough attention to XPS, and the detailed description of XPS testing conditions in most of the work manuscripts is also

quite insufficient as shown in Tab. S10 and Tab. S11. They generally lack descriptions of XPS testing conditions, especially vacuum level and area of analysis during XPS testing. In order to improve the standardization of XPS testing and the accuracy of the obtained peak fitting results and HABD, we referred to the standards of International Organization for Standardization (ISO 20579), American Society for Testing Materials (E1078), and National Standard of the People's Republic of China (GB/T SJT10458-1993) and related researches (*J. Vac. Sci. Technol. A*, 2019, 37: 031401; *J. Vac. Sci. Technol. A*, 2020, 38, 061203; *J. Vac. Sci. Technol. A*, 2020, 38: 063202) to provide a series of points about sample preparation and testing conditions that should be paid attention to when analyzing the surface of polyamides.

We look forward to your suggestions to improve the accuracy of these points for readers of our work to better conduct XPS testing. We have summarized the key points of XPS testing in the Section 5.2 in revised SI. The recommended work is also cited in Section 5.2.

In addition, the samples used for XPS peak fitting example in our work were prepared and tested under the following conditions. XPS test measured through Thermo Scientific K-Alpha+ at Al K α line (1486.6 eV, 15 mA \times 15 kV), the vacuum level is 5×10^{-9} mbar, X-ray beam spot is 300 μ m, total X-ray irradiation time is 154.1 s (68 s for full spectrum, 28.7 s for C, 30.2 s for O, and 27.2 s for N).

In Section 3.2, SI:

“DNC data in Tab. S3 is directly obtained from reference or calculated by Formula 1 in main text based on the measured element ratio obtained from reference. The salt permeability coefficient in RO can be calculated by Formula S3-3 from rejection, water permeance and applied pressure during the measurement.”

In Section 3.5, SI:

“DNC data in Tab. S5 is directly obtained from reference or calculated by Formula 1 in main text based on the measured element ratio obtained from reference. The salt permeability coefficient in NF can be calculated by Formula S3-3 from rejection, water permeance and applied pressure during the measurement.”

In Section 5.2, SI:

“In order to improve the standardization of XPS testing as well as the accuracy of the obtained peak fitting results and calculated HAD, we referred to the standards of International Organization for Standardization (ISO 20579), American Society for Testing Materials (E1078), and National Standard of the People's Republic of China (GB/T SJT10458-1993) and related researches^{16,105,106} to provide a series of recommendations regarding sample preparation and testing conditions for analyzing the surface of polyamides. We have summarized the key points as follows:

Sample preparation: For the conventional polyamide thin-film composite membranes, cleaning the solvents and unreacted monomers is crucial. We suggest choosing to soak and clean with ethanol for more than 24 hours. For the free-standing polyamide nanofilms developed in recent years, it can be considered to use silicon wafers or gold-plated silicon wafers to load samples for testing to avoid interference from foreign elements. After drying, the polyamide nanofilm can firmly adhere to the surface of the silicon wafer and should also be cleaned before testing.

Storage time: Minimize the storage time of test samples as much as possible. If storage is necessary, water can be considered to seal the samples, but drying should be carried out before testing. If possible, it is recommended to use rare gas to protect the sample. In our method, it is not recommended to protect the sample with N₂, because the subsequent analysis mainly uses N as the calibration.

Vacuum level: The vacuum level during XPS test should be below than 10⁻⁷ mbar.

X-ray beam spots: We recommend using larger X-ray beam spots (more than 200 μm) to observe polyamide in order to obtain results that better reflect the overall element proportion of polyamide. Readers can also perform mapping imaging with small X-ray beam spots and take the average of multiple test points on a large scale.

X-ray source: A stronger X-ray source is not recommended, as it may cause damage to the C-chain polymer^{107,108}. Mg Kα (1253.6 eV with natural linewidth of 0.7 eV) and Al Kα (1486.6 eV with natural linewidth of 0.9 eV) can obtain information about the surface of polyamide at

around 7~10 nm, which is acceptable. Moreover, the reported polyamide nanofilm has become thinner in recent years, and the comprehensiveness of Mg K α or Al K α is acceptable.

X-ray exposure time: X-rays will also cause certain damage to polyamide with exposure time prolong^{56,109}. Considering that polyamides typically require testing full spectrum and fine spectrum of C, N, and O in XPS, the total XPS irradiation time should not exceed 5 min.

We expect researchers to refer to relevant standards and researches before conducting XPS test. Following above key points can help to improve the accuracy of peak fitting result and to assist in the analysis of RO and NF materials along with the proposed HABD.”

Tab. S3. Summary of rejection, permeability of NaCl and water permeance in RO and DNC of MPD-TMC polyamide membranes in Fig. 3A and B.

C (%)	O (%)	N (%)	DNC (%)	NaCl rejection (%)	Test conditions	Water permeance (L m ⁻² h ⁻¹ bar ⁻¹)	NaCl permeability coefficient (L m ⁻² h ⁻¹)	Ref.
76.33	12.8	10.87	75.5	95.0	2000	5.60	2.95	Sep. Purif.
75.44	15.92	11.64	53.4	93.0	ppm	5.40	4.06	Technol.
69.91	17.68	12.41	47.5	91.5	@ 10 bar	4.80	4.46	2022, 281 , 119884. ¹⁸
75.92	13.23	10.85	70.3 ^a	98.0	2000 ppm @ 15.5 bar	4.00	1.27	J. Membr. Sci. 2021, 640 , 119805. ¹⁹
75.57	14.08	10.35	54.2 ^a	97.2		6.20	2.77	
75.36	13.64	11.00	67.9 ^a	98.0		5.70	1.80	
75.05	14.33	10.62	55.3 ^a	97.8		3.80	1.32	
75.31	14.49	10.20	47.9 ^a	97.5		4.90	1.95	
75.08	14.08	10.83	60.9 ^a	98.9		4.50	0.78	
73.49	16.16	10.36	34.4	92.1	1000 ppm @ 8 bar	1.92	1.32	J. Environ. Chem. Eng.
73.55	15.19	11.26	55.4	94.5	1.86	0.87		
72.97	16.9	10.31	27.3	91.2	1.95	1.51		

								2022, 10 , 106958. ²⁰
75.5	13.62	10.88	66.4	99.6	35000 ppm @50 bar	1.10	0.22	J. Membr.
74.9	13.73	11.37	71.8	99.4		1.60	0.48	Sci. 2021,
75.29	13.35	11.36	75.8	99.5		1.70	0.43	633,
79.27	13.45	7.28	10.7	98.7		1.20	0.79	119395. ²¹
68.9	16.95	9.02	8.4	99.0	2000 ppm @ 15 bar	2.80	0.42	Chem. Eng. Res. Des. 2021, 165 , 1-11. ²²
74.58	14.78	9.71	37.9	99.0		5.47	0.83	
73.5	14.28	10.94	60.3	98.0		7.00	2.14	
72.76	15.82	9.39	23.5	98.0		5.33	1.63	
72.45	15.5	9.5	28.0	98.0		5.80	1.78	
72.47	15.13	9.08	25.0	98.0		3.47	1.06	
73.53	13.9	9.8	48.1	99.0		3.33	0.51	
72	17.2	10.8	31.4 ^a	99.4	2000 ppm @ 15.5 bar	2.10	0.20	J. Membr.
73.2	16.1	10.7	39.6 ^a	99.4		3.00	0.28	Sci. 2021,
73.1	15.9	11	45.4 ^a	98.3		4.50	1.21	620,
73.5	15.4	11.1	51.3 ^a	95.2		6.30	4.92	118870. ²³
76.9	11.87	11.23	91.7 ^a	98.8	2000 ppm @ 15.5 bar	0.68	0.13	J. Membr.
76.74	12.47	10.71	77.2 ^a	98.7		1.69	0.35	Sci. 2021,
76.56	12.27	11.17	85.9 ^a	98.2		1.00	0.28	618, 118677. ²⁴
72.3	14.6	13.1	83.8	97.5	1000 ppm @ 10 bar	1.36	0.35	Desalinatio n 2021, 516, 115222. ²⁵
72.1	14.6	13.3	86.0	98.3		2.12	0.37	
71.7	15.5	12.8	71.4	95.3		2.58	1.27	
72.6	15.3	12.1	65.0	94.7		2.76	1.54	
73	15.3	11.7	60.0	92.8		2.41	1.87	
77	11.7	11.2	93.4	99.1	2000 ppm @ 15 bar	1.73	0.24	J. Membr.
76.9	11.8	11.4	94.8	99.2		2.78	0.34	Sci. 2021,

								638, 119680. ²⁶
68.23	15.03	9.17	27.4 ^a	98.3		2.90	0.75	ACS Appl.
63.71	19.24	8.21	-20.5 ^a	62.8	2000 ppm @ 15 bar	18.00	159.94	Mater. Interfaces 2020, 12 , 25304- 25315. ²⁷
74.6	13.1	12.3	90.6	98.0		1.73	0.53	
74.4	13.6	12	81.3	98.3		1.87	0.48	J. Membr.
74.3	13.8	11.9	77.8	98.5	2000 ppm @ 15 bar	2.07	0.47	Sci. 2019,
73.7	14.2	12.1	76.0	99.0		2.13	0.32	574 , 1-9. ²⁸
73.2	14.7	12.1	70.9	93.2		2.80	3.06	
72.21	17.51	10.28	22.0	94.7		1.99	0.22	
72.7	16.65	10.65	34.1	94.0		2.26	0.29	J. Membr.
72.52	16.52	10.96	39.3	94.3	1000 ppm	2.67	0.32	Sci. 2019,
77.31	13.36	9.33	46.7	95.9	@ 2 bar	3.44	0.29	570 , 112-
76.76	13.52	9.72	50.9	96.7		3.21	0.22	119. ²⁹
75.73	13.83	10.44	58.1	91.1		2.32	0.45	
77.41	12.12	10.47	78.1	98.1		3.07	0.95	Rsc Adv.
76.28	13.3	10.41	N. A.	98.4	2000 ppm	3.64	0.95	2018, 8 ,
75.67	14.23	10.1	N. A.	97.3	@ 16 bar	2.94	1.30	25236- 25247. ³⁰
76.41	14.23	9.36	38.1	N. A.		N.A.	N.A.	
73.66	14.63	11.71	66.7	N.A.		N.A.	N.A.	J. Appl.
73.59	14.39	12.02	73.1	N.A.	2000 ppm	N.A.	N.A.	Polym. Sci.
73.5	14.26	12.24	77.1	N.A.	@ 6.8 bar	N.A.	N.A.	2018, 135 ,
75.27	14.7	10.03	43.3	N.A.		N.A.	N.A.	46261. ³¹

74.84	16.04	9.12	17.5	N.A.		N.A.	N.A.	
73.06	15.07	11.24	56.3	87.9		2.01	1.89	
70	12	11	87.0 ^a	97.5	2000 ppm @ 10 bar	2.87	0.74	Science 2018, 361 , 682-685. ³²
70.13	17.47	11.28	35.4	92.6	2000 ppm @ 20 bar	1.40	2.24	Desalinatio n 2020, 480 , 114342. ³³
74.54	15.39	10.07	37.3	92.0		1.40	2.43	
73.32	15.33	11.35	55.2	94.0		1.60	2.04	
71.44	16.19	12.38	60.0	95.8		1.77	1.55	
73.78	14.3	11.92	72.8	90.0		2.00	4.44	
73.1	13.1	13.9	108.9	99.5	2000 ppm @ 15.5 bar	2.60	0.20	J. Membr. Sci. 2020, 614 , 118449. ³⁴
76.46	12.75	10.8	75.2	98.0	2000 ppm @ 15 bar	1.07	0.33	Desalinatio n 2020, 491 , 114345. ³⁵
77.14	12.44	10.42	73.5	97.0		1.33	0.62	
79.49	13.69	6.82	-0.5 ^a	98.0		1.60	0.49	
77.99	15.26	6.75	-16.0 ^a	96.0		1.47	0.92	
77.73	14.86	7.41	-0.4 ^a	97.0		1.80	0.84	
77.2	12.3	10.4	74.9 ^a	99.0	35000 ppm @ 55 bar	1.62	0.90	J. Membr. Sci. 2020, 611 , 118407. ³⁶
77.3	11.8	10.5	82.5 ^a	99.0		1.17	0.65	
93.36	4.04	2.6	34.9	N. A.	2000 ppm @ 15.5 bar	N. A.	N. A.	J. Membr. Sci. 2020, 612 , 118412. ³⁷
71.09	15.84	13.06	71.1	95.0		0.97	0.79	
73.25	13.58	13.17	95.4	98.0		3.23	1.02	
75.01	14.71	10.28	46.8 ^a	97.0		1.13	0.70	

75.21	17.12	10.67	30.4 ^a	97.9	2000 ppm @ 20 bar	2.32	1.00	J. Membr. Sci. 2020, 614 , 118498. ³⁸
75.57	13.81	10.62	60.8 ^a	97.3		3.15	1.75	
75.58	13.74	10.68	62.4 ^a	97.1		3.31	1.98	
76.35	12.75	10.7	73.8	98.9	2000 ppm @ 15 bar	1.26	0.21	J. Membr. Sci. 2020, 604 , 118065. ³⁹
76.28	12.52	11.07	81.6	99.2		1.68	0.20	
72.1	16.8	11.1	38.7 ^a	99.6	2000 ppm @ 15.5 bar	1.60	0.10	J. Membr. Sci. 2019, 578 , 220- 229. ⁴⁰
72.3	17.6	10.1	18.8 ^a	99.4		2.50	0.23	
73	16.7	10.3	28.9 ^a	99.4		1.90	0.18	
75.5	14.2	10.3	52.2 ^a	95.7		10.00	6.96	
73	16.4	10.6	35.6 ^a	99.6		3.80	0.24	
75.5	10.3	14.2	147.8 ^a	99.0	2000 ppm @ 15.5 bar	0.86	0.13	J. Membr. Sci. 2017, 526 , 52-59. ⁴¹
75.3	9.7	15	164.4 ^a	99.0		0.86	0.13	
75.5	12.5	12.1	95.1 ^a	96.7		0.56	0.30	
75.8	12.9	11.3	80.2 ^a	95.7		0.81	0.57	
N. A.	N. A.	N. A.	46.9 ^b	93.6	2000 ppm @ 15.2 bar	4.76	4.95	J. Membr. Sci. 2017, 535 , 248- 257. ⁴²
N. A.	N. A.	N. A.	38.1 ^b	96.9		3.16	1.54	
N. A.	N. A.	N. A.	37.2 ^b	94.1		4.99	4.75	
N. A.	N. A.	N. A.	28.1 ^b	96.2		3.64	2.18	
67.31	17.37	15.32	81.2 ^a	96.2	1000 ppm @ 6 bar	4.03	0.96	J. Taiwan Inst. Chem. Eng. 2017, 80 , 25-33. ⁴³
65.94	18.69	15.36	70.7 ^a	93.6		6.54	2.68	
73.96	17.22	8.82	3.2 ^a	N. A.	2000 ppm @ 15 bar	N. A.	N. A.	J. Membr. Sci. 2017,
75.52	15.6	8.88	17.6 ^a	N. A.		N. A.	N. A.	
74.85	15.4	9.75	32.6 ^a	98.8		3.00	0.55	

73.57	15.89	10.54	39.3 ^a	98.8		4.00	0.73	541, 174-188. ⁴⁴
73.35	15.85	10.8	43.2 ^a	98.8		3.47	0.63	
73.69	15.01	11.3	57.7 ^a	40.0		0.76	17.10	
70.34	16.45	13.18	66.9 ^a	99.4	2000 ppm @ 15 bar	2.84	0.27	J. Membr. Sci. 2017, 541 , 39-52. 45
75.47	13.2	11.33	77.1 ^a	99.3		3.68	0.40	
74.26	13.45	12.29	86.5 ^a	99.4		4.26	0.40	
73.93	14.01	12.06	77.6 ^a	99.4		3.55	0.33	
72.24	15.91	11.85	56.1 ^a	99.3		3.23	0.35	
76.12	14.98	8.91	23.8	94.0	2000 ppm @ 15.5 bar	2.38	2.35	J. Membr. Sci. 2017, 541 , 510-518. ⁴⁶
70.5	17.26	12.25	49.1	96.0		1.84	1.19	
71.8	16.7	11.1	39.6	96.0		2.92	1.89	
75.12	14.84	10.05	42.3	N. A.		N. A.	N. A.	
73.7	12.9	13.4	105.7 ^a	97.5	2000 ppm @ 15.5 bar	2.25	0.89	J. Membr. Sci. 2017, 539 , 441-450. ⁴⁷
73.8	12.5	13.7	113.7 ^a	97.5		2.30	0.91	
73.6	12.9	13.5	106.8 ^a	98.0		2.60	0.82	
74.3	12.8	12.9	101.2 ^a	97.5		2.60	1.03	
74.1	12.4	13.5	112.7 ^a	98.0		3.00	0.95	
73.6	12.8	13.6	109.1 ^a	98.0		2.70	0.85	
75.5	12.5	12.1	95.1 ^a	99.4	2000 ppm @ 15.3 bar	0.56	0.05	J. Membr. Sci. 2017, 527 , 121-128. ⁴⁸
73.8	13.3	12.9	95.4 ^a	96.8		0.84	0.42	
73.54	15.48	10.98	49.0 ^a	99.0	2000 ppm @ 15.5 bar	2.21	0.35	J. Polym. Res. 2016, 24 , 5. ⁴⁹
73.48	15.25	11.27	55.0 ^a	99.0		2.32	0.36	
73.51	15.16	11.33	56.6 ^a	99.0		2.58	0.40	
73.42	14.89	11.69	63.9 ^a	99.0		2.90	0.45	
73.4	14.58	12.02	71.1 ^a	99.0		3.13	0.49	
73.46	14.59	11.59	65.6 ^a	99.0		2.71	0.42	

74.47	12.92	12.61	96.4	94.3	2000 ppm @ 15.2 bar	6.05	5.56	Sci. Rep. 2016, 6 , 22069. ⁵⁰
72.13	14.93	12.94	78.6	97.0		3.49	1.64	
69.61	17.4	12.98	56.4	98.5		0.66	0.15	
71.3	17.09	11.61	42.7	99.0		1.61	0.25	
73.0	14.7	11.6	64.6 ^a	92.0	1500 ppm @ 15.1 bar	0.70	0.92	J. Membr. Sci. 2016, 515 , 79-85. 51
74.0	15.6	10.4	40.0 ^a	98.0	2000 ppm @ 15.5 bar	1.94	0.61	Sep. Purif. Technol. 2010, 75 , 145-155. ⁵²
72.9	15.5	11.6	56.8 ^a	98.0		2.71	0.86	
73.1	15.2	11.7	61.0 ^a	98.0		3.10	0.98	
73.2	15.0	11.8	64.2 ^a	98.0		3.23	1.02	
73.4	14.8	11.8	66.2 ^a	97.0		3.23	1.55	
78.47	11.46	10.07	80.6	98.2	2000 ppm @ 10 bar	6.80	1.25	Angew. Chem., Int. Ed. 2021, 60 , 14636-14643. ⁹
74.27	15.5	10.23	38.6	98.3	2000 ppm @ 15 bar	0.80	0.21	Ind. Eng. Chem. Res. 2020, 59 , 8230-8242. 53
72.46	16.05	11.49	50.3	97.4		1.30	0.52	
72.06	16.53	11.41	45.0	96.7		1.09	0.56	
72.6	15.57	11.83	59.1	97.9		1.64	0.53	
70.33	19.96	9.7	-3.8	88.7	2000 ppm @ 8 bar	2.97	3.02	Sep. Purif. Technol. 2023, 310 , 123122. ⁵⁴
68.28	22.86	8.85	-32.5	93.3		2.57	1.48	
71.06	19.94	9.01	-13.3	94.5		2.42	1.13	
68.86	21.89	9.25	-21.8	96.2		2.12	0.66	
65.91	27.14	6.96	-77.5	97.4		1.89	0.40	

67.53	24.43	8.03	-51.6	97.8		1.77	0.31	
65.61	28.27	6.12	-93.2	97.0		1.82	0.45	
69.18	21.55	9.28	-19.4	97.1		1.73	0.42	
73.64	14.71	11.65	65.2	95.0		2.30	1.21	
74.33	13.71	11.96	79.5	90.0	2000 ppm	2.25	2.50	Desalinatio n 2023, 545 , 116166 ⁵⁵
74.81	13.25	11.93	84.3	94.5	@ 10 bar	2.60	1.51	
74.54	12.92	12.54	95.5	98.7		3.05	0.40	
69.63	19.19	10.91	17.5	97.20		6.20	0.71	
69.09	19.77	10.89	13.1	97.30		8.30	0.92	Environ. Sci. Technol.
68.31	20.20	10.92	10.5	92.30	600 ppm @ 4 bar	11.40	3.80	2021, 55 , 6984-6994 ⁵⁶
75.40	13.60	11.00	68.3 ^a	94.26	2000 ppm @ 10.3 bar	3.85	2.41	Desalinatio n 2011, 274 , 136-143. ⁵⁷
75.50	13.20	11.20	75.4 ^a 60,61	99.20 58,59	20 mM ppm @ 15.5 bar	1.34	N. A.	SWC4
74.30	12.60	13.20	107.0 ^a	96.50		6.04	N. A.	XLE ⁶⁰
74.80	13.10	12.10	88.1 ^a	95.80	10 mM	4.29	N. A.	LE ⁶⁰
74.30	12.80	12.90	101.2 ^a	94.90	ppm @	7.52	N. A.	ESPA3 ⁶⁰
72.90	15.80	11.30	50.2 ^a	91.50	13.8 bar	9.04	N. A.	NE90 ⁶⁰
73.90	14.00	12.10	78.2 ^a	94.40		11.20	N. A.	NF90 ⁶⁰

^a This work did not provide the DNC.

^b This work directly provided the DNC without proportion of C, O, and N.

Tab. S5. Summary of rejection, permeability of Na₂SO₄ and water permeance in NF and DNC of PIP-TMC polyamide membranes in Fig. 4A and B.

C (%)	O (%)	N (%)	DNC (%)	Na ₂ SO ₄ rejection (%)	Test conditions	Water permeance (L m ⁻² h ⁻¹ bar ⁻¹)	Na ₂ SO ₄ permeability coefficient (L m ⁻² h ⁻¹)	Ref.
72.1	16.4	11.5	47.3	99.0	1000 ppm @ 6 bar	7.4	0.45	J. Environ.
72.4	16	11.6	52.2	99.0		13.9	0.84	Chem.
72.9	15.5	11.6	56.8	99.0		16.8	1.02	Eng. 2022,
73.9	14.6	11.5	64.4	99.0		25.2	1.53	10,
73.2	15.3	11.5	57.5	99.0		21.4	1.30	107015. ⁶²
70.98	15.63	13.16	74.3	96.5	1000 ppm @ 6 bar	8.1	1.76	Proc. Natl.
70.46	16.15	13.16	69.4	96.6		26.5	5.60	Acad. Sci. U. S. A. 2021, 118, 118. ⁶³
73.35	15.25	11.4	56.7	99.0	1000 ppm @ 6 bar	12.0	0.73	J. Membr.
71.5	15.49	12.5	68.0	98.0		17.0	2.08	Sci. 2022,
69.3	16.21	13.2	69.3	95.8		20.0	5.26	641, 119887. ⁶⁴
74.45	13.84	11.19	68.2	91.0	1000 ppm @ 8 bar	0.6	0.49	Sep. Purif.
74.28	14.32	11.23	63.7	94.0		0.9	0.45	
73.78	14.19	11.88	73.4	90.0		1.1	1.00	Technol. 2022, 280, 119964. ⁶⁵
71.7	16.15	11.96	55.3	85.0		1.6	2.29	
75.03	12.46	12.51	100.6	78.0		2.3	5.08	
71.4	15.02	13.58	84.9	92.2	500 ppm @ 5 bar	4.9	2.07	J. Membr. Sci. 2021,
68.73	16.42	14.46	81.0	N. A.		N. A.	N. A.	
69.56	15.5	13.4	78.2	91.0		9.6	4.75	

								625, 119154. ⁶⁶
73.87	15.35	10.78	47.5	99.1		9.1	0.41	Sep. Purif.
72.19	17.05	10.76	32.1	98.4	2000 ppm @ 5 bar	21.0	1.70	Technol. 2021, 270, 118802. ⁶⁷
70.36	16.13	12.98	67.5	98.0		26.0	2.12	Macromol.
68.1	16.72	14.79	81.6	98.5		30.0	1.83	Chem.
70.39	16.64	12.66	59.2	93.0	20 mM @ 4	28.0	8.43	Phys.
70.66	15.73	13.11	72.7	92.0	bar	8.0	2.78	2021, 222,
70.38	16.25	12.95	66.1	95.0		10.0	2.11	2100222.
69.22	16.6	13.91	73.5	98.1		24.0	1.86	⁶⁸
70.97	18.15	10.88	24.9	98.0		5.3	0.54	Sep. Purif.
76.42	15.03	8.55	17.6	79.7		9.3	11.84	Technol.
69.21	19.61	11.18	17.9	90.0	1000 ppm @ 5 bar	7.0	3.89	2021, 275,
73.81	16.38	9.81	24.7	93.0		4.8	1.81	119227. ⁶⁹
79.31	13.37	7.16	9.3	84.0		2.5	2.38	
68.51	18.94	10.29	11.2	54.0		42.5	181.02	J. Membr.
69.11	19.08	11.08	20.4	91.0		25.0	12.36	Sci. 2021,
69.46	18.13	12.17	41.0	94.0	1000 ppm @ 5 bar	23.0	7.34	638,
72.26	14.35	11.73	69.9	98.0		20.0	2.04	119699. ⁷⁰
69.73	15.09	14.32	92.1	92.0		19.5	8.48	
72.86	16.09	11.05	44.3	98.1		46.6	1.81	J. Membr.
71.19	17.62	11.2	33.2	92.0	1000 ppm @ 2 bar	55.0	9.57	Sci. 2021, 634, 119450. ⁷¹
69.65	18.19	12.17	40.5	97.2	2000 ppm	7.5	2.15	Desalinati
70.21	16.73	13.06	63.0	99.2	@ 10 bar	13.4	1.08	on 2021,

69.06	19.71	11.23	17.8	97.8		9.2	2.07	512,	
69.05	18.99	11.95	31.7	98.4		15.4	2.50	115118. ⁷²	
70.01	16.93	13.06	61.3	98.8	1000 ppm @ 10 bar	4.8	0.58	J. Mater.	
73.14	14.87	11.99	67.8	97.0		7.0	2.16	Chem. A	
66.53	18.51	14.96	68.2	96.0		10.0	4.17	2021, 9,	
70.92	16.08	13	68.2	98.7		14.5	1.91	26159-	
67.35	17.75	14.9	73.8	94.7		20.6	11.53	26171. ⁷³	
77.42	14.46	8.12	15.8 ^a	95.0	1000 ppm @ 8 bar	3.0	1.26	Ind. Eng.	
80.38	17.46	2.15	-134.2 _a	34.0		21.0	326.12	Chem.	
77.63	16.24	6.13	-35.6 ^a	95.0		9.3	3.92	Res. 2021,	
78.22	13.87	7.91	17.9 ^a	97.0		7.0	1.73	60, 9167-	
76.64	13.67	9.69	48.9 ^a	98.0		5.0	0.82	9178. ⁷⁴	
71.8	17.44	10.76	28.9	95.0	1000 ppm @ 4 bar	17.0	3.58	Membrane s 2021, 11,	
73.75	15.86	10.47	38.6	97.0		23.0	2.85		435. ⁷⁵
74.48	14.85	10.26	45.2	96.0		25.5	4.25		
73.41	15.49	10.54	43.0	93.5		26.0	7.23		
71.48	16.04	11.52	50.8	97.5		30.5	3.13		
53.88	28.43	17.69	30.1 ^a	97.6	N. A.	38.2	N. A.	Compos. Commun. 2021, 24, 100695. ⁷⁶	
73.82	14.7	11.48	63.1	99.0	1000 ppm @ 4 bar	6.8	0.27	J. Membr. Sci. 2021, 617, 118645. ⁷⁷	
73.45	14.91	11.64	63.1	97.0		14.2	1.75		
73.12	15.14	11.74	62.1	96.0		17.0	2.83		
72.9	15.42	11.68	58.6	95.5		21.2	4.00		
71.24	16.78	11.88	48.7	86.0		28.7	18.67		
70.3	19.89	9.82	-1.7 ^a	92.0		26.0	10.87		

70.29	19.12	10.59	13.9 ^a	95.5	1000 ppm @ 4.8 bar	41.7	9.42	ACS Appl. Mater. Interfaces 2021, 13 , 23142-23152. ⁷⁸
73.61	15.18	11.21	54.9	94.0	2000 ppm @ 10 bar	6.8	4.34	Sep. Purif. Technol. 2021, 264 , 118391. ⁷⁹
71.74	16.37	11.89	52.4	98.5		13.0	1.98	
70.5	16.67	12.83	60.9	98.5		17.0	2.59	
72.82	14.9	12.28	71.1	98.5		16.5	2.51	
74.69	13.69	11.62	75.5	98.5		15.0	2.28	
87.18	6.93	5.69	70.5	91.3	2000 ppm @ 5 bar	6.5	3.10	J. Membr. Sci. 2021, 635 , 119523. ⁸⁰
77.49	12.29	9.86	67.1	96.4		8.5	1.59	
73.95	13.8	12.25	82.1	99.6	N. A.	22.0	N. A.	J. Ind. Eng. Chem. 2021, 103 , 373-380. ⁸¹
73.82	13.8	12.38	83.7	99.6		48.0	N. A.	
73.73	13.9	12.37	82.5	99.6		35.0	N. A.	
74.05	13.7	12.25	83.2	99.6		32.0	N. A.	
69.8	15.2	14.5	92.9	87.6	1000 ppm @ 2 bar	50.2	14.26	Chem. Res. Chin. Univ. 2021, 37 , 1101-1109. ⁸²
66.82	17.67	10.58	24.7 ^a	99.5		27.0	0.81	

56.37	20.52	8.91	-18.3 ^a	87.5	2000 ppm @ 6 bar	36.5	31.29	ACS Appl. Mater. Interfaces 2020, 12 , 25304-25315. ²⁷
76.77	13.79	9.43	43.7	90.8	1000 ppm @ 6 bar	26.6	16.17	J. Membr. Sci. 2020, 593 , 117444. ⁸³
74.5	15.83	9.67	27.5	96.0		25.0	6.25	
74.6	16.2	9.2	17.3	95.3		28.2	8.34	
75.08	16.08	8.84	12.8	95.6		32.3	8.93	
70.26	20.24	9.5	-8.3 ^a	N. A.	N. A.	N. A.	N. A.	J. Mater. Chem. A 2020, 8 , 3238-3245. ⁸⁴
68.74	21.76	9.5	-17.7 ^a	N. A.		N. A.	N. A.	
70.04	21.07	8.89	-22.0 ^a	94.0		50.0	19.15	
69.06	21.92	9.01	-25.2 ^a	91.0		55.0	32.64	
70.51	17.34	12.15	47.2 ^a	N. A.	1000 ppm @ 6 bar	N. A.	N. A.	Membranes 2020, 10 , 12. ⁸⁵
67.72	21.68	10.6	-3.0 ^a	97.5		14.5	2.23	
67.62	21.47	10.91	2.2 ^a	97.8		15.8	2.14	
68.38	19.82	11.8	23.9 ^a	98.2		20.0	2.20	
69.01	19.42	11.57	24.0 ^a	97.4		15.0	2.40	
68.81	19.07	12.12	33.2 ^a	98.2		18.3	2.02	
69.65	17.15	13.19	46.9 ^a	98.4		23.3	2.28	
71.25	15.69	13.06	38.1	98.0	2000 ppm @ 6 bar	16.5	2.02	Desalination 2020, 491 , 114499. ⁸⁶
71.24	15.52	13.24	37.2	98.0		16.0	1.96	
71.33	15.38	13.29	28.1	97.0		15.0	2.78	
71.28	14.78	13.94	91.2	90.0		11.0	7.33	
71.01	14.7	14.29	95.8	31.0		24.0	320.52	
73	16	11	44.4	99.0		8.0	0.81	

78.6	12	9.4	63.6	99.0	2000 ppm @ 10 bar	24.0	2.42	Desalinati on 2020, 488 , 114525. ⁸⁷
71.65	16.38	11.97	53.3	98.4	2000 ppm @ 10 bar	8.4	1.37	Desalinati on 2020, 496 , 114340. ⁸⁸
71.93	15.95	12.12	59.1	99.2		9.3	0.75	
71.9	15.72	12.38	64.3	99.5		11.7	0.59	
71.89	15.47	12.64	69.8	99.5		13.5	0.68	
72.07	14.96	12.97	78.6	99.2		16.8	1.35	
72.8	15.11	13.01	77.6	99.0		16.9	1.71	
78.2	11.8	10	75.2 ^a	79.0	1000 ppm @ 7.6 bar	3.3	6.65	Sep. Purif. Technol. 2020, 230 , 11585. ⁸⁹
74.8	13.8	11.4	71.4 ^a	80.0		5.3	10.00	
75.5	13.9	10.6	59.6 ^a	95.0		6.1	2.42	
76.2	12.7	11.1	79.8 ^a	96.0		7.5	2.38	
73.81	14.39	11.8	70.3	97.0	1000 ppm @ 4 bar	13.4	1.66	J. Mater. Chem. A 2020, 8 , 25028. ⁹⁰
76.54	13.07	10.49	67.1	98.0		24.0	1.96	
70.22	17.56	11.71	40.0	97.0		27.6	3.41	
73.32	16.15	10.33	34.1	93.9		48.9	12.71	
78.28	11.75	9.97	75.4 ^a	96.5	1000 ppm @ 6 bar	40.0	8.70	ACS Nano 2019, 13 , 5278- 5290. ⁹¹
72.45	17.45	10.1	20.0 ^a	96.0	2000 ppm @ 6 bar	6.3	1.58	J. Membr. Sci. 2018, 550 , 36- 44. ⁹²
74.8	14.51	10.69	54.5 ^a	93.0		1.7	0.75	
70.93	16.92	12.15	50.8	94.0	1000 ppm	3.8	1.47	Langmuir 2017, 33 ,
70.92	15.68	13.4	76.5	97.0	@ 6 bar	9.3	1.73	

								2318- 2324. ⁹³
73.95	15.34	10.33	41.4	90.0	1000 ppm @ 4 bar	10.4	4.62	Chem.
72.18	15.58	11.55	55.4	94.0		14.5	3.70	Eng. J.
70.68	15.98	12.32	61.2	97.0		21.5	2.66	2021, 416 ,
70.02	16.12	12.78	65.3	96.5		23.9	3.47	129154. ⁹⁴
71.35	15.86	12.79	67.9	99.7	1000 ppm @ 6 bar	13.8	0.29	J. Membr. Sci. 2021, 627 , 119142. ⁹⁵
71.85	15.7	12.45	65.4	99.6		14.9	0.40	
71.24	16.48	12.29	56.3	99.5		17.5	0.57	
70.54	17.79	11.66	37.6	99.4		20.9	0.76	
71.87	17.05	11.08	36.3	97.8		24.5	3.27	
71.55	14.54	13.91	93.4	99.6	1000 ppm @ 6 bar	20.0	0.48	J. Membr.
72.63	14.82	12.82	78.3	98.8		17.5	1.28	Sci. 2020,
71.71	14.71	13.59	88.1	98.0		20.0	2.45	616 , 118557. ⁹⁶
75.26	21.16	3.59	-113.0 a	98.0	1000 ppm @ 6 bar	7.5	0.92	J. Membr.
75.08	19.17	5.75	-61.6 ^a	97.6		17.5	2.58	Sci. 2016,
75.94	17.69	6.37	-41.1 ^a	94.2		12.7	4.68	515 , 238-
76.46	17.04	6.5	-34.3 ^a	95.8		10.5	2.76	244. ⁹⁷
76.63	17.65	6.72	-34.6 ^a	94.8		8.7	2.85	
73.12	16.65	10.23	28.3 ^a	N. A.	1000 ppm @ 6 bar	N. A.	N. A.	J. Mater.
70.54	20.23	9.23	-12.0 ^a	95.2		34.0	10.29	Chem. A
71.72	18.14	10.14	15.1 ^a	97.0		18.0	3.34	2017, 5 ,
74.74	15.4	9.85	34.1 ^a	98.0		10.0	1.22	16289- 16295 ⁹⁸
67.35	18.45	9.84	8.7 ^a	98.6	2000 ppm @ 2 bar	45.5	1.29	J. Membr. Sci. 2021,

								618, 118738. ⁹⁹
71.20	16.40	12.50	59.5 ⁶⁰	95.20 ¹⁰⁰	2000 ppm @ 4 bar	9.43	1.90	NF 270

a: This work did not provide the DNC.

Comment 5: In this technique, as in many others, it is necessary to fit the signal to quantify the abundance of an atom in a certain chemical environment. This fitting process has many more variables than the exact position in binding energy (BE) of the peaks. The signal used to correct the charge effect does not have a great influence on the concentration of certain type of atom if one works within each signal (N1s, O1s or C1s) with relative displacements between the different chemical environments of the atom. The binding energy (or BE displacements) of the different bands in a peaks is what allows chemical states to be identified by XPS. This is the fundamentals of this analysis technique. What is important is not the exact BE values, but the displacements of some bands with respect to others within the complete peak. There is no other way to quantify the N1 and N2 number of atoms by XPS. Therefore, the discussion between lines 68 and 73 of the Introduction is absolutely incorrect.

Response: We thank the reviewer for the recognition and valuable suggestions. We agree with the reviewer that the peaking fitting process of XPS data are crucial for obtaining accurate N1 and N2 data. What plays a crucial role is the BE shift and relative position of atoms in different chemical environments. Using which elements for peak shift itself does not bring errors in XPS peak fitting results. In our work, we want to emphasize that: In the peak fitting process of polyamide, the error occurs when researchers use recommended peak position directly without considering the reason of conducting charge shift.

Using C1s BE for charge shift is a classic criterion for processing XPS data of metallic or non-metallic materials, which is based on prerequisite: “This reference energy is based on the assumption that the carbon is in the form of a hydrocarbon or graphite and that other carbon

species either are not present or can be distinguished from this peak.” (ISO 19318:2004 and ASTM E1523-15)

But for polymers, this prerequisite does not exist. The chemical environment of C in polyamide is complex, and the peaks in XPS are very wide. The highest position of C peak is usually the superposition of the peak positions of skeleton C (C1), oxygen-containing C (C2) and N-related C atoms (C3), which is difficult to confirm as the commonly used BE for calibration (284.8-285.0 eV). PIP-TMC polyamide is one of example (Fig. R2-3). On the other hand, XPS is difficult to avoid the interference of adventitious carbon, which has a wide range due to its poor electrical contact with the sample and will interfere with the peak shape of C. It is challenging to determine the extent to which the C 1s spectrum is affected by the presence of adventitious carbon. Therefore, conducting charge shift with C is likely to result in the peak of N not appearing in the theoretical BE of amide bond N. In this case, it is easy to obtain inaccurate results by directly using the theoretical N peak position for peak fitting, considering the current research's insufficient attention to XPS. As a comparison, we utilized the N element (specifically, BE of N2) for the charge shift of the full-spectrum. The N peak in polyamide is mainly contributed by the N atom of the amide bond, which minimally affected and interfered by external impurities, making its N2 peak the most easily identifiable. This step is beneficial for subsequent data analysis and peak fitting process, although it does not significantly improve the accuracy of the analysis.

Nonetheless, accurate peak positions play a crucial role in peak fitting process in our method as commented by the reviewer. Our previous discussions in lines 68~73 have been changed to:

“This approach, however, is not ideally compatible with carbon-based organic polymer materials with complex chemical environments ^{27,28}, which further undermines the credibility of DNC as a reliable parameter for characterizing polyamide networks. On the one hand, the chemical environment of C in polyamide is complex, and the peaks in XPS are very wide. The highest position of C peak is usually the superposition of the peak positions of skeleton C and oxygen-containing C, which is difficult to confirm as the C1s BE commonly used for calibration (284.8-285.0 eV). On the other hand, XPS is difficult to avoid the interference of exogenous C,

which has a wide range due to its poor electrical contact with the sample and will interfere with the peak shape of C. It is challenging to determine the extent to which the C 1s spectrum is affected by the presence of adventitious carbon. Therefore, conducting charge shift with C is likely to result in the peak of N and O not appearing in the theoretical BE. In this case, it is easy to obtain inaccurate peak fitting results by directly using the theoretical peak position for peak fitting.”

Following the reviewer’s suggestion, we have optimized XPS data process with more specific relative peak positions of atoms by referring to relevant work (*J. Colloid Interface Sci.*, 2002, 247: 149-158; *Science*, 2015, 348, 1347-1351; *Environ. Sci. Technol.*, 2012, 46: 852-859). Tab. S1 and Tab. S2 were added in the revised Supporting Information, which shown the recommended peak positions for XPS peak fitting of MPD-TMC polyamide and PIP-TMC polyamide:

Fig. R2-3. Recommended peak fitting result of C in example PIP-TMC polyamide. In PIP-TMC polyamide, the highest peak of C is contributed by three types of C (skeleton C (C1), oxygen-containing C (C2) and N-related C atoms (C3)). The polyamide example were synthesized according to the following conditions: [PIP] = 1.0 g L⁻¹, [TMC] = 1.5 g L⁻¹, reaction time = 120 s. XPS test measured through Thermo Scientific K-Alpha+ at Al K α line (1486.6 eV, 15 mA \times 15 kV), the vacuum level is 5 \times 10⁻⁹ mbar, X-ray beam spot is 300 μ m, total X-ray irradiation time is 154.1 s (68 s for full spectrum, 28.7 s for C, 30.2 s for O, and 27.2 s for N).

Tab. S1. Classification and the number of various atoms of α , β , and γ structure cells in MPD-TMC polyamide and recommended peak positions for XPS peak fitting of MPD-TMC polyamide ¹²⁻¹⁴.

Type	Group	MPD-TMC polyamide			
		XPS peak position	α	β	γ
C1	-C- C OOH	288.8 \pm 0.1 eV	1	0	0
C2	-C- C ONH-	288.3 \pm 0.2 eV	0	1	0
C3	- C -NH ₂ , -CONH- C -,	286.3 \pm 0.2 eV	0	1	1
C4	- C -COOH, - C -CONH-	285.7 \pm 0.2 eV	1	1	0
C5	- C -H, - C - C , - C = C	284.8 \pm 0.2 eV	1	3	2
O1	- C OOH	533.1 \pm 0.3 eV	2	0	0
O2	- C ONH-	531.6 \pm 0.4 eV	0	1	0
N1	- N H ₂	398.5 \pm 0.2 eV	0	0	1
N3	- N H ₃ ⁺	401.7 \pm 0.2 eV	0	0	1
N2	-CONH- N	400.0 eV	0	1	0

Tab. S2. Classification and the number of various atoms of α , β , and γ structure cells in PIP-TMC polyamide and recommended peak positions for XPS peak fitting of PIP-TMC polyamide ¹²⁻¹⁴.

Type	Group	PIP-TMC polyamide			
		XPS peak position	α	β	γ
C1	-C- C OOH	288.8 \pm 0.1 eV	1	0	0
C2	-C- C ONR-	288.3 \pm 0.2 eV	0	1	0
C3	- C -NH-, -CONR- C -,	286.3 \pm 0.2 eV	0	2	2
C4	- C -COOH, - C -CONR-	285.7 \pm 0.2 eV	1	1	0
C5	- C -H, - C - C , - C = C	284.8 \pm 0.2 eV	1	1	0
O1	- C OOH	533.1 \pm 0.3 eV	2	0	0
O2	- C ONR-	531.6 \pm 0.4 eV	0	1	0

N1	- N H-	398.5 ± 0.2 eV	0	0	1
N3	- N H ₂ ⁺ -	401.7 ± 0.2 eV			
N2	-CON R -	400.0 eV	0	1	0

Comment 6: The authors do not explain how they have worked. The only XPS spectrum they show is the N1s signal in Fig 2, where they observe two types of nitrogen atoms (N1 which I guess is the area highlighted in green, and N2 which apparently is the blue peak), and both bands are at the same binding energy. In this case, the band N1 may be due entirely to uncertainty in the background type that has been subtracted from the original spectrum. The authors have to give binding energies of the different species of atoms that they want to quantify to obtain the HABD parameter. Tab. S1 only shows the binding energies of N2 and O2 bands. What displacement did they use to get respectively N1 and O1? What width have they assume for the N2 and O2 bands? Have they used Gaussian or Lorentzian bands? Without answering these questions, the N1/N2 and O1/O2 ratios necessary to calculate the HABD index are not credible.

Response: We thank the reviewer for the recognition and valuable suggestions. In our original N peak fitting diagram, N2 is the blue peak and N1 is the remaining area with green (including amino N and protonated amino N), without defining the peak. Our original intention was that N1 does not need to appear as a peak in the N peak fitting result, only the proportion of N1 to N2 are needed to calculate HABD. We agree with the reviewer that this expression is misleading, and we have recounted the our XPS data process in the revised manuscript. The main points are as follows:

We recommend to follow the following process for XPS data analysis and peak fitting with MPD-TMC polyamide as an example (Fig. R2-4). Here, the amino N and protonated amino N are further classified as N1 and N3 for the convenience of peak fitting during XPS data process. The XPS data for the example here is obtained by recommended testing process mentioned in our response for Comment 4-2.

Fig. R2-4 Flowchart of XPS data analysis process.

Firstly, the charge shift is still based on the BE of N (N2). The N peak in polyamide is mainly contributed by the N atom of the amide bond, which minimally affected and interfered by external impurities, making its N2 peak the most easily identifiable. This step beneficial for subsequent data analysis and peak position fitting, although itself does not significantly improve the accuracy of the analysis.

Secondly, N peak are divided into three peaks based on the relative peak positions of N1, N2 and N3 given in the related researches or handbooks in Tab. R2-1 (*J. Colloid Interface Sci.*, 2002, 247: 149-158; *Science*, 2015, 348, 1347-1351; *Environ. Sci. Technol.*, 2012, 46: 852-859). Peak fitting should be carried out under Gaussian–Lorentzian product pseudo-Voigt peak shape, with a Gaussian component comprising 60% ~ 80% of the peak, as Gaussian components are commonly observed in XPS peaks of polymers (*J. Vac. Sci. Technol. A*, 2020, 38: 061203; *J.*

Vac. Sci. Technol. A, 2022, 40: 063201). During this process, attention should be paid to the peak positions of the three peaks of N should be constrained within the recommended range, and their FWHM should be basically close to each other.

Thirdly, the peak fitting of C and O is carried out on the basis of strict adherence to the self-consistent relationship (*Appl. Surf. Sci.*, 2016, 387: 294), beyond paying attention to curve shape, peak position, and FWHM. Proportion of O2 and C2 can be obtained based on the self-consistent relationship between the elements of the amide bond or amino in polyamide. A single amide bond should have one C atom, one O atom, and one N atom, thus the quantities of C2, O2, and N2 should be equal. In MPD-TMC, each N atom is connected to a benzene ring C atom, implying the number of C3 should be equal to the total number of N atoms:

$$C2 = O2 = N2 \quad (R2-15)$$

$$C3 = N1 + N2 + N3 \quad (R2-16)$$

The content of C1, C4, C5, and O1, cannot be directly calculated through the peak fitting results of N. However, self-consistent relationships also provide cross-peak self-consistency constraints for them.

If there are no impurities, C and O all come from polyamide. One carboxyl group should have two O1, one C1 and connects to a C4 atom. One amide bond connects to a C4 atom as well, thus:

$$C1 = \frac{O1}{2} \quad (R2-17)$$

$$C4 = \frac{O1}{2} + N2 \quad (R2-18)$$

According to the division and statistics of our structure cells, we can also obtain statistical relationships:

$$C5 = \frac{O1}{2} + 3N2 + 2(N1 + N3) \quad (R2-19)$$

The remaining atoms that cannot be separated should be considered as impurities, rather than forcibly attributed to specific atomic type such as C1, C4, C5 or O1. For example, if the total

number of C after peak fitting exceeds the sum of C1-C5 under these constraint conditions, it indicates the presence of impurity C in the XPS test. Similarly, if the total number of C is insufficient to satisfy the Formula R2-17 to R2-19 after dividing the peak of O, it indicates that some impurities in O have been separated into O1.

Fig. R2-5 and Fig. R2-6 show the peak fitting results of our self-prepared sample as an example. We use N as the standard for correction. N peak is fully attributed to different N atoms within the polyamide. When the peak conditions and cross-peak self-consistency constraints are met, the C peak does not exclusively correspond to O1~O2 and C1~C5 atoms of polyamide. The remaining portions are treated as impurities with unspecific peak positions. The quantities of various atoms (Fig.R2-5) needs to meet the self-consistent requirements of Formula R2-15 to R2-19, which is important in our method.

Fig. R2-5 Recommended peak fitting result of N, O, and C of example MPD-TMC polyamide. The polyamide example was synthesized according to the following conditions: [MPD] = 20.0 g L⁻¹, [TMC] = 1.5 g L⁻¹, reaction time = 120 s. XPS test measured through Thermo Scientific K-Alpha+ at Al K α line (1486.6 eV, 15 mA \times 15 kV), the vacuum level is 5 \times 10⁻⁹ mbar, X-ray beam spot is 300 μ m, total X-ray irradiation time is 154.1 s (68 s for full spectrum, 28.7 s for C, 30.2 s for O, and 27.2 s for N).

Fig. R2-6. Recommended atom proportion result of N, O, and C of example MPD-TMC polyamide. The polyamide example was synthesized according to the following conditions: [MPD] = 20.0 g L⁻¹, [TMC] = 1.5 g L⁻¹, reaction time = 120 s. XPS test measured through Thermo Scientific K-Alpha+ at Al K α line (1486.6 eV, 15 mA \times 15 kV), the vacuum level is 5 \times 10⁻⁹ mbar, X-ray beam spot is 300 μ m, total X-ray irradiation time is 154.1 s (68 s for full spectrum, 28.7 s for C, 30.2 s for O, and 27.2 s for N).

Tab. R2-1 Recommended peak positions for XPS peak fitting of MPD-TMC polyamide (*J. Colloid Interface Sci.*, 2002, 247: 149-158; *Science*, 2015, 348, 1347-1351; *Environ. Sci. Technol.*, 2012, 46: 852-859).

Type	Group	MPD-TMC polyamide			
		XPS peak position	α	β	γ
C1	-C-COOH	288.8 \pm 0.1 eV	1	0	0
C2	-C-CONH-	288.3 \pm 0.2 eV	0	1	0
C3	-C-NH ₂ , -CONH-C-	286.3 \pm 0.2 eV	0	1	1
C4	-C-COOH, -C-CONH-	285.7 \pm 0.2 eV	1	1	0
C5	-C-H, -C-C, -C=C	284.8 \pm 0.2 eV	1	3	2
O1	-COOH	533.1 \pm 0.3 eV	2	0	0
O2	-CONH-	531.6 \pm 0.4 eV	0	1	0
N1	-NH ₂	398.5 \pm 0.2 eV			
N3	-NH ₃ ⁺	401.7 \pm 0.2 eV	0	0	1

In our peak fitting process, the peak fitting of C and O is to obtain O2 more accurately. The most crucial step in the entire process is the peak fitting of N. The reason for choosing N for such a critical step is also due to the fact that N impurities are not common in XPS testing. Therefore, if the testing is handled properly, all N should be attributed to polyamide. This is also why we refer to this method as an N-based processing method. The proposed XPS data analysis method has been detailed in SI and will be briefly described in the main text. Any characterization method that can help obtain O2 can also become a constraint during XPS peak fitting process.

With the hope of improving credibility and accuracy obtained HABD through XPS data, we have added these contents in the revised manuscript on page 9 to 11. Fig. R2-4 was added in the Supporting Information marked as Fig. S3. Fig. R2-5 was added in the Fig. 2 in revised manuscript. Fig. R2-6 was added in the Supporting Information marked as Fig. S4. Tab. R2-1 was added revised Supporting Information marked as Tab. S1. Tab. S2 was added in the Supporting Information, showing recommended peak positions for XPS peak fitting of PIP-TMC polyamide.

After optimizing the XPS processing method, the obtained HABD data in Fig. 3 and Fig. 4 has changed accordingly. But the superiority of HABD in correlation performance remains unchanged.

Tab. S2. Classification and the number of various atoms of α , β , and γ structure cells in PIP-TMC polyamide and recommended peak positions for XPS peak fitting of PIP-TMC polyamide

12-14.

Type	Group	PIP-TMC polyamide			
		XPS peak position	α	β	γ
C1	-C-COOH	288.8 ± 0.1 eV	1	0	0
C2	-C-CONR-	288.3 ± 0.2 eV	0	1	0

C3	$\underline{\text{C}}\text{-NH-}$, $\text{-CONR-}\underline{\text{C}}\text{-}$,	$286.3 \pm 0.2 \text{ eV}$	0	2	2
C4	$\underline{\text{C}}\text{-COOH}$, $\underline{\text{C}}\text{-CONR-}$	$285.7 \pm 0.2 \text{ eV}$	1	1	0
C5	$\underline{\text{C}}\text{-H}$, $\underline{\text{C}}\text{-C}$, $\underline{\text{C}}\text{=C}$	$284.8 \pm 0.2 \text{ eV}$	1	1	0
O1	-COOH	$533.1 \pm 0.3 \text{ eV}$	2	0	0
O2	-CONR-	$531.6 \pm 0.4 \text{ eV}$	0	1	0
N1	$\underline{\text{N}}\text{H-}$	$398.5 \pm 0.2 \text{ eV}$	0	0	1
N3	$\underline{\text{N}}\text{H}_2^+\text{-}$	$401.7 \pm 0.2 \text{ eV}$	0	0	1
N2	-CONR-	400.0 eV	0	1	0

Fig. 2. Graphical representation of analyzing the polyamide structure and calculating HABD using MPD-TMC polyamide as an example. The polyamide example were

synthesized according to the following conditions: [MPD] = 20.0 g L⁻¹, [TMC] = 1.5 g L⁻¹, reaction time = 120 s. XPS test measured through Thermo Scientific K-Alpha+ at Al K α line (1486.6 eV, 15 mA \times 15 kV), the vacuum level is 5 \times 10⁻⁹ mbar, X-ray beam spot is 300 μ m, total X-ray irradiation time is 154.1 s (68 s for full spectrum, 28.7 s for C, 30.2 s for O, and 27.2 s for N). Peak fitting was based on the recommended peak position in Tab. S1^{11,34,35}.

Fig. 3. The correlation analysis between the performance of reverse osmosis membranes with the DNC or HABD of MPD-TMC polyamides. The DNC parameter does not show a reliable correlation with the NaCl rejection (A) and water permeance (B) of reverse osmosis polyamide membranes. In contrast, HABD demonstrates a reliable correlation with the NaCl rejection (C) and water permeance (D) of those membranes (The detailed statistical data in this figure is shown in Tab. S3 and S4 in SI).

Fig. 4. The correlation analysis between the performance of nanofiltration membranes with the DNC or HABD of PIP-TMC polyamides. DNC cannot form a reliable correlation with the Na₂SO₄ rejection (A) and water permeance (B) of nanofiltration polyamide membranes. In contrast, HABD demonstrates a reliable correlation with the Na₂SO₄ rejection (C) and water permeance (D) of those membranes. (The detailed statistical data in this figure is shown in Tab. S5 and S6 in SI).

Comment 7: All the above does not detract from the validity of the method they propose to estimate a parameter related to the structure of the polyamide. I only point out that the manuscript must show or give indications that the XPS measurements have been worked on and interpreted correctly, and that is to be done in this work.

Although I consider the model of the polyamide structure is good, the derivation of the HABD from the XPS analysis is not sufficiently explained, and there are indications in the manuscript

that it has not been done correctly. For these reasons, I recommend that the manuscript be rejected until all of these issues are properly addressed.

Response: We thank the reviewer for the recognition and valuable suggestions. The comments of XPS methods are very helpful for us to improve the reliability of our manuscript.

We had provided a detailed explanation about how to obtain XPS data results from recent research, how to analyze XPS data, and how to calculate HABD using XPS data. Overall, we develop structure parameters called HABD, HCD and HAD respectively, which comprehensively consider the characteristic structure of polyamide crosslinked network. To accurately obtain the values of these parameters, we provide an optimized XPS analysis method for polyamides, improving the current lack of unified standards and strict treatment in XPS testing. We hope that all the response will address the issues in our previous XPS analysis and provide researchers with powerful tools for analyzing the structure-performance relationship of polyamide membranes.

Response to the comments by Reviewer 3

Comment 1: “Moreover, the calculation of DNC usually relies on X-ray photoelectron spectroscopy (XPS) data processing^{28,29}, which employs the charge shift of the full spectrum, theoretical binding energy, and peak fitting primarily based on the 1s peak of adventitious carbon^{11,30}. This approach, however, is not ideally compatible with carbon-based organic polymer materials with complex chemical environments^{31,32}, which further undermines the credibility of DNC as a reliable parameter for characterizing polyamide networks.”

While I sympathize with all these comments I think that Authors should be more concrete in defining the main problem which is the lack of reliable charge reference in XPS studies of insulating materials. The commonly used C 1s method based on the adventitious carbon has been shown to suffer from many issues which make it unreliable.). Adventitious carbon is in general an unknown compound, not an inherent part of the sample, does not make proper electrical contact to the analyzed sample, BE of the C 1s peak varies in a wide range. When it comes to samples which contain C (as is the case for materials considered in this paper) the situation is potentially better as one can directly refer to the 1s level of the specific chemical group. However, the main concern is – how much is the C 1s spectrum affected by the presence of adventitious carbon? The latter is present on all surfaces exposed to air which obviously adds to the confusion. Greater care during sample handling and minimized exposure time could be considered for more reliable results. I would suggest extending the discussion along these lines to better motivate the need for an alternative approach.

Response: We thank the reviewer for the recognition and valuable suggestions. Following the reviewer’s suggestion, we have highlighted the influence of unstable C 1s on XPS data analysis and our countermeasures in the revised manuscript on page 4.

On page 4:

“This approach, however, is not ideally compatible with carbon-based organic polymer materials with complex chemical environments^{27,28}, which further undermines the credibility of DNC as a reliable parameter for characterizing polyamide networks. On the one hand, the chemical environment of C in polyamide is complex, and the peaks in XPS are very wide. The

highest position of C peak is usually the superposition of the peak positions of skeleton C and oxygen-containing C, which is difficult to confirm as the C 1s BE commonly used for calibration (284.8-285.0 eV). On the other hand, XPS is difficult to avoid the interference of exogenous C, which has a wide range due to its poor electrical contact with the sample and will interfere with the peak shape of C. It is challenging to determine the extent to which the C 1s spectrum is affected by the presence of adventitious carbon. Therefore, conducting charge shift with C is likely to result in the peak of N and O not appearing in the theoretical BE. In this case, it is easy to obtain inaccurate peak fitting results by directly using the theoretical peak position for peak fitting.”

When it comes to the sample preparation of XPS, we referred to the standards of International Organization for Standardization (ISO 20579), American Society for Testing Materials (E1078), and National Standard of the People's Republic of China (GB/T SJT10458-1993) and related researches (*J. Vac. Sci. Technol. A*, 2019, 37: 031401; *J. Vac. Sci. Technol. A*, 2020, 38, 061203; *J. Vac. Sci. Technol. A*, 2020, 38: 063202) to provide a series of points about sample preparation that should be paid attention to when analyzing the surface of polyamides. We look forward to your suggestions to improve the accuracy of these points for readers of our work to better conduct XPS testing. We have summarized the key points of XPS testing in Section 5.2 in the revised SI.

In Section 5.2, SI:

“Sample preparation: For the conventional polyamide thin-film composite membranes, cleaning the solvents and unreacted monomers is crucial. We suggest choosing to soak and clean with ethanol for more than 24 hours. For the free-standing polyamide nanofilms developed in recent years, it can be considered to use silicon wafers or gold-plated silicon wafers to load samples for testing to avoid interference from foreign elements. After drying, the polyamide nanofilm can firmly adhere to the surface of the silicon wafer and should also be cleaned before testing.

Storage time: Minimize the storage time of test samples as much as possible. If storage is necessary, water can be considered to seal the samples, but drying should be carried out before

testing. If possible, it is recommended to use rare gas to protect the sample. In our method, it is not recommended to protect the sample with N₂, because the subsequent analysis mainly uses N as the calibration.”

In addition, the samples used for XPS peak fitting example in our work were prepared and tested under the above conditions. The sample was stored for only 24 hours before XPS test and sealed with ionized water. XPS test measured through Thermo Scientific K-Alpha+ at Al K α line (1486.6 eV, 15 mA \times 15 kV), the vacuum level is 5×10^{-9} mbar, X-ray beam spot is 300 μ m, total X-ray irradiation time is 154.1 s (68 s for full spectrum, 28.7 s for C, 30.2 s for O, and 27.2 s for N).

Comment 2: For the specific class of materials addressed by the proposed HABD method a very critical experimental variable, often completely neglected, is the X-ray exposure time. It is well known that such materials are prone to x-ray damage, which directly affects the bonding and, hence, the degree of crosslinking. This point is not mentioned at all in the paper.

Response: We thank the reviewer for the recognition and valuable suggestions. The XPS data involved in our work mainly consists of two aspects. One is the XPS data involved in our correlation analysis, which are obtained from recent works directly. We have listed the X-ray exposure time and other related XPS testing conditions of referenced researches in Tab. S10 and Tab. S11 the revised SI. Other is the samples used for XPS peak fitting examples, which were prepared and tested under the above conditions. XPS test measured through Thermo Scientific K-Alpha+ at Al K α line (1486.6 eV, 15 mA \times 15 kV), the vacuum level is 5×10^{-9} mbar, X-ray beam spot is 300 μ m, total X-ray irradiation time is 154.1 s (68 s for full spectrum, 28.7 s for C, 30.2 s for O, and 27.2 s for N).

Recent researches generally lack descriptions of XPS testing conditions, especially vacuum level and area of analysis during XPS testing. In order to improve the standardization of XPS testing and the accuracy of the obtained peak fitting results and HABD, we referred to the standards of International Organization for Standardization (ISO 20579), American Society for Testing Materials (E1078), and National Standard of the People's Republic of China (GB/T

SJT10458-1993) and related researches (*J. Vac. Sci. Technol. A*, 2019, 37: 031401; *J. Vac. Sci. Technol. A*, 2020, 38, 061203; *J. Vac. Sci. Technol. A*, 2020, 38: 063202) to provide a series of points about testing conditions that should be paid attention to when analyzing the surface of polyamides. We look forward to your suggestions to improve the accuracy of these points for readers of our work to better conduct XPS testing. We have summarized the key points of XPS testing in the Section 5.2 in revised SI.

In Section 5.2, SI:

“Vacuum level: The vacuum level during XPS test should be below than 10^{-7} mbar.

X-ray beam spots: We recommend using larger X-ray beam spots (more than 200 μm) to observe polyamide in order to obtain results that better reflect the overall element proportion of polyamide. Readers can also perform mapping imaging with small X-ray beam spots and take the average of multiple test points on a large scale.

X-ray source: A stronger X-ray source is not recommended, as it may cause damage to the C-chain polymer^{107,108}. Mg K α (1253.6 eV with natural linewidth of 0.7 eV) and Al K α (1486.6 eV with natural linewidth of 0.9 eV) can obtain information about the surface of polyamide at around 7~10 nm, which is acceptable. Moreover, the reported polyamide nanofilm has become thinner in recent years, and the comprehensiveness of Mg K α or Al K α is acceptable.

X-ray exposure time: X-rays will also cause certain damage to polyamide with exposure time prolong^{56,109}. Considering that polyamides typically require testing full spectrum and fine spectrum of C, N, and O in XPS, the total XPS irradiation time should not exceed 5 min.

We expect researchers to refer to relevant standards and researches before conducting XPS test. Following above key points can help to improve the accuracy of peak fitting result and to assist in the analysis of RO and NF materials along with the proposed HABD.”

Tab. S10. XPS test conditions of RO membranes in Fig. 3.

Sample conditioning	Instruments	Vacuum level	Area of analysis	X-ray type and energy	X-ray exposure time	Data analysis	Ref.
-------------	--------------	------------------	-----------------------	---------------------	---------------	------

TFC membrane	Axis Ultra DLD, Kratos Analytical	N. A.	N. A.	N. A.	N. A.	CasaX PS software	Sep. Purif. Technol. 2022, 281 , 119884. 18
TFC membrane	AXIS Supra	N. A.	300 × 700 μm	Al Kα (1486.6 eV)	N. A.	CasaX PS software	J. Membr. Sci. 2021, 640 , 119805. 19
TFC membrane	Thermo Scientific 49 ESCALAB 250Xi,	N. A.	N. A.	N. A.	N. A.	N. A.	J. Environ. Chem. Eng. 2022, 10 , 106958. 20
TFC membrane, rinsed 3 times with ultrapure water	Kratos AXIS Supra (UK) system	N. A.	N. A.	Al Kα	N. A.	N. A.	J. Membr. Sci. 2021, 633 ,

followed by freeze- drying							119395. 21
TFC membrane	K-alpha	N. A.	N. A.	Al K α	N. A.	N. A.	Chem. Eng. Res. Des. 2021, 165 , 1- 11. ²²
TFC membrane	X-tool spectrometer, ULVAC-PHI	N. A.	N. A.	N. A.	N. A.	N. A.	J. Membr. Sci. 2021, 620 , 118870. 23
TFC membrane	Kratos Inc., AXIS-His	N. A.	N. A.	Al (1486.6eV)	N. A.	N. A.	J. Membr. Sci. 2021, 618 , 118677. 24

TFC membrane	JPS-9010 MC, JEOL	N. A.	N. A.	N. A.	N. A.	N. A.	Desalination 2021, 516 , 115222. 25
TFC membrane	ESCALAB 250 XI	N. A.	N. A.	N. A.	N. A.	N. A.	J. Membr. Sci. 2021, 638 , 119680. 26
TFC membrane	Quanta 200 spectrometer	N. A.	N. A.	N. A.	N. A.	N. A.	ACS Appl. Mater. Interfaces 2020, 12 , 25304- 25315. 27
EDX	N. A.	N. A.	N. A.	N. A.	N. A.	N. A.	J. Membr. Sci. 2019,

							574, 1-9. ²⁸
TFC membrane	VG Multilab 2000, Thermo VG Scientific	N. A.	N. A.	Al K α (1486.6 eV)	N. A.	N. A.	J. Membr. Sci. 2019, 570 , 112-119. ²⁹
TFC membrane	Axis Ultra DLD, Kratos Analytical	N. A.	N. A.	N. A.	N. A.	N. A.	Rsc Adv. 2018, 8 , 25236-25247. ³⁰
TFC membrane	Escalab 250Xi, Thermo Fisher Scientific	N. A.	N. A.	N. A.	N. A.	N. A.	J. Appl. Polym. Sci. 2018, 135 , 46261. ³¹
EDX	N. A.	N. A.	N. A.	N. A.	N. A.	N. A.	Science 2018, 361 , 682-685. ³²

TFC membrane	Thermal Scientific K-Alpha	N. A.	N. A.	N. A.	N. A.	N. A.	Desalination 2020, 480 , 114342. 33
TFC membrane	ULVAC PHI X-tool	N. A.	N. A.	N. A.	N. A.	N. A.	J. Membr. Sci. 2020, 614 , 118449. 34
TFC membrane	Shimadzu, AXIS Ultra DLD	N. A.	N. A.	N. A.	N. A.	N. A.	Desalination 2020, 491 , 114345. 35
TFC membrane	N. A.	6×10^{-10} Torr	N. A.	Al K α (1486.6 eV)	N. A.	N. A.	J. Membr. Sci. 2020, 611 , 118407. 36

TFC membrane	Quanta 200 spectrometer, FEI.	N. A.	N. A.	N. A.	N. A.	N. A.	J. Membr. Sci. 2020, 612 , 118412. 37
TFC membrane	ESCALAB 250Xi	N. A.	N. A.	N. A.	N. A.	N. A.	J. Membr. Sci. 2020, 614 , 118498. 38
TFC membrane	ESCALAB 250 XI	N. A.	N. A.	N. A.	N. A.	N. A.	J. Membr. Sci. 2020, 604 , 118065. 39

TFC membrane	X-tool spectrometer, ULVAC-PHI	N. A.	N. A.	N. A.	N. A.	N. A.	J. Membr. Sci. 2019, 578 , 220-229. ⁴⁰
Self-standing PA nanofilm	PHI-5000	N. A.	N. A.	Al K α (1.49 keV)	N. A.	N. A.	J. Membr. Sci. 2017, 526 , 52-59. ⁴¹
TFC membrane	Kratos AXIS ULTRA, UK	N. A.	N. A.	N. A.	N. A.	N. A.	J. Membr. Sci. 2017, 535 , 248-257. ⁴²
TFC membrane	Thermo escalab 250Xi	N. A.	N. A.	Al K α (1486.6 eV)	N. A.	N. A.	J. Taiwan Inst. Chem. Eng. 2017,

								80, 25-33. ⁴³
TFC membrane	PHI-1600	N. A.	N. A.	N. A.	N. A.	N. A.	N. A.	J. Membr. Sci. 2017, 541 , 174-188. ⁴⁴
TFC membrane	Thermo escalab 250Xi, USA	N. A.	N. A.	N. A.	N. A.	N. A.	N. A.	J. Membr. Sci. 2017, 541 , 39-52. ⁴⁵
TFC membrane	Thermo Electron, K-Alpha	N. A.	N. A.	N. A.	N. A.	N. A.	N. A.	J. Membr. Sci. 2017, 541 , 510-518. ⁴⁶

							2016, 515 , 79-85. 51
TFC membrane, storage in deionized water and completely dried under vacuum before XPS	PHI-1600	N. A.	N. A.	Mg K α	N. A.	PHI- MATL AB softwa re packa ge	Sep. Purif. Technol . 2010, 75 , 145- 155. ⁵²
Self- standing PA nanofilm loaded on gold-plated silicon wafers	Escalab250Xi, ThermoFisher	N. A.	N. A.	Al K α (1486.6 eV)	N. A.	N. A.	Angew. Chem., Int. Ed. 2021, 60 , 14636- 14643. 9

TFC membrane	PHI-1600	N. A.	N. A.	Al K α (1486.6 eV)	N. A.	N. A.	Ind. Eng. Chem. Res. 2020, 59 , 8230-8242. ⁵³
Self-standing PA nanofilm loaded on silicon wafers	Thermo Scientific K-Alpha	N. A.	N. A.	Al K α (1486.6 eV)	N. A.	N. A.	Sep. Purif. Technol. 2023, 310 , 123122. ⁵⁴
Self-standing PA nanofilm loaded on silicon wafers	Thermo Scientific K-Alpha	N. A.	N. A.	Al K α (1486.6 eV)	N. A.	N. A.	Desalination 2023, 545 , 116166. ⁵⁵
TFC membrane	K-Alpha, Thermo Fisher Scientific	N. A.	N. A.	N. A.	N. A.	N. A.	Environ. Sci. Technol. 2021, 55 ,

							6984-6994 ⁵⁶
TFC membrane	ThermoFisher K-alpha	N. A.	N. A.	Al K α (1486.6 eV)	N. A.	N. A.	Desalination 2011, 274, 136-143. ⁵⁷
TFC membranes, extensively rinsed and soaked in MilliQ water for 24 h before dried in a vacuum.	SSI S-Probe Monochromatized XPS Spectrometer	N. A.	250 μm \times 1000 μm	Al K α (1486.6 eV)	N. A.	N. A.	Desalination 2009, 242, 168-182. ⁶⁰

Tab. S11. XPS test conditions of NF membranes in Fig. 4.

Sample conditioning	Instruments	Vacuum level	Area of analysis	X-ray type and energy	X-ray exposure time	Data analysis	Ref.
-------------	--------------	------------------	-----------------------	---------------------	---------------	------

TFC membranes	ESCALAB 250	N. A.	N. A.	N. A.	N. A.	N. A.	J. Environ. Chem. Eng. 2022, 10 , 107015. 62
TFC membranes	Thermal Fisher Scientific ESCALAB 250 Xi)	N. A.	N. A.	N. A.	N. A.	XPSPE AK41 software	Proc. Natl. Acad. Sci. U. S. A. 2021, 118 , 118. 63
TFC membranes	XRF-1800	N. A.	N. A.	N. A.	N. A.	N. A.	J. Membr. Sci. 2022, 641 , 119887. 64
TFC membranes	K-alpha, Thermo Fisher, USA	N. A.	N. A.	N. A.	N. A.	N. A.	Sep. Purif. Technol. 2022, 280 ,

							119964. 65
TFC membranes	K-alpha, Thermo Fisher Scientific	N. A.	N. A.	Al K α (1486.6 eV)	N. A.	N. A.	J. Membr. Sci. 2021, 625 , 119154. 66
TFC membranes	Kratos Analytical-A	N. A.	N. A.	N. A.	N. A.	N. A.	Sep. Purif. Technol. 2021, 270 , 118802. 67
TFC membranes	Thermal Fisher Scientific ESCALAB	N. A.	N. A.	N. A.	N. A.	XPSPE AK software	Macromol. Chem. Phys. 2021, 222 , 2100222 68

TFC membranes	Axis Ultra DLD	N. A.	N. A.	N. A.	N. A.	N. A.	Sep. Purif. Technol. 2021, 275 , 119227. 69
TFC membranes	AXIS UltraDLD	N. A.	N. A.	N. A.	N. A.	N. A.	J. Membr. Sci. 2021, 638 , 119699. 70
N. A.	ESCALAB25 0Xi	N. A.	N. A.	N. A.	N. A.	N. A.	J. Membr. Sci. 2021, 634 , 119450. 71
N. A.	N. A.	N. A.	N. A.	N. A.	N. A.	N. A.	Desalination 2021, 512 , 115118. 72

							2021, 617 , 118645. 77
TFC membranes, samples were placed under the infrared lamp for 1 h.	Thermo Kalpha	N. A.	N. A.	N. A.	N. A.	N. A.	ACS Appl. Mater. Interfaces 2021, 13 , 23142-23152. ⁷⁸
TFC membranes	Quanta200 spectrometer	N. A.	N. A.	N. A.	N. A.	N. A.	Sep. Purif. Technol. 2021, 264 , 118391. 79
TFC membranes	N. A.	N. A.	N. A.	N. A.	N. A.	N. A.	J. Membr. Sci. 2021, 635 , 119523. 80

TFC membranes	Thermo Scientific MultiLab 2000	N. A.	specific spot size of 650 μ m	Al K α (1486.6 eV)	N. A.	N. A.	J. Ind. Eng. Chem. Chem. 2021, 103 , 373-380. 81
TFC membranes	Thermo Fisher S4 Scientific Escalab 250Xi	N. A.	N. A.	Al K α (1486.6 eV)	N. A.	N. A.	Chem. Res. Chin. Univ. 2021, 37 , 1101-1109. ⁸²
TFC membranes	Quanta 200 spectrometer	N. A.	N. A.	N. A.	N. A.	N. A.	ACS Appl. Mater. Interface s 2020, 12 , 25304-25315. ²⁷

TFC membranes	PerkinElmer	N. A.	N. A.	Al K α (1486.6 eV)	N. A.	N. A.	J. Membr. Sci. 2020, 593 , 117444. 83
TFC membranes	AXIS Supra	N. A.	N. A.	Al K α (1486.6 eV)	N. A.	CasaXP S software	J. Mater. Chem. A 2020, 8 , 3238- 3245. ⁸⁴
TFC membranes	Thermo Scientific	N. A.	N. A.	N. A.	N. A.	N. A.	Membranes 2020, 10 , 12. ⁸⁵
TFC membranes	ESCALAB25 0Xi	N. A.	N. A.	N. A.	N. A.	N. A.	Desalination 2020, 491 , 114499. 86

TFC membranes	Quanta 200	N. A.	N. A.	N. A.	N. A.	N. A.	Desalination 2020, 488 , 114525. ⁸⁷
TFC membranes	Quanta 200	N. A.	N. A.	N. A.	N. A.	N. A.	Desalination 2020, 496 , 114340. ⁸⁸
TFC membranes	Kratos AXIS Ultra DLD	N. A.	N. A.	N. A.	N. A.	The Shirley-type background and the Gaussian-Lorentz peak deconvolution were used for fitting the high-resolution	Sep. Purif. Technol. 2020, 230 , 11585. ⁸⁹

						n C 1s spectra	
TFC membranes	K-alpha, Thermo Fisher	N. A.	N. A.	N. A.	N. A.	N. A.	J. Mater. Chem. A 2020, 8 , 25028. ⁹⁰
TFC membranes	Thermo Fisher Scientific ESCALAB 250Xi XPS	N. A.	N. A.	N. A.	N. A.	N. A.	ACS Nano 2019, 13 , 5278-5290. ⁹¹
TFC membranes, freeze dried before test	K-alpha, Thermo Fisher	N. A.	N. A.	N. A.	N. A.	N. A.	J. Membr. Sci. 2018, 550 , 36-44. ⁹²
TFC membranes	PerkinElmer 5300	N. A.	N. A.	Al K α (1486.6 eV)	N. A.	N. A.	Langmuir 2017, 33 , 2318-2324. ⁹³
TFC membranes	K-alpha, Thermo Fisher Scientific	N. A.	N. A.	N. A.	N. A.	N. A.	Chem. Eng. J. 2021, 416 ,

							129154. 94
TFC membranes	ThermoScientific, K-Alpha+	N. A.	N. A.	Al K α (1486.6 eV)	N. A.	N. A.	J. Membr. Sci. 2021, 627 , 119142. 95
TFC membranes	ThermoScientific, K-Alpha+	N. A.	N. A.	Al K α (1486.6 eV)	N. A.	N. A.	J. Membr. Sci. 2020, 616 , 118557. 96
TFC membranes	PerkinElmer	N. A.	N. A.	Al K α (1486.6 eV)	N. A.	N. A.	J. Membr. Sci. 2016, 515 , 238-244. 97
TFC membranes	PerkinElmer	N. A.	N. A.	Al K α (1486.6 eV)	N. A.	N. A.	J. Mater. Chem. A 2017, 5 , 16289- 16295 ⁹⁸

TFC membranes	Thermo-Fisher	N. A.	N. A.	N. A.	N. A.	N. A.	J. Membr. Sci. 2021, 618 , 118738. 99
TFC membranes, extensively rinsed and soaked in MilliQ water for 24 h before dried in a vacuum.	SSI S-Probe Monochromatized XPS Spectrometer	N. A.	250 μm \times 1000 μm	Al K α (1486.6 eV)	N. A.	N. A.	Desalination 2009, 242 , 168-182. 60

Comment 3: The critical ingredient of the HABD method is the peak fitting of XPS spectra. Peak areas extracted from such models are used in the calculations of bond density parameter. Thus, the accuracy of the proposed approach relies on how accurate these peak models are. As Authors must be aware there is no unique way to decompose XPS spectra into component peaks (J. Vac. Sci. Technol. A 38, 061203 (2020); J. Vac. Sci. Technol. A 40, 063201 (2022)). Here, I'm missing the rigorous description of the peak fitting procedure. N 1s spectrum shown in Fig. 2B brings more questions than answers. Why are both peaks (N1 and N2) at the same binding energy if the chemical environment is not identical? How was the area ratio optimized?

Response: We thank the reviewer for the recognition and valuable suggestions. In our original N peak fitting diagram, N2 is the blue peak and N1 is the remaining area with green (including amino N and protonated amino N), without defining the peak. Our original intention was that

N1 does not need to appear as a peak in the N peak fitting result, only the proportion of N1 to N2 are needed to calculate HABD. We agree with the reviewer that this expression is misleading, and we have recounted the our XPS data process in the revised manuscript. The main points are as follows:

Following the reviewer's suggestion, N peak are divided into three peaks based on the relative peak positions of N1, N2 and N3 given in the related researches or handbooks (Tab. R3-1). Peak fitting is recommended be carried out under Gaussian–Lorentzian product pseudo-Voigt peak shape, with 60% ~ 80% Gaussian, because the XPS peaks of polymers typically have more Gaussian components (*J. Vac. Sci. Technol. A*, 2020, 38: 061203; *J. Vac. Sci. Technol. A*, 2022, 40: 063201). During this process, attention should be paid to the peak positions of the three peaks of N should be constrained within the recommended range, and their FWHM should be basically close to each other.

Tab. R3-1 Recommended peak positions for XPS peak fitting of MPD-TMC polyamide (*J. Colloid Interface Sci.*, 2002, 247: 149-158; *Science*, 2015, 348, 1347-1351; *Environ. Sci. Technol.*, 2012, 46: 852-859).

Type	Group	MPD-TMC polyamide			
		XPS peak position	α	β	γ
C1	-C-COOH	288.8 ± 0.1 eV	1	0	0
C2	-C-CONH-	288.3 ± 0.2 eV	0	1	0
C3	-C-NH ₂ , -CONH-C-,	286.3 ± 0.2 eV	0	1	1
C4	-C-COOH, -C-CONH-	285.7 ± 0.2 eV	1	1	0
C5	-C-H, -C-C, -C=C	284.8 ± 0.2 eV	1	3	2
O1	-COOH	533.1 ± 0.3 eV	2	0	0
O2	-CONH-	531.6 ± 0.4 eV	0	1	0
N1	-NH ₂	398.5 ± 0.2 eV	0	0	1
N3	-NH ₃ ⁺	401.7 ± 0.2 eV			
N2	-CONH-	400.0 eV	0	1	0

The peak fitting of C and O is carried out on the basis of strict adherence to the self-consistent relationship (*Appl. Surf. Sci.*, 2016, 387: 294), beyond paying attention to curve shape, peak position, and FWHM. Proportion of O2 and C2 can be obtained based on the self-consistent relationship between the elements of the amide bond or amino in polyamide. A single amide bond should have one C atom, one O atom, and one N atom, thus the quantities of C2, O2, and N2 should be equal. In MPD-TMC, each N atom is connected to a benzene ring C atom, implying the number of C3 should be equal to the total number of N atoms:

$$C2 = O2 = N2 \quad (R3-1)$$

$$C3 = N1 + N2 + N3 \quad (R3-2)$$

The content of C1, C4, C5, and O1 cannot be directly calculated through the peak fitting results of N. However, self-consistent relationships also provide cross-peak self-consistency constraints for them.

If there are no impurities, C and O all come from polyamide. One carboxyl group should have two O1, one C1 and connects to a C4 atom. One amide bond connects to a C4 atom as well, thus:

$$C1 = \frac{O1}{2} \quad (R3-3)$$

$$C4 = \frac{O1}{2} + N2 \quad (R3-4)$$

According to the division and statistics of our structure cells, we can also obtain statistical relationships:

$$C5 = \frac{O1}{2} + 3N2 + 2(N1 + N3) \quad (R3-5)$$

The remaining atoms that cannot be separated should be considered as impurities, rather than forcibly attributed to specific atomic type such as C1, C4, C5 or O1. For example, if the total number of C after peak fitting exceeds the sum of C1-C5 under these constraint conditions, it indicates the presence of impurity C in the XPS test. Similarly, if the total number of C is insufficient to satisfy the Formula R3-3 to R3-5 after dividing the peak of O, it indicates that some impurities in O have been separated into O1.

Fig. R3-1 and Fig. R3-2 show the peak fitting results of our self-prepared sample as an example. We use N as the standard for correction. N peak is fully attributed to different N atoms within the polyamide. When the peak conditions and cross-peak self-consistency constraints are met, the C peak does not exclusively correspond to O1~O2 and C1~C5 atoms of polyamide. The remaining portions are treated as impurities with unspecific peak positions. The quantities of various atoms (Fig.R3-2) needs to meet the self-consistent requirements of Formula R3-1 to R3-5, which is important in our method.

Fig. R3-1 Recommended peak fitting result of N, O, and C of example MPD-TMC polyamide. The polyamide example was synthesized according to the following conditions: [MPD] = 20.0 g L⁻¹, [TMC] = 1.5 g L⁻¹, reaction time = 120 s. XPS test measured through Thermo Scientific K-Alpha+ at Al K α line (1486.6 eV, 15 mA \times 15 kV), the vacuum level is 5 \times 10⁻⁹ mbar, X-ray beam spot is 300 μ m, total X-ray irradiation time is 154.1 s (68 s for full spectrum, 28.7 s for C, 30.2 s for O, and 27.2 s for N).

Fig. R3-2 Recommended peak fitting result of N, O, and C of example MPD-TMC polyamide. The polyamide example was synthesized according to the following conditions: [MPD] = 20.0 g L⁻¹, [TMC] = 1.5 g L⁻¹, reaction time = 120 s. XPS test measured through Thermo Scientific K-Alpha+ at Al K α line (1486.6 eV, 15 mA \times 15 kV), the vacuum level is 5 \times 10⁻⁹ mbar, X-ray beam spot is 300 μ m, total X-ray irradiation time is 154.1 s (68 s for full spectrum, 28.7 s for C, 30.2 s for O, and 27.2 s for N).

In our peak fitting process, the peak fitting of C and O is to obtain O2 more accurately. The most crucial step in the entire process is the peak fitting of N. The reason for choosing N for such a critical step is also due to the fact that N impurities are not common in XPS testing. Therefore, if the testing is handled properly, all N should be attributed to polyamide. This is also why we refer to this method as an N-based processing method. Any characterization method that can help obtain O2 can also become a constraint during XPS peak fitting process. In response to the reviewer's suggestion, we have added these contents in the revised manuscript on page 9 to 11. Tab. R3-1 was added revised Supporting Information marked as Tab. S1. Fig. R3-1 was added in the Fig. 2 in revised manuscript. Fig. R3-2 was added in the Supporting Information marked as Fig. S4.

After optimizing the XPS processing method, the obtained HABD data in Fig. 3 and Fig. 4 has changed accordingly. But the superiority of HABD in correlation performance remains unchanged.

Fig. 2. Graphical representation of analyzing the polyamide structure and calculating HABD using MPD-TMC polyamide as an example. The polyamide example were synthesized according to the following conditions: [MPD] = 20.0 g L⁻¹, [TMC] = 1.5 g L⁻¹, reaction time = 120 s. XPS test measured through Thermo Scientific K-Alpha+ at Al K α line (1486.6 eV, 15 mA×15 kV), the vacuum level is 5×10⁻⁹ mbar, X-ray beam spot is 300 μ m, total X-ray irradiation time is 154.1 s (68 s for full spectrum, 28.7 s for C, 30.2 s for O, and 27.2 s for N). Peak fitting was based on the recommended peak position in Tab. S1^{11,34,35}.

Fig. 3. The correlation analysis between the performance of reverse osmosis membranes with the DNC or HABD of MPD-TMC polyamides. The DNC parameter does not show a reliable correlation with the NaCl rejection (A) and water permeance (B) of reverse osmosis polyamide membranes. In contrast, HABD demonstrates a reliable correlation with the NaCl rejection (C) and water permeance (D) of those membranes (The detailed statistical data in this figure is shown in Tab. S3 and S4 in SI).

Fig. 4. The correlation analysis between the performance of nanofiltration membranes with the DNC or HABD of PIP-TMC polyamides. DNC cannot form a reliable correlation with the Na₂SO₄ rejection (A) and water permeance (B) of nanofiltration polyamide membranes. In contrast, HABD demonstrates a reliable correlation with the Na₂SO₄ rejection (C) and water permeance (D) of those membranes. (The detailed statistical data in this figure is shown in Tab. S5 and S6 in SI).

Comment 4: "In this method, peak division of the N element is the only requirement, and the proportions of O and C atoms are calculated by assigning values based on the peak splitting results of the N element. For example, ..."

If I understand it correctly Authors make the point here that the XPS peak models need to be self-consistent. For example, COHN unit should give peaks in C 1s, O 1s, and N 1s spectra such that the quantified ratios would be close to 1:1:1. IS that correct? If yes, then the following

paper can be useful in further development of this approach: Applied Surface Science 387 (2016) 294.

Response: We thank the reviewer for the valuable suggestions and recommended work. We make the point here that the XPS peak models need to be self-consistent as reviewer understand. It is necessary to ensure the cross-peak self-consistent based on polyamide structure during XPS peak fitting process, which overlooked in many recent works. We highlighted this point in revised manuscript on page 5 and pages 10-11. The recommended work has greatly inspired us and is cited on page 10 (Ref. 38).

On page 5:

“Additionally, our XPS data processing method with cross-peak self-consistency precisely determines the proportion of atoms in diverse chemical environments, ensuring accurate calculation of HABD.”

On page 10 to 11:

“The peak fitting of C and O is carried out with strict adherence to the cross-peak self-consistent relationship,³⁸ in addition to considering curve shape, peak position, and FWHM. The proportion of O2 and C2 can be determined based on the cross-peak self-consistent relationship between the elements within the amide bond or amino in polyamide. In a single amide bond, there should be one C atom, one O atom, and one N atom, resulting in equal quantities of C2, O2, and N2. In MPD-TMC, each N atom is connected to a C atom in a benzene ring, implying that the number of C3 should be equal to the total number of N atoms (Fig. 2B):

$$C2 = O2 = N2 \tag{3}$$

$$C3 = N1 + N2 + N3 \tag{4}$$

The content of C1, C4, C5, and O1, cannot be directly calculated through the peak fitting results of N. However, self-consistent relationships also impose constraints on their cross-peak self-consistency.

In the absence of impurities, both C and O originate from polyamide. A carboxyl group consist of two O1 atoms, one C1 atom and is connected to a C4 atom. Similarly, an amide bond is also connected to a C4 atom, thus:

$$C1 = \frac{O1}{2} \quad (5)$$

$$C4 = \frac{O1}{2} + N2 \quad (6)$$

According to the division and statistics of our structure cells, we can also obtain statistical relationships:

$$C5 = \frac{O1}{2} + 3N2 + 2(N1 + N3) \quad (7)$$

The remaining atoms that cannot be separated should be considered as impurities, rather than forcibly attributed to specific atomic type such as C1, C4, C5 or O1. For example, if the total number of C after peak fitting exceeds the sum of C1-C5 under these constraint conditions, it indicates the presence of impurity C in the XPS test. Similarly, if the total number of C is insufficient to satisfy the Formula 5~7 after dividing the peak of O, it indicates that some impurities in O have been separated and attributed to O1.”

Comment 5: I find the description of the new method rather unclear. I think that showing an example of all XPS core level peak models (not only N 1s) would be appreciated by NC readers. This should be further used to illustrate how the fitting procedure should be conducted for reliable results (see point 3 above).

Response: We thank the reviewer for the recognition and valuable suggestions. Following the reviewer’s suggestion, we have added all XPS core level peak fitting results in Fig. R3-1. Moreover, we emphasize stricter constraints, including cross-peak self-consistency requirement, more accurate recommendations for peak positions, peak shape, and full width at half maxima (FWHM), to limit the peak fitting process of XPS data (*J. Vac. Sci. Technol. A*, 2020, 38: 061203; *J. Vac. Sci. Technol. A*, 2022, 40: 063201).

With the hope of improving credibility and accuracy obtained HABD through XPS data, we have added these contents in the revised manuscript on pages 9 to 11. Fig. S3 was added in the Supporting Information, which shown the flowchart of XPS data analysis process with MPD-TMC polyamide as an example. Fig. S4 was added in the Supporting Information, which shown recommended atom proportion result of N, O, and C of example MPD-TMC polyamide.

On pages 9 to 11:

“To accurately determine the values of HABD, we propose a matching XPS data processing method with MPD-TMC polyamide as an example (Fig. 2 and Fig. S3 in SI). Firstly, we categorize the atoms of polyamide based on the structure cell segmentation (Fig. 2A and Tab. S1 in SI). Nitrogen (N) atoms are divided into amino N atoms (N1, in amino group), amide N atoms (N2, in amide bond), and protonated amino N atoms (N3, in amino group), while Oxygen (O) atoms are categorized into non-amide O atoms (O1) and amide O atoms (O2, in amide bond). Carbon (C) atoms are also divided into carboxyl C atoms (C1, in carboxyl group), amide C atoms (C2, in amide bond), N-related C atoms (C3, connected to the N atom, but not in amide bond), and O-related C atoms (C4, connected to O atom, but not in carboxyl group) and common C atoms (C5).

Subsequently, we utilize the N element (specifically, the binding energy of N2) for the charge shift of the full-spectrum. The N peak in polyamide is mainly contributed by the N atom of the amide bond, which minimally affected and interfered by external impurities, making its N2 peak the most easily identifiable. This step is beneficial for subsequent data analysis and peak fitting process, although it does not significantly improve the accuracy of the analysis.

Then, the N peak is divided into three peaks based on the given relative peak positions of N1, N2 and N3 as outlined in the relevant research or handbooks (Fig. 2B and Tab. S1 in SI)^{11,34,35}. Peak fitting should be carried out under Gaussian–Lorentzian product pseudo-Voigt peak shape, with a Gaussian component comprising 60% ~ 80% of the peak, as Gaussian components are commonly observed in XPS peaks of polymers^{36,37}. Throughout this process, attention should be paid to constraining the peak positions of the three N segments within the recommended range while ensuring that their full width at half maxima (FWHM) are similar.

The peak fitting of C and O is carried out with strict adherence to the cross-peak self-consistent relationship,³⁸ in addition to considering curve shape, peak position, and FWHM. The proportion of O2 and C2 can be determined based on the cross-peak self-consistent relationship between the elements within the amide bond or amino in polyamide. In a single amide bond, there should be one C atom, one O atom, and one N atom, resulting in equal quantities of C2,

O2, and N2. In MPD-TMC, each N atom is connected to a C atom in a benzene ring, implying that the number of C3 should be equal to the total number of N atoms (Fig. 2B):

$$C2 = O2 = N2 \quad (3)$$

$$C3 = N1 + N2 + N3 \quad (4)$$

The content of C1, C4, C5, and O1, cannot be directly calculated through the peak fitting results of N. However, self-consistent relationships also impose constraints on their cross-peak self-consistency.

In the absence of impurities, both C and O originate from polyamide. A carboxyl group consist of two O1 atoms, one C1 atom and is connected to a C4 atom. Similarly, an amide bond is also connected to a C4 atom, thus:

$$C1 = \frac{O1}{2} \quad (5)$$

$$C4 = \frac{O1}{2} + N2 \quad (6)$$

According to the division and statistics of our structure cells, we can also obtain statistical relationships:

$$C5 = \frac{O1}{2} + 3N2 + 2(N1 + N3) \quad (7)$$

The remaining atoms that cannot be separated should be considered as impurities, rather than forcibly attributed to specific atomic type such as C1, C4, C5 or O1. For example, if the total number of C after peak fitting exceeds the sum of C1-C5 under these constraint conditions, it indicates the presence of impurity C in the XPS test. Similarly, if the total number of C is insufficient to satisfy the Formula 5~7 after dividing the peak of O, it indicates that some impurities in O have been separated and attributed to O1.

Fig. 2B shows the peak fitting results of our self-prepared sample. We use N as the standard for correction. N peak is fully attributed to different N atoms within the polyamide. When the peak conditions and cross-peak self-consistency constraints are met, the C peak does not exclusively correspond to O1~O2 and C1~C5 atoms of polyamide. The remaining portions are treated as impurities with unspecific peak positions. The quantities of various atoms (Fig. S4) must adhere to the self-consistent requirements in Formula 3~7, which is a crucial aspect of our method.

Once the values of N1, N2, N3, O1, and O2 are obtained, HABD can be calculated by the formula in Fig. 2C (the proof process is shown in Formula S2-1~S2-4 in SI). A similar analysis process for PIP-TMC polyamide is also provide (Section 2.4 in SI).”

Fig. 2. Graphical representation of analyzing the polyamide structure and calculating HABD using MPD-TMC polyamide as an example. The polyamide example were synthesized according to the following conditions: [MPD] = 20.0 g L⁻¹, [TMC] = 1.5 g L⁻¹, reaction time = 120 s. XPS test measured through Thermo Scientific K-Alpha+ at Al K α line (1486.6 eV, 15 mA×15 kV), the vacuum level is 5×10⁻⁹ mbar, X-ray beam spot is 300 μ m, total X-ray irradiation time is 154.1 s (68 s for full spectrum, 28.7 s for C, 30.2 s for O, and 27.2 s for N). Peak fitting was based on the recommended peak position in Tab. S1^{11,34,35}.

Fig. S3. Flowchart of XPS data analysis process of MPD-TMC polyamide.

Tab. S1. Classification and the number of various atoms of α , β , and γ structure cells in MPD-TMC polyamide and recommended peak positions for XPS peak fitting of MPD-TMC polyamide¹²⁻¹⁴.

Type	Group	MPD-TMC polyamide			
		XPS peak position	α	β	γ
C1	-C-COOH	288.8 ± 0.1 eV	1	0	0
C2	-C-CONH-	288.3 ± 0.2 eV	0	1	0
C3	-C-NH ₂ , -CONH-C-,	286.3 ± 0.2 eV	0	1	1
C4	-C-COOH, -C-CONH-	285.7 ± 0.2 eV	1	1	0
C5	-C-H, -C-C, -C=C	284.8 ± 0.2 eV	1	3	2

O1	- COOH	533.1 ± 0.3 eV	2	0	0
O2	- CONH-	531.6 ± 0.4 eV	0	1	0
N1	- NH₂	398.5 ± 0.2 eV	0	0	1
N3	- NH₃⁺	401.7 ± 0.2 eV	0	0	1
N2	- CONH-	400.0 eV	0	1	0

Fig. S4. Recommended atom proportion result of N, O, and C of example MPD-TMC polyamide. The polyamide example was synthesized according to the following conditions: [MPD] = 20.0 g L⁻¹, [TMC] = 1.5 g L⁻¹, reaction time = 120 s. XPS test measured through Thermo Scientific K-Alpha+ at Al K α line (1486.6 eV, 15 mA \times 15 kV), the vacuum level is 5 \times 10⁻⁹ mbar, X-ray beam spot is 300 μ m, total X-ray irradiation time is 154.1 s (68 s for full spectrum, 28.7 s for C, 30.2 s for O, and 27.2 s for N).

Reviewers' Comments:

Reviewer #1:

Remarks to the Author:

The authors have done tremendous works to revise the paper and reply our comments. In my view, the HARD value may give some useful indications to explain transport behavior of RO membranes. I agree with other reviewers that the IP process is too complicated and transport property of the TFC membranes is very difficult to predict. However, this model may give some trend among different RO membranes. So, it has value to publish and check by other researchers for its validity. At this point, I suggest to publish this paper and hope more relative works to be published.

Reviewer #2:

Remarks to the Author:

The revised manuscript has been adequately improved and completed, following the reviewers' comments.

On the one hand, new data are provided on the structure of the polyamide film and its correlation with the HABD parameter, and the effect of film thickness in membrane performance is also discussed.

They also propose two parameters (HCD and HAD) to estimate the surface charge of the polyamide film and highlight the importance of Donnan effect in the membrane selectivity.

On the other hand, sufficient details are provided about the fitting of the XPS spectra, and the method followed is correct. The flowchart in Fig. S3 is very illustrative of the XPS fitting method followed to obtain the HABD. The calculation of the HABD (and HCD and HAD) parameter from XPS results is also sufficiently explained. Details of how they have calculated the HABD parameter from the bibliography data are now explained, and the XPS experimental conditions is now summarized in the supporting information. In this way, the reliability of data used for the statistical analysis, which in my opinion was already well done in the first version of the manuscript, has improved substantially.

In summary, I congratulate the authors for the work carried out, which has led to a substantial improvement in the quality of the manuscript, and I recommend its publication in Nature Communications.

Some minor issues are:

To improve readability, I recommend including the references in Tab. S3 and S5 only by the reference number, as it is already done for example in Tab. S6.

In Tab. S9, revised the title cell of column 7: "Testing conditions of zeta potential testing conditions"

Reviewer #3:

Remarks to the Author:

All of my comments were properly addressed.

Point-to-point response to the reviewers

Response to the comments by Reviewer 1

Comment: The authors have done tremendous works to revise the paper and reply our comments. In my view, the HARD value may give some useful indications to explain transport behavior of RO membranes. I agree with other reviewers that the IP process is too complicated and transport property of the TFC membranes is very difficult to predict. However, this model may give some trend among different RO membranes. So, it has value to publish and check by other researchers for its validity. At this point, I suggest to publish this paper and hope more relative works to be published.

Response: We appreciate the reviewer's recognition of our work. The reviewer's comments have significantly enhanced the validity and reliability of our work.

Response to the comments by Reviewer 2

Comment 1: The revised manuscript has been adequately improved and completed, following the reviewers' comments.

On the one hand, new data are provided on the structure of the polyamide film and its correlation with the HADB parameter, and the effect of film thickness in membrane performance is also discussed. They also propose two parameters (HCD and HAD) to estimate the surface charge of the polyamide film and highlight the importance of Donnan effect in the membrane selectivity.

On the other hand, sufficient details are provided about the fitting of the XPS spectra, and the method followed is correct. The flowchart in Fig. S3 is very illustrative of the XPS fitting method followed to obtain the HADB. The calculation of the HADB (and HCD and HAD) parameter from XPS results is also sufficiently explained. Details of how they have calculated the HADB parameter from the bibliography data are now explained, and the XPS experimental conditions is now summarized in the supporting information. In this way, the reliability of data used for the statistical analysis, which in my opinion was already well done in the first version of the manuscript, has improved substantially.

In summary, I congratulate the authors for the work carried out, which has led to a substantial improvement in the quality of the manuscript, and I recommend its publication in Nature Communications.

Response: We thank the reviewer's recognition and all comments about our work, which have significantly enhanced the validity and reliability of our work.

Comment 2: To improve readability, I recommend including the references in Tab. S3 and S5 only by the reference number, as it is already done for example in Tab. S6.

Response: We thank the reviewer's recommendation. We have made relevant changes to the Table S3, S5, Table S10 and S11 (Supplementary Table 3, 5, 10, and 11). To further enhance readability, we changed the Table S10 and S11 (Supplementary Table 10 and 11) to horizontal layout, and provided an extra Excel document to display all the statistical analysis data involved in this work (Supplementary Table 3 to 11).

Comment 3: In Tab. S9, revised the title cell of column 7: "Testing conditions of zeta potential testing conditions"

Response: We thank the reviewer's comment. We have corrected the relevant omissions and checked the wording throughout the manuscript and Supplementary Information.

Response to the comments by Reviewer 3

Comment: All of my comments were properly addressed.

Response: We appreciate the reviewer's approval of our work. The reviewer's suggestions have significantly helped us improve the validity and reliability of our work.